# A widely distributed metalloenzyme class enables gut microbial metabolism of host- and diet-derived catechols

**Vayu Maini Rekdal[1†], Paola Nol Bernadino[2,3†], Michael U Luescher[1†], Sina Kiamehr[1], Chip Le[1], Jordan E Bisanz[4], Peter J Turnbaugh[4,5], Elizabeth N Bess[2,3], Emily P Balskus[1*]**

[1]Department of Chemistry and Chemical Biology, Harvard University, Cambridge, United States; [2]Department of Chemistry and Molecular Biology, University of California, Irvine, Irvine, United States; [3]Department of Chemistry and Molecular Biochemistry, University of California, Irvine, Irvine, United States; [4]Department of Microbiology and Immunology, University of California, San Francisco, San Francisco, United States; [5]Chan Zuckerberg Biohub, San Francisco, United States

**Abstract** Catechol dehydroxylation is a central chemical transformation in the gut microbial metabolism of plant- and host-derived small molecules. However, the molecular basis for this transformation and its distribution among gut microorganisms are poorly understood. Here, we characterize a molybdenum-dependent enzyme from the human gut bacterium *Eggerthella lenta* that dehydroxylates catecholamine neurotransmitters. Our findings suggest that this activity enables *E. lenta* to use dopamine as an electron acceptor. We also identify candidate dehydroxylases that metabolize additional host- and plant-derived catechols. These dehydroxylases belong to a distinct group of largely uncharacterized molybdenum-dependent enzymes that likely mediate primary and secondary metabolism in multiple environments. Finally, we observe catechol dehydroxylation in the gut microbiotas of diverse mammals, confirming the presence of this chemistry in habitats beyond the human gut. These results suggest that the chemical strategies that mediate metabolism and interactions in the human gut are relevant to a broad range of species and habitats.

**\*For correspondence:**
balskus@chemistry.harvard.edu

[†]These authors contributed equally to this work

## Introduction

The human gastrointestinal tract is one of the densest microbial habitats on Earth. Possessing 150-fold more genes than the human genome, the trillions of organisms that make up this community (the human gut microbiota) harbor metabolic capabilities that expand the range of chemistry taking place in the body (*Koppel et al., 2017*; *Qin et al., 2010*; *Sender et al., 2016*). Microbial metabolism affects host nutrition and health by breaking down otherwise inaccessible carbohydrates, biosynthesizing essential vitamins, and transforming endogenous and exogenous small molecules into bioactive metabolites (*Koppel and Balskus, 2016*). Gut microbial activities can also vary significantly between individuals, affecting the toxicity and efficacy of drugs (*Zimmermann et al., 2019*; *Koppel et al., 2018*; *Gopalakrishnan et al., 2018*; *Wallace et al., 2010*; *Haiser et al., 2013*), susceptibility to infection (*Buffie et al., 2015*; *Devlin and Fischbach, 2015*), and host metabolism (*Yao et al., 2018*; *Romano et al., 2017*). To decipher the biological roles of gut microbial metabolism, it is critical that we uncover the enzymes responsible for prominent transformations. This will not only increase the information gained from microbiome sequencing data but may also illuminate strategies for manipulating and studying microbial functions. Yet, the vast majority of gut microbial metabolic reactions have not yet been linked to specific enzymes.

**eLife digest** Inside the human gut there are trillions of bacteria. These microbes are critical for breaking down and modifying molecules that the body consumes (such as nutrients and drugs) and produces (such as hormones). Although metabolizing these molecules is known to impact health and disease, little is known about the specific components, such as the genes and enzymes, involved in these reactions.

A prominent microbial reaction in the gut metabolizes molecules by removing a hydroxyl group from an aromatic ring and replacing it with a hydrogen atom. This chemical reaction influences the fate of dietary compounds, clinically used drugs and chemicals which transmit signals between nerves (neurotransmitters). But even though this reaction was discovered over 50 years ago, it remained unknown which microbial enzymes are directly responsible for this metabolism.

In 2019, researchers discovered the human gut bacteria *Eggerthella lenta* produces an enzyme named Dadh that can remove a hydroxyl group from the neurotransmitter dopamine. Now, Maini Rekdal et al. – including many of the researchers involved in the 2019 study – have used a range of different experiments to further characterize this enzyme and see if it can break down molecules other than dopamine. This revealed that Dadh specifically degrades dopamine, and this process promotes *E. lenta* growth.

Next, Maini Rekdal et al. uncovered a group of enzymes that had similar characteristics to Dadh and could metabolize molecules other than dopamine, including molecules derived from plants and nutrients in food. These Dadh-like enzymes were found not only in the guts of humans, but in other organisms and environments, including the soil, ocean and plants.

Plant-derived molecules are associated with human health, and the discovery of the enzymes that break down these products could provide new insights into the health effects of plant-based foods. In addition, the finding that gut bacteria harbor a dopamine metabolizing enzyme has implications for the interaction between the gut microbiome and the nervous system, which has been linked to human health and disease. These newly discovered enzymes are also involved in metabolic reactions outside the human body. Future work investigating the mechanisms and outputs of these reactions could improve current strategies for degrading pollutants and producing medically useful molecules.

A prominent but poorly understood gut microbial activity is the dehydroxylation of catechols (1,2-dihydroxylated aromatic rings), a structural motif commonly found in a diverse range of compounds that includes dietary phytochemicals, host neurotransmitters, clinically used drugs, and microbial siderophores (*Wilson et al., 2016*; *Ozdal et al., 2016*; *Yang et al., 2007*) (*Figure 1A*). Discovered over six decades ago, catechol dehydroxylation is a uniquely microbial reaction that selectively replaces the *para* hydroxyl group of the catechol with a hydrogen atom (*Scheline et al., 1960*) (*Figure 1A*). This reaction is particularly challenging due to the stability of the aromatic ring system. Prominent substrates for microbial dehydroxylation include the drug fostamatinib (*Sweeny et al., 2010*), the catecholamine neurotransmitters norepinephrine and dopamine (*Smith et al., 1964*; *Sandler et al., 1971*), the phytochemicals ellagic acid (found in nuts and berries), caffeic acid (a universal lignin precursor in plants), and catechin (present in chocolate and tea) (*Peppercorn and Goldman, 1972*; *Cerdá et al., 2005*; *Takagaki and Nanjo, 2010*) (*Figure 1B*). Dehydroxylation alters the bioactivity of the catechol compound (*Kim et al., 2016*; *Ryu et al., 2016*) and produces metabolites that act both locally in the gut and systemically to influence human health and disease (*Sweeny et al., 2010*; *Ryu et al., 2016*; *Kang et al., 2016*; *Pietinen et al., 2001*; *Mabrok et al., 2012*; *Maini Rekdal et al., 2019*; *Singh et al., 2019*). However, the gut microbial enzymes responsible for catechol dehydroxylation have remained largely unknown.

We recently reported the discovery of a catechol dehydroxylating enzyme from the prevalent human gut Actinobacterium *Eggerthella lenta*. This enzyme participates in an interspecies gut microbial pathway that degrades the Parkinson's disease medication L-dopa by catalyzing the regioselective *p*-dehydroxylation of dopamine to *m*-tyramine (*Maini Rekdal et al., 2019*). To identify the enzyme, we grew *E. lenta* strain A2 with and without dopamine and used RNA sequencing (RNA-seq) to find genes induced by dopamine. Only 15 genes were significantly upregulated in the presence of dopamine, including a putative molybdenum-dependent enzyme that was induced >2500

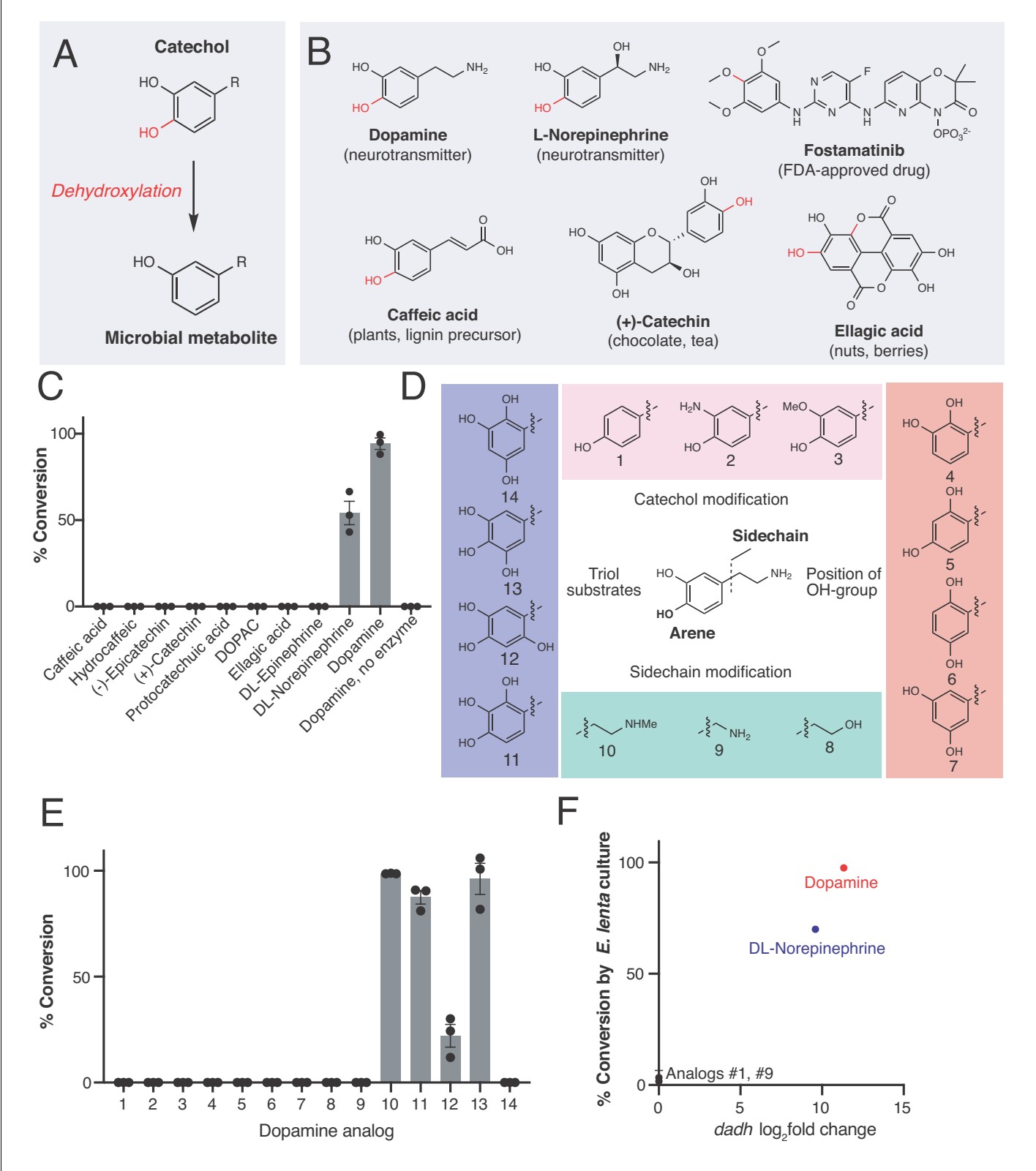

**Figure 1.** An enzyme from the prevalent human gut Actinobacterium *Eggerthella lenta* specifically metabolizes catecholamines that are available in the gut. (**A**) The catechol structural motif is dehydroxylated by the gut microbiota. (**B**) Examples of catechols known to be dehydroxylated by gut microbes. Red indicates the carbon-oxygen bond that is broken or the hydroxyl group that is removed in the dehydroxylation reaction. (**C**) Activity of natively purified Dadh towards a panel of physiologically relevant catechol substrates. Enzyme (0.1 μM) was incubated with substrate (500 μM) for 22 hr at room

*Figure 1 continued on next page*

*Figure 1 continued*

temperature, followed by analysis using LC-MS. Bars represent the mean ±the standard error (SEM) of three biological replicates (enzyme reactions). This experiment was repeated three times. See *Supplementary file 1a* for the full chemical structures. (D) Dopamine analogs evaluated in this study. (E) Activity of natively purified Dadh towards dopamine analogs in C). Enzyme (0.1 µM) was incubated with substrate (500 µM) for 22 hr at room temperature, followed by analysis using LC-MS. Bars represent the mean ±the SEM of three biological replicates (enzyme reactions). See *Supplementary file 1b* for the full chemical structures. This experiment was performed three times. (F) Transcriptional induction and whole-cell dehydroxylation activity of *E. lenta* A2 in response to dopamine and a subset of dopamine analogs (500 µM each). Transcriptional induction was assessed using RNA-seq, with the fold induction shown on the x-axis (foldchange >2, FDR < 0.01). To assess whole-cell metabolism, *E. lenta* was grown anaerobically for 48 hr in BHI medium with 500 µM of each substrate, and the culture supernatant was analyzed for dehydroxylated metabolites using LC-MS. RNA-sequencing data represent the log2fold change from n = 3 independent cultures for each condition (compound/vehicle). The metabolism data represent the mean ±the SEM of three biological replicates (independent bacterial cultures). The culturing and analysis of metabolism was performed twice, while RNA-sequencing was done once. All raw data from *Figure 1* can be found in *Figure 1—source data 1*.

The online version of this article includes the following source data and figure supplement(s) for figure 1:

**Source data 1.** Data from Dadh enzyme reactions and from studies of dadh regulation (*Figure 1*).
**Figure supplement 1.** SDS-PAGE of natively purified dopamine dehydroxylase from *E. lenta* A2.
**Figure supplement 2.** Proposed mechanism for dopamine dehydroxylation by *E. lenta* A2 dopamine dehydroxylase.

fold. Hypothesizing this gene encoded the dopamine dehydroxylase, we purified the enzyme from *E. lenta* and confirmed its activity in vitro. Dopamine dehydroxylase (Dadh) is predicted to bind bis-molybdopterin guanine nucleotide (bis-MGD), a complex metallocofactor that contains a catalytically essential molybdenum atom (*Hille et al., 2014*). Our previous work illuminated a role for Dadh in dopamine metabolism by pure strains and complex communities. Here, we sought to explore the substrate scope of Dadh and its broader role in catechol dehydroxylation by the gut microbiota.

## Results

### A molybdenum-dependent enzyme from *Eggerthella lenta* specifically metabolizes catecholamines that are available in the gut

Because the human gut microbiota metabolizes a range of catecholic compounds (*Figure 1B*), we first investigated whether the recently discovered Dadh possessed promiscuous dehydroxylase activity. We evaluated the reactivity of natively purified *E. lenta* A2 Dadh towards a panel of established or potential host- and diet-derived catechol substrates (*Supplementary file 1a* and *Figure 1—figure supplement 1*). This enzyme displayed a narrow substrate scope, metabolizing only dopamine and the structurally related neurotransmitter norepinephrine, which differ only by the presence of a benzylic hydroxyl group (*Figure 1C*). To identify the elements necessary for substrate recognition by Dadh, we profiled its activity towards synthetic and commercially available dopamine analogs (*Figure 1D*, *Figure 1—figure supplement 1*, and *Supplementary file 1b*). We found that Dadh tolerated only minor modifications to the dopamine scaffold, including a single *N*-methylation and the presence of additional hydroxyl groups on the aromatic ring (*Figure 1E*). The catechol moiety was absolutely necessary for activity, and dehydroxylation required that at least one hydroxyl group be in the *para* position relative to the aminoethyl substituent. These data demonstrated that Dadh specifically recognizes the catecholamine scaffold.

This result prompted us to explore whether the transcriptional regulation of Dadh displayed similar specificity. Thus, we cultured *E. lenta* A2 in the presence of a subset of the dopamine analogs that we had tested in the previous experiment, measured dehydroxylation using liquid chromatography-mass spectrometry (LC-MS), and profiled the global transcriptome using RNA-seq. We found that the regulation of *dadh* was also specific for the catecholamine scaffold (*Figure 1F*, *Supplementary file 1c*). While the catecholamines dopamine and norepinephrine induced *dadh* expression and were dehydroxylated by *E. lenta*, analogs lacking the catechol (analog **1** in *Figure 1D*) or having a shorter side chain (analog **9** in *Figure 1D*) did not induce a transcriptional or metabolic response (*Figure 1F*, *Supplementary file 1c*) (*Maini Rekdal et al., 2019*). Together with our biochemical results, these transcriptional data suggest that Dadh may have evolved for the purpose of catecholamine neurotransmitter metabolism in *E. lenta*. We propose that dopamine is an endogenous substrate of this enzyme, because it was the best substrate both in vitro and in vivo,

induced the highest levels of expression in *E. lenta*, and is produced at substantial levels within the human gastrointestinal tract (*Eisenhofer et al., 1997*).

In addition to uncovering a preference for the catecholamine scaffold, the substrate scope of Dadh reveals potential mechanistic distinctions between this enzyme and the only other biochemically characterized reductive aromatic dehydroxylase, 4-hydroxybenzoyl Coenzyme A (CoA) reductase (4-HCBR) (*Unciuleac et al., 2004*). 4-HCBR is a distinct molybdenum dependent-enzyme containing a monomeric molybdopterin co-factor that uses a Birch reduction-like mechanism to remove a single aromatic hydroxyl group from 4-hydroxybenzoyl CoA. While 4-HCBR requires an electron-withdrawing thioester group to stabilize radical anion intermediates (*Unciuleac et al., 2004*), Dadh does not require an electron-withdrawing substituent and can tolerate additional electron-donating hydroxyl groups (*Figure 1D and E*, analogs **11–13**). We preliminarily propose a mechanism for Dadh in which the dopamine *p*-hydroxyl group coordinates to the molybdenum center. This could be followed by tautomerization of the *m*-hydroxyl group to a ketone with protonation of the adjacent carbon atom. Oxygen atom transfer to molybdenum could be accompanied by rearomatization, providing the dehydroxylated product (*Figure 1—figure supplement 2*). Our proposal is consistent with the postulated mechanisms of other oxygen transfer reactions catalyzed by bis-MGD enzymes (*Tenbrink et al., 2011*; *Hille et al., 2014*).

## Dopamine promotes gut bacterial growth by serving as an alternative electron acceptor

The specificity of Dadh for dopamine suggested this metabolic activity might have an important physiological role in *E. lenta*. We noted the chemical parallels between catechol dehydroxylation and reductive dehalogenation, a metabolic process in which halogenated aromatics serve as alternative electron acceptors in certain environmental bacteria (*Holliger et al., 1998*). This insight inspired the hypothesis that dopamine dehydroxylation could serve a similar role in gut bacteria. While we observed no growth benefit when *E. lenta* was grown in complex BHI medium containing dopamine (*Figure 2—figure supplement 1*), we found that including dopamine in a minimal medium lacking electron acceptors (basal medium) increased the endpoint optical density of *E. lenta* cultures (*Figure 2A*). This growth-promoting effect was only observed in dopamine-metabolizing *E. lenta* strains, as non-metabolizing strains that express an apparently inactive enzyme (*Maini Rekdal et al., 2019*) did not gain a growth advantage (*Figure 2A* and *Figure 2—figure supplement 2*). The effect of dopamine on *E. lenta* contrasts with recent studies of digoxin, a drug that is reduced by *E. lenta* without impacting growth in the same medium (*Koppel et al., 2018*).

We further investigated the relationship between dopamine and bacterial growth in the metabolizing strain *E. lenta* A2. The growth increase observed in response to dopamine was dose-dependent (*Figure 2—figure supplement 3*), mirrored the effects of the known electron acceptors DMSO and nitrate (*Koppel et al., 2018*; *Sperry and Wilkins, 1976*), and did not derive from the product of dopamine dehydroxylation, *m*-tyramine (*Figure 2B*). Additionally, the growth benefit was directly tied to dopamine dehydroxylation. Inclusion of tungstate in the growth medium, which inactivates the big-MGD cofactor of Dadh, blocked metabolism and inhibited the growth increase. In contrast, inclusion of molybdate in the growth medium did not impact growth or metabolism (*Figure 2B* and *Figure 2—figure supplement 4*). Molybdate and tungstate alone did not impact *E. lenta* A2 growth in the basal medium (*Figure 2—figure supplement 5*). Taken together, these results indicate that active metabolism of dopamine provides a growth advantage to *E. lenta*, likely by serving as an alternative electron acceptor.

We next examined whether dopamine could promote *E. lenta* growth in microbial communities. First, we competed dopamine metabolizing and non-metabolizing *E. lenta* strains in minimal medium. *E. lenta* is genetically intractable, preventing the use of engineered plasmids encoding defined fluorescence or antibiotic resistance as markers of strain identity. Instead, we took advantage of intrinsic differences in tetracyline (Tet) resistance to differentiate the closely related strains in pairwise competitions (*Bisanz et al., 2018*). Inclusion of dopamine in growth medium significantly increased the proportion of the metabolizer relatively to the non-metabolizer in this competition experiment (p<0.001, two-tailed unpaired t-test) (*Figure 2C* and *Figure 2—figure supplement 6*). This was driven by the growth increase of the metabolizer rather than a decrease in the non-metabolizer (*Figure 2C* and *Figure 2—figure supplement 6*).

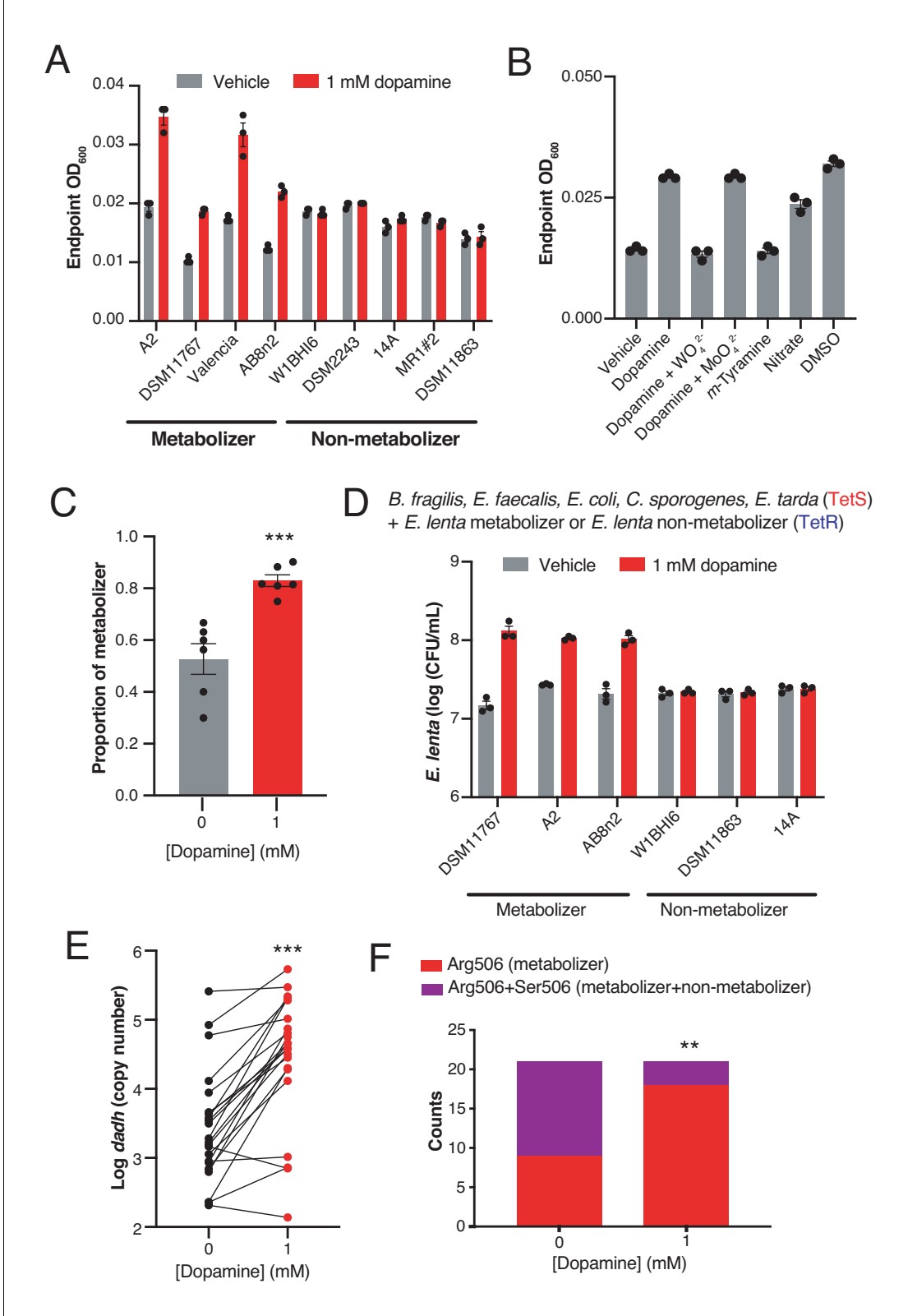

**Figure 2.** Dopamine increases gut bacterial growth by serving as an alternative electron acceptor. (A) Growth of dopamine metabolizing and non-metabolizing *E. lenta* strains in minimal medium limited in electron acceptors (basal medium) containing 10 mM acetate. Strains were grown anaerobically for 48–72 hr at 37°C before growth was assessed. Bars represent the mean ±the SEM of three biological replicates (bacterial cultures). The experiment was performed once. (B) Tungstate inhibits of growth and dopamine metabolism by *E. lenta* A2 in basal medium containing 10 mM
*Figure 2 continued on next page*

*Figure 2 continued*

acetate. *E. lenta* was grown anaerobically for 48 hr at 37°C. Dopamine, *m*-tyramine, and nitrate were added to a final concentration of 1 mM, while DMSO was added to a final concentration of 14 mM at the time of inoculation. Tungstate ($WO_4^{2-}$) and molybdate ($MoO_4^{2-}$) were added to a final concentration of 0.5 mM. Bars represent the mean ±the SEM of three biological replicates (bacterial cultures). The experiment was performed twice. (C) Competition of dopamine metabolizing (Valencia) and non-metabolizing (W1BHI6) *E. lenta* strains in basal medium containing 10 mM acetate. Strains were grown together for 72 hr at 37°C and were then plated on BHI medium. Antibiotic resistance was used to determine strain identity. Bars represent the mean ±the SEM of six biological replicates (bacterial cultures). (***p=0.0007, two-tailed unpaired t-test). The experiment was performed twice. (D) Growth of defined gut bacterial consortia containing dopamine metabolizing and non-metabolizing *E. lenta* strains in basal medium containing 10 mM acetate. Tetracycline resistant (TetR) *E. lenta* strains were grown with tetracycline sensitive (TetS) gut isolates for 48 hr at 37°C. Plating on BHI medium containing tetracycline allowed enumeration of *E. lenta*. Bars represent the mean ±the SEM of three biological replicates (bacterial cultures). The experiment was performed twice. (E) Abundance of *dadh* in complex human gut communities cultured ex vivo. Samples from unrelated individuals (n = 24) were grown for 72 hr at 37°C in basal medium containing 10 mM acetate with or without dopamine and qPCR was used to assess abundance of *dadh*. Two individuals were excluded from this analysis as they did not demonstrate quantitative metabolism of dopamine after incubation. Each point represents a different individual. Lines connect data from the same individual between the two conditions. (***p=0.0005, two-tailed unpaired t-test, n = 22 samples per group). The experiment was performed twice. (F) Counts of *dadh* variants in the presence and absence of dopamine. The same gDNA used in E) was used to amplify full-length *dadh* and determine the SNP status at position 506 using Sanger sequencing. As in panel E, two individuals were removed prior to analysis as they did not demonstrate quantitative metabolism of dopamine after incubation. In addition, one individual was not included in this analysis due to failure of obtaining high quality sequencing data. (**p=0.008, Fisher's exact test, n = 9 CGC samples and n = 12 CGC/AGC samples for vehicle; n = 18 CGC samples and n = 3 CGC/AGC samples for dopamine). The sequencing was performed once. All data , and details of the statistical tests, can be found in *Figure 2—source data 1*.

The online version of this article includes the following source data and figure supplement(s) for figure 2:

**Source data 1.** Growth data from studies of impact of dopamine on E. lenta growth in basal medium lacking electron acceptors (*Figure 2*).
**Figure supplement 1.** Growth of *E. lenta* A2 in BHI medium with and without dopamine.
**Figure supplement 2.** Dehydroxylation of dopamine by *E. lenta* strains grown in in basal medium.
**Figure supplement 3.** Dose dependence of dopamine-promoted growth of *E. lenta* A2 in basal medium.
**Figure supplement 4.** Dehydroxylation of dopamine by *E. lenta* A2 in the presence of tungstate and molybdate in basal medium.
**Figure supplement 5.** Impact of tungstate and molybdate on *E. lenta* A2 growth in basal medium.
**Figure supplement 6.** CFU counts of *E. lenta* strains W1BHI6 and Valencia co-cultured in basal medium with and without dopamine.
**Figure supplement 7.** *E. lenta* qPCR abundance in human fecal samples cultured with and without dopamine ex vivo.

Next, we explored the impact of dopamine on Tet-resistant *E. lenta* in the presence of a defined bacterial community representing the major phylogenetic diversity in the human gut (*Figure 2D* and *Supplementary file 1d*) (*Devlin et al., 2016*; *Romano et al., 2015*). We found that including dopamine in the medium boosted the growth of metabolizers by an order of magnitude while non-metabolizing strains did not gain a growth advantage (*Figure 2D*). Finally, we evaluated the impact of dopamine on *E. lenta* strains present in complex human gut microbiotas. We cultured fecal samples from 24 unrelated subjects ex vivo in the presence and absence of dopamine and used qPCR to assess the abundance of *E. lenta* and *dadh*. We found that both *dadh* and *E. lenta* significantly increased by an order of magnitude in cultures containing dopamine (p<0.005, two-tailed unpaired t-test) (*Figure 2E*) (*Figure 2—figure supplement 7*). Finally, we amplified the full length *dadh* gene from these cultures and sequenced the region harboring the SNP that distinguishes metabolizing and non-metabolizing strains (*Maini Rekdal et al., 2019*). These assays indicated that the increase in *dadh* abundance in the complex communities was accompanied by a shift from a mixture of inactive and active *dadh* variants to a dominance of the metabolizing R506 variant (p<0.01, Fisher's exact test) (*Figure 2F*). Finally, we noticed in these growth assays that a small number of samples did not display an increase in *E. lenta* or *dadh* abundance (n = 4 and n = 3 samples, respectively) (*Figure 2E* and *Figure 2—figure supplement 7*). While the factors influencing this outcome are unclear, they could include the possibility that these specific communities support the growth of *E. lenta* in other ways that such that dopamine metabolism does not provide any additional advantage, that these samples contain inhibitory factors, or that organisms not targeted by our primers were responsible for metabolism. Altogether, these results are consistent with the hypothesis that dopamine dehydroxylation can increase the fitness of metabolizing *E. lenta* strains in microbial communities.

## A screen of human gut Actinobacteria uncovers dehydroxylation of host- and plant-derived catechols

Having uncovered Dadh's specialized role in gut bacterial dopamine metabolism, we sought to identify additional gut bacterial strains and enzymes that could dehydroxylate other catechol substrates. Among human gut bacteria, only *Eggerthella* and closely related members of the Actinobacteria phylum have been reported to perform catechol dehydroxylation. For example, *Eggerthella* metabolizes dopamine (*Maini Rekdal et al., 2019*) and (+)-catechin (*Takagaki and Nanjo, 2015*), while related *Gordonibacter* species dehydroxylate ellagic acid (*Selma et al., 2014*) and didemethylsecoisolariciresinol (dmSECO), an intermediate in the multi-step biosynthesis of the anti-cancer metabolite enterodiol (*Bess et al., 2020*). These reports suggest that Actinobacteria could be a promising starting point to identify new dehydroxylating strains and enzymes. Thus, we screened a library of related gut Actinobacteria (*Bisanz et al., 2018*) (n = 3 replicates for each strain) for metabolism of a range of compounds relevant in the human gut, including plant- and host-derived small molecules, bacterial siderophores, and FDA-approved catecholic drugs (*Wilson et al., 2016*; *Ozdal et al., 2016*; *Yang et al., 2007*) (*Supplementary file 1e*) (*Figure 3A*). We initially used a colorimetric assay that detects catechols to assess metabolism, which allowed us to rapidly screen for potential catechol depletion across the collection of 25 strains.

We observed complete depletion of several host and diet-derived catechols in this initial screen (*Figure 3—figure supplement 1*). We chose to focus on the dehydroxylation of hydrocaffeic acid, (+)-catechin, and DOPAC for further characterization, repeating the incubations with these compounds and using LC-MS/MS to confirm the production of dehydroxylated metabolites. This analysis showed that both DOPAC and hydrocaffeic acid are directly dehydroxylated by members of this library, while (+)-catechin undergoes benzyl ether reduction followed by dehydroxylation into derivative **2**, as has been observed previously (*Takagaki and Nanjo, 2015*) (*Figure 3*). While (+)-catechin metabolism has been previously linked to *Eggerthella* (*Takagaki and Nanjo, 2015*), the dehydroxylation of DOPAC and hydrocaffeic acid has only been previously observed by complex gut microbiota communities (*Scheline et al., 1960*; *Peppercorn and Goldman, 1972*). The variability in these activities across closely related gut bacterial strains suggests that distinct enzymes might dehydroxylate different catechols.

## Gut Actinobacteria dehydroxylate individual catechols using distinct enzymes

We next sought to determine the molecular basis of the dehydroxylation reactions examined above. To test the hypothesis specific rather than promiscuous enzymes were involved, we first established that dehydroxylation is an inducible activity in *Gordonibacter* and *Eggerthella* strains (*Figure 4—figure supplement 1*). This allowed us to use the dehydroxylase activity of cell lysates as a proxy for transcriptional induction and a means of examining dehydroxylase activity. We grew *E. lenta* A2 in the presence of (+)-catechin, hydrocaffeic acid, and dopamine, and grew *G. pamelaeae* 3C in the presence of DOPAC. We then screened each anaerobic lysate for its activity towards all of these substrates. Consistent with our prediction, each lysate quantitively dehydroxylated only the catechol substrate with which the strain had been grown. While the *E. lenta* lysates did not display any promiscuity (*Figure 4A*), cell lysate from *G. pamelaeae* grown in the presence of DOPAC displayed reduced activity (<45% conversion) towards hydrocaffeic acid, which structurally resembles DOPAC (*Figure 4B*). Overall, these results suggest that different catechol substrates induce the expression of distinct dehydroxylase enzymes that are specific in their activity and transcriptional regulation. We expected that these enzymes would likely resemble Dadh, as only molybdenum-dependent enzymes are known to catalyze aromatic dehydroxylation (*Maini Rekdal et al., 2019*; *Hille et al., 2014*; *Unciuleac et al., 2004*).

To identify the molecular basis of (+)-catechin and hydrocaffeic acid dehydroxylation in *E. lenta* A2, we turned to RNA-seq, the approach that we used previously to identify the dopamine dehydroxylase (*Maini Rekdal et al., 2019*). We grew *E. lenta* A2 to early exponential phase and then added each catechol substrate, harvesting the cells after ~1.5 hr of induction. Hydrocaffeic acid and (+)-catechin each upregulated a number of genes (21 and 43, respectively), including two predicted molybdenum-dependent enzymes (*Supplementary files 2a and 2b*). While one of these predicted molybdenum-dependent enzymes was among the highest upregulated genes in response to each

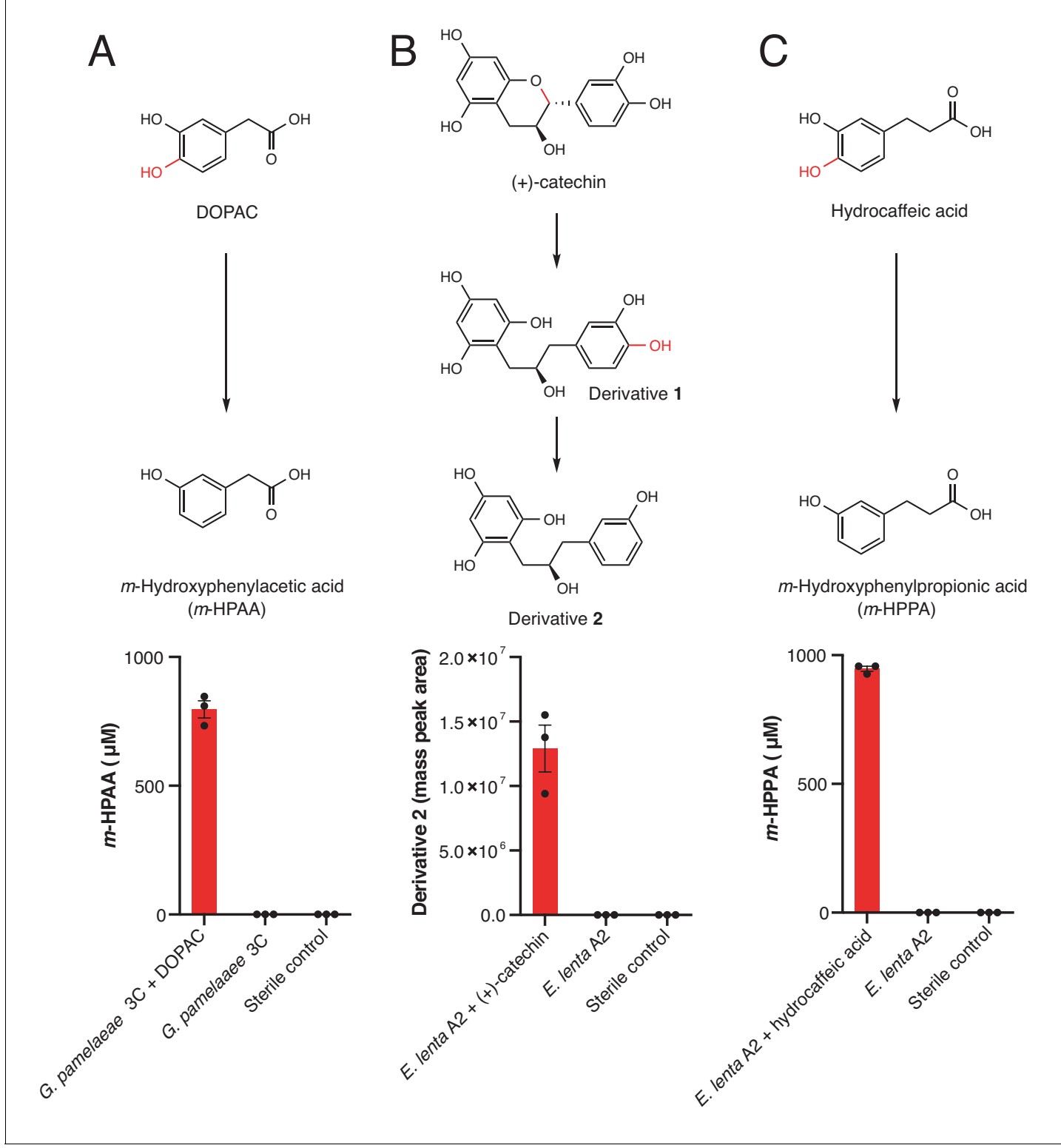

**Figure 3.** Dehydroxylation of DOPAC, (+)-catechin, and hydrocaffeic acid by *Gordonibacter pamelaeae* 3C and *Eggerthella lenta* A2. (A-C) Pathways for metabolism of (A) DOPAC, (B) (+)-catechin, (C) and hydrocaffeic acid by human gut Actinobacteria. While DOPAC and hydrocaffeic acid are dehydroxylated directly, (+)-catechin metabolism proceeds by initial benzyl ether reduction followed by dehydroxylation of the catecholic derivative. (A) Metabolism of DOPAC by *G. pamelaeae* 3C. This strain was grown in BHI medium with and without 1 mM DOPAC for 48 hr at 37°C. Metabolism was assessed using LC-MS/MS. Bars represent the mean ±the SEM concentration of the metabolite *m*-hydroxyphenylacetic acid (*m*-HPAA) resulting from direct DOPAC dehydroxylation (three biological replicates, e.g. bacterial cultures). The experiment was performed twice. (B) (+)-catechin metabolism by

*Figure 3 continued on next page*

*Figure 3 continued*

*E. lenta* A2. This strain was grown in BHI medium with and without 1 mM (+)-catechin for 48 hr at 37°C. Metabolism was assessed using high resolution LC-MS. Bars represent the mean ±the SEM mass peak area of the Extracted Ion Chromatogram (EIC) for the dehydroxylated derivative **2** shown above the bar graph in B) (three biological replicates, e.g. bacterial cultures). Due to the absence of an authentic standard, integrated peak area of the high-resolution mass is displayed. The experiment was performed twice. C) Metabolism of hydrocaffeic acid by *E. lenta* A2. This strain was grown in BHI medium with and without 1 mM hydrocaffeic acid for 48 hr at 37°C. Metabolism was assessed using LC-MS/MS. Bars represent the mean ±the SEM concentration of *m*-hydroxyphenylpropionic acid (*m*-HPPA) resulting from the direct dehydroxylation of hydrocaffeic acid (three biological replicates, e.g. bacterial cultures). The experiment was performed twice. All data can be found in *Figure 3—source data 1*.

The online version of this article includes the following source data and figure supplement(s) for figure 3:

**Source data 1.** Metabolism data from incubations of G. pamelaeae 3C with DOPAC and from E. lenta A2 with hydrocaffeic acid and (+)-catechin (*Figure 3*).

**Figure supplement 1.** Colorimetric screen for catechol dehydroxylation by human gut Actinobacteria.

substrate (450-fold upregulated in response to catechin, >2000 fold with hydrocaffeic acid), the expression of the other enzyme was only increased 3-fold relative to the vehicle. Thus, we propose that the most highly upregulated molybdenum-dependent enzyme in each dataset is the most reasonable candidate dehydroxylase. The candidate hydrocaffeic acid dehydroxylase (Elenta-A2_02815, named *hcdh*) shares 35.3% amino acid identity with Dadh, while the candidate (+)-catechin dehydroxylase (E. lenta-A2_00577, named *cadh*) shares 50.9% amino acid identity with Dadh (*Supplementary files 2a-2c*).

To evaluate the involvement of a molybdenum enzyme in each dehydroxylation reaction, we cultured the genetically intractable *E. lenta* A2 in the presence of tungstate (*Maini Rekdal et al., 2019*; *Rothery et al., 2008*). As with dopamine dehydroxylation, tungstate inhibited dehydroxylation of (+)-catechin and hydrocaffeic acid by *E. lenta* A2 without inhibiting growth in the rich BHI medium, suggesting these activities are indeed molybdenum dependent (*Figure 4—figure supplements 2* and *3*). Tungstate did not inhibit benzyl ether reduction of (+)-catechin, indicating this step is likely performed by a distinct enzyme (*Figure 4—figure supplement 3*). Finally, we found that the overall distribution of the genes encoding these the putative hydrocaffeic acid and (+)-catechin dehydroxylating enzymes across closely related *Eggerthella* strains correlated with metabolism of each substrate (*Figure 4C*). For example, all *Eggerthella* strains except AN5LG harbored the putative hydrocaffeic acid dehydroxylase and could dehydroxylate this substrate. Similarly, carriage of the putative catechin dehydroxylase correlated with (+)-catechin metabolism, except for in the case of strain AB12#2, which did not encode for the enzyme but still had low metabolism (<10%) (*Figure 4C* and *Figure 4—source data 1*). This suggests that another enzyme might metabolize (+)-catechin in this strain. Overall, these data suggest that *Eggerthella* uses distinct molybdenum-dependent enzymes dehydroxylate hydrocaffeic acid and (+)-catechin.

We next sought to identify the enzyme responsible for DOPAC dehydroxylation in *G. pamelaeae* 3C. We added DOPAC to *G. pamelaeae* 3C cultures at mid-exponential phase and harvested cells after 3 hr of induction when the cultures had reached early stationary phase. In this experiment, *G. pamelaeae* 3C upregulated 100 different genes, including four distinct molybdenum-dependent enzyme-encoding genes (*Supplementary file 2d*). One of these genes (C1877_13905) was among the highest upregulated genes across the dataset (>1700 fold induced). To further explore the association between this gene and DOPAC dehydroxylation, we repeated the RNA-seq experiment, growing *G. pamelaeae* 3C in the presence of DOPAC from the time of inoculation and then harvesting cells in mid-exponential phase as soon we could detect metabolism (12 hr of growth). In this experiment, the same molybdenum-dependent enzyme-encoding gene (C1877_13905) that was highly upregulated in our first experiment (*Supplementary file 2e*) was among the highest upregulated genes. The only two other molybdenum-dependent enzymes induced in this experiment were expressed at an order of magnitude lower levels (<2.5-fold induced). We propose that the molybdenum-dependent enzyme encoded by C1877_13905 is a likely candidate DOPAC dehydroxylase.

This assignment is also supported by comparative genomics. First, carriage of C1877_13905 (named *dodh*) correlated with DOPAC dehydroxylation among members of our gut Actinobacterial library (*Figure 4C*). Consistent with our lysate assays, those organisms harboring this gene also had activity towards hydrocaffeic acid, which could explain the pattern of hydrocaffeic acid metabolism across the gut Actinobacterial library (*Figure 4C*). Finally, the functionally annotated gene most

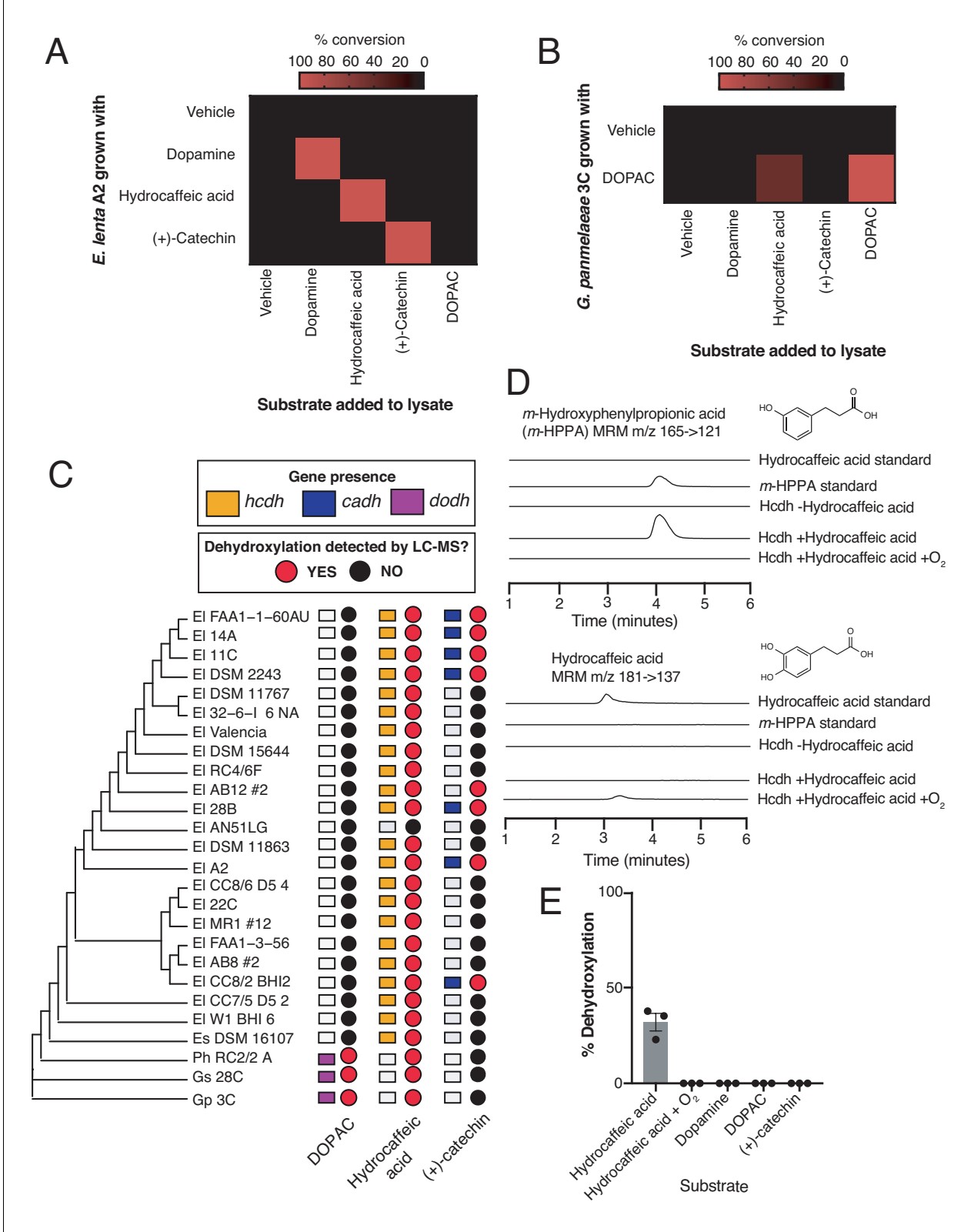

**Figure 4.** Gut Actinobacteria dehydroxylate individual catechols using distinct enzymes. (**A**) Specificity of dehydroxylase regulation and activity in *E. lenta* A2. *E. lenta* A2 was grown anaerobically in BHI medium containing 1% arginine and 10 mM formate. 0.5 mM of catechol was added to induce dehydroxylase expression, followed by anaerobic lysis and enzyme assays. Crude lysates were exposed to different substrates (500 μM) and reactions were allowed to proceed anaerobically for 20 hr. Assays mixtures were analyzed using LC-MS. Heat map represents the mean of three biological

*Figure 4 continued on next page*

*Figure 4 continued*

replicates (lysate reactions). The experiment was performed twice. (B) Specificity of DOPAC dehydroxylase regulation and activity in *G. pamelaeae* A2. *G. pamelaeae* 3C was grown anaerobically in BHI medium containing 10 mM formate. 0.5 mM of catechol was added to induce dehydroxylase expression, followed by anaerobic lysis and enzyme assays. Crude lysates were exposed to different substrates (500 μM) and reactions were allowed to proceed anaerobically for 20 hr. Assays mixtures were analyzed using LC-MS. Heat map represents the mean of three biological replicates (lysate reactions). The experiment was performed twice. (C) Distribution of putative catechol dehydroxylases and their associated metabolic activities across the gut Actinobacterial library used in our study. The tree represents the phylogeny of gut Actinobacterial strains adapted from *Koppel et al. (2018)*. El = *Eggerthella lenta*, Es = *Eggerthella sinesis*, Ph = *Paraggerthella*, Gs = *Gordonibacter* sp., Gp = *Gordonibacter pamelaeae*. Squares represent gene presence/absence of select dehydroxylases across gut Actinobacterial strains (90% coverage, 75% amino acid identity cutoff, e-value = 0). To evaluate catechol metabolism, individual strains were grown in triplicate in the presence of a single catechol substrate for 48 hr at 37°C in BHI medium. Metabolism was assessed using LC-MS/MS. A red dot indicates that the mass of the dehydroxylated product was detected in cultures from this strain, while a black dot indicates lack of metabolism. The experiment was performed once. (D) In vitro activity of Hcdh-containing fractions purified from *E. lenta* A2. EICs for detection of hydrocaffeic acid (MRM m/z 181- > 137) and *m*-hydroxyphenylpropionic acid (*m*-HPPA) (MRM m/z 165- > 121) after 26 hr of anaerobic incubation of enzyme preparation with 500 μM hydrocaffeic acid, 500 μM methyl viologen, and 1 mM sodium dithionite at room temperature. Peak heights show the relative intensity of each mass, and all chromatograms are shown on the same scale. The experiment was performed under anaerobic conditions unless otherwise indicated. The experiment was performed once. (E) Substrate scope of Hcdh-containing fractions purified from *E. lenta* A2. The enzyme preparation used in D) was diluted 1:5 in buffer and was incubated with 500 μM catechol substrate, 500 μM methyl viologen, and 1 mM sodium dithionite at room temperature for 26 hr under anaerobic conditions. The enzyme reactions were analyzed by LC-MS/MS. Bars represent the mean ±the SEM of three independent enzyme reactions. The experiment was performed once. All data can be found in *Figure 4—source data 1*.

The online version of this article includes the following source data and figure supplement(s) for figure 4:

**Source data 1.** Data from lysate assays in E. lenta A2 and G. pamelaeae 3C, screening of Actinobacterial library for metabolism of DOPAC, hydrocaffeic acid, and (+)-catechin, and enzyme assays with hydrocaffeic acid dehydroxylase (*Figure 4*).
**Figure supplement 1.** Inducibility and oxygen sensitivity of catechol dehydroxylation by whole cell suspensions of *E. lenta* A2 and *G. pamelaeae* 3C.
**Figure supplement 2.** Impact of tungstate on *E. lenta* A2 growth in BHI medium.
**Figure supplement 3.** Impact of tungstate on catechol metabolism by *E. lenta* A2 and *G. pamelaeae* 3C.
**Figure supplement 4.** SDS-PAGE of partially purified hydrocaffeic acid dehydroxylase from *E. lenta* A2.
**Figure supplement 5.** Peptide coverage of the hydrocaffeic acid dehydroxylase from *E. lenta* A2 from proteomics.
**Figure supplement 6.** Genomic contexts and predicted subunit composition of *Gordonibacter* and *E. lenta* dehydroxylases.

similar to the candidate DOPAC dehydroxylase is Cldh (45% amino acid ID), a *Gordonibacter* enzyme recently implicated in the dehydroxylation of the lignan dmSECO (*Bess et al., 2020*) (*Supplementary file 2c*). Though this functional assignment awaits biochemical confirmation, we propose that the highest upregulated enzyme across our two independent datasets is the DOPAC dehydroxylase. Interestingly, unlike with the *Eggerthella* dehydroxylases, tungstate did not inhibit dehydroxylation of DOPAC by *G. pamelaeae* (*Figure 4—figure supplement 3*). This may be explained if the dehydroxylating enzyme can use both molybdenum and tungsten for catalysis, as is seen in certain closely related enzymes (*Rosner and Schink, 1995*).

To biochemically validate one of our candidate dehydroxylases, we adapted the native purification protocol used for Dadh to fractionate the hydrocaffeic acid dehydroxylase activity from *E. lenta* A2 cell lysates. This yielded an active fraction that contained five major bands as assessed by SDS-PAGE (*Figure 4—figure supplement 4*) and quantitatively dehydroxylated hydrocaffeic acid into *m*-hydroxyphenylacetic acid under anaerobic conditions (*Figure 4D*). We confirmed that the band with the apparent correct size (*Figure 4—figure supplement 4*) contained the proposed hydrocaffeic acid dehydroxylase (Hcdh, Elenta-A2_02815) (*Figure 4—figure supplement 5* and *Supplementary file 2f*) using proteomics. We also performed an additional set of enzyme assays using this preparation to evaluate the substrate scope of Hcdh. Consistent with our experiments in cell lysates (*Figure 4A*), we observed dehydroxylation only of hydrocaffeic acid and not of dopamine, (+)-catechin, or DOPAC (*Figure 4E*). These data biochemically link the newly identified *hcdh* gene to hydrocaffeic acid dehydroxylation, further supporting the proposal that different enzymes dehydroxylate distinct catechol substrates.

Despite being expected to perform the same type of chemical reaction, the putative catechol dehydroxylases from *E. lenta* and *G. pamelaeae* differ in sequence identity, genomic context, and predicted subunit composition (*Supplementary file 3a* and *Figure 4—figure supplement 6*). The dopamine, catechin and hydrocaffeic acid dehydroxylases from *E. lenta* (*dadh*, *cadh* and *hcdh*, respectively) are likely membrane-bound complexes as they co-localize with genes encoding an

electron shuttling 4Fe-4S ferredoxin and a putative membrane anchor (*Figure 4—figure supplement 6*) (*Rothery et al., 2008*; *Rothery and Weiner, 2015*). These enzymes all carry a Twin-Arginine-Translocation (TAT) signal sequence, suggesting they are exported from the cytoplasm before the signal sequence is cleaved off. We found no peptide coverage of the TAT signal sequence in the proteomics experiment that identified *E. lenta* Hcdh, further confirming that this sequence is cleaved in the mature protein as in other membrane-anchored moco enzymes (*Figure 4—figure supplement 5*) (*Iobbi-Nivol and Leimkühler, 2013*). In contrast, *dodh* and similar enzymes from *G. pamelaeae* do not harbor a TAT signal sequence, are smaller than the *E. lenta* enzymes, and co-localize with a gene predicted to encode a small electron shuttling 4Fe-4S protein, suggesting they are likely soluble protein complexes (*Figure 4—figure supplement 6*). These putative *G. pamelaeae* dehydroxylases are also encoded adjacent to members of the Major Facilitator Superfamily, transporters that may import or export the catechol substrates or dehydroxylated metabolites (*Figure 4—figure supplement 6*). Altogether, these data indicate the existence of distinct subtypes of molybdenum-dependent catechol dehydroxylases.

## Catechol dehydroxylases are variably distributed in metagenomes and correlate with metabolism by complex gut microbiota samples ex vivo

Our finding that catechol dehydroxylases and their associated metabolic activities are variably distributed among closely related gut Actinobacteria made us wonder whether human gut microbial communities would harbor similar genetic and metabolic diversity. To address this, we first searched >1800 publicly available human gut metagenomes (*Nayfach et al., 2015*) for *dadh, hcdh, cadh, dodh,* and the recently identified *cldh* (*Bess et al., 2020*) genes. Although found at generally low abundances, these catechol dehydroxylases were widely but variably distributed across these metagenomes. *Dadh* and *hcdh* were the most prevalent (in >70% and >90% of individuals, respectively), followed by *cadh* (30%), *dodh* (20%), and *cldh* (25%) (*Figure 5A*). Notably, the prevalence of the different genes in metagenomes is consistent with their distribution among individual human gut Actinobacterial isolates (*Figure 4C*).

To assess the presence of catechol dehydroxylation in complex gut microbiotas, we incubated fecal samples from unrelated humans (n = 12) ex vivo with hydrocaffeic acid, (+)-catechin, and stable-isotope deuterium-labeled dopamine and DOPAC and analyzed dehydroxylation by LC-MS/MS. In this experiment, we observed dehydroxylation of dopamine, hydrocaffeic acid, and DOPAC across the majority of subjects, indicating that metabolic activities of low-abundance gut Actinobacteria are indeed prevalent (*Figure 5B–D*). However, catechol metabolism varied between compounds and subjects, with some individuals metabolizing all compounds and some metabolizing none (*Figure 5B–D*). (+)-Catechin was depleted without production of the corresponding dehydroxylated metabolites, consistent with this compound undergoing a wide range of metabolic reactions in complex communities (*Figure 5—source data 1*) (*Takagaki and Nanjo, 2013*; *van't Slot and Humpf, 2009*; *Aura et al., 2008*).

To investigate whether metabolic variability correlated with the presence of specific dehydroxylase enzymes, we further investigated DOPAC metabolism. We separated the 12 samples into a group of 9 metabolizers and three non-metabolizers (in which no biological replicate displayed dehydroxylation activity). qPCR enumeration in these cultures revealed that the abundance of the candidate DOPAC dehydroxylase gene *dodh* discriminated metabolizing and nonmetabolizing subjects (p<0.001, unpaired t-test) (*Figure 4E*), and correlated significantly with dehydroxylation activity within the nine metabolizers (Pearson's correlation, r = 0.73, $R^2$ = 0.53, p<0.05) (*Figure 4F*). Altogether, these data are consistent with our previous finding that *dadh* SNP status correlates with dopamine metabolism in human gut microbiotas ex vivo (*Maini Rekdal et al., 2019*) and suggest that the candidate dehydroxylases may be active in complex gut communities.

## Catechol dehydroxylases are distinct from other molybdenum-dependent enzymes and are widely distributed across sequenced microbes

We next investigated the relationship of catechol dehydroxylases to other characterized molybdenum-dependent enzymes. These enzymes bear no sequence homology to the only other biochemically characterized aromatic dehydroxylase, 4-HCBR; whereas 4-HCBR belongs to the xanthine

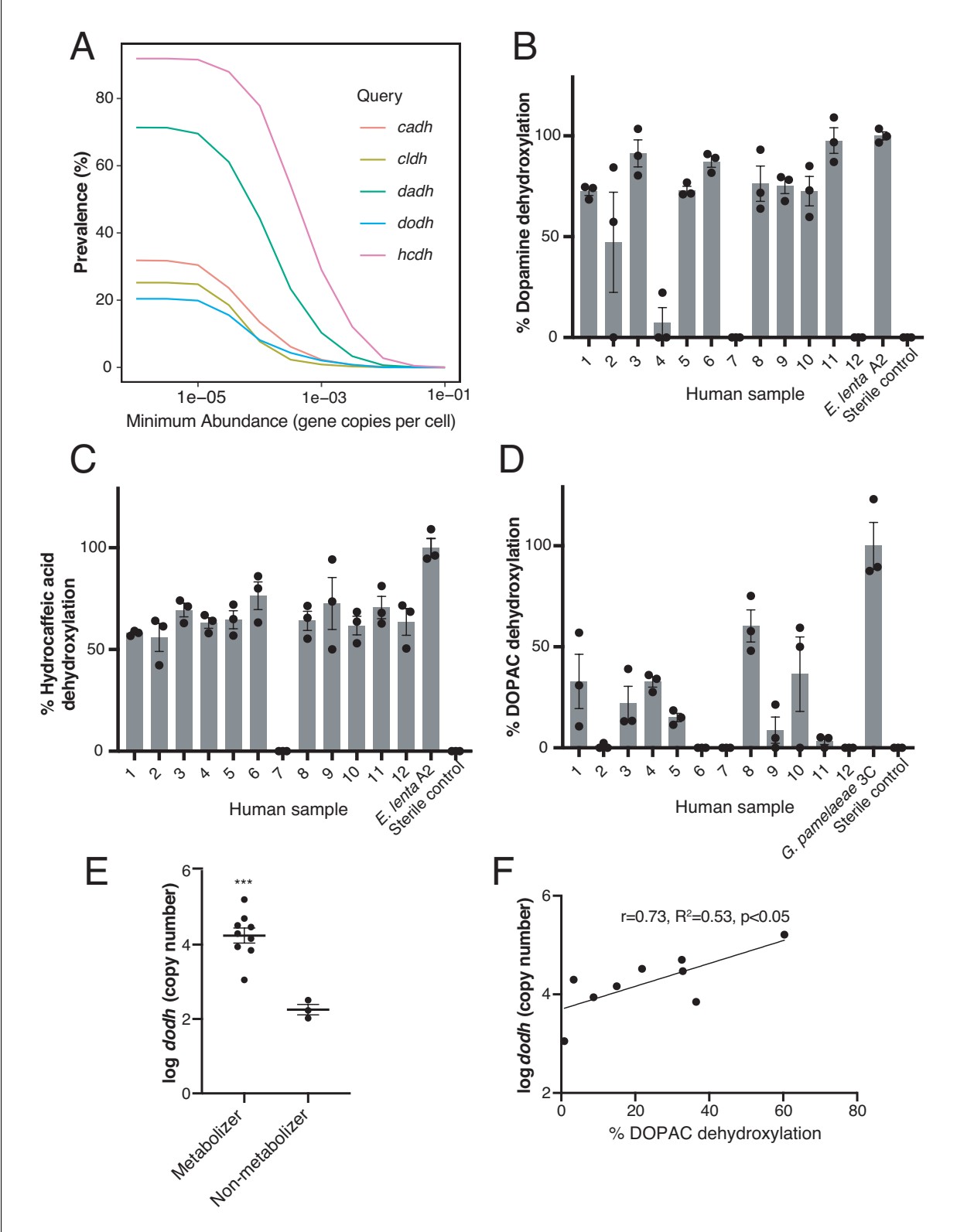

**Figure 5.** Catechol dehydroxylases are variably distributed in human gut metagenomes and correlate with metabolism by complex gut microbiota samples ex vivo. (A) Prevalence estimates (%) as a function of minimum abundance reveals that the candidate dehydroxylases are variably distributed but prevalent in human gut metagenome samples at low relative abundances. B-D) Metabolism of $d_4$-dopamine (B), hydrocaffeic acid (C), and $d_5$-DOPAC (D) by 12 unrelated human gut microbiota samples ex vivo. Samples were cultured anaerobically in BHI medium with 500 µM substrate for 72

*Figure 5 continued on next page*

*Figure 5 continued*

hr and metabolism was analyzed by LC-MS/MS. Bars are mean % dehydroxylation ± SEM (n = 3 independent cultures for each fecal sample). Metabolizing *E. lenta* A2 or *G. pamelaeae* 3C strains were included as positive controls. The experiment was performed once. (E) The abundance of *dodh* correlates with DOPAC dehydroxylation in human gut microbiota samples. Data represent the average *dodh* abundance (as assessed with qPCR) across the three replicates for samples in (D). Results are mean abundance ± SEM (***p<0.001, two-tailed t- test). (F) The abundance of *dodh* among metabolizers correlates with DOPAC dehydroxylation. The data plotted represent the average *dodh* abundance (as assessed with qPCR) and the average % dehydroxylation across the three replicates for metabolizing samples in (D) (all samples except 6,7, and 12). The line represents the best-fit trendline for linear regression. There was a significant linear correlation between *dodh* abundance and % DOPAC dehydroxylation (Pearson's correlation, r = 0.73, $R^2$ = 0.53, p<0.05). All data can be found in *Figure 5—source data 1*.

The online version of this article includes the following source data for figure 5:

**Source data 1.** Data from incubations of human fecal samples with catechols (*Figure 5*).

oxidase family of molybdenum-dependent enzymes, the catechol dehydroxylases belong to the bis-MGD family of molybdenum-dependent enzymes, suggesting independent evolutionary origins (*Hille et al., 2014*; *Unciuleac et al., 2004*; *Tenbrink et al., 2011*). Further phylogenetic analysis revealed that catechol dehydroxylases form a unique clade within the bis-MGD enzyme family, clustering away from pyrogallol hydroxytransferase (Pht), the only other bis-MGD enzyme known to modify the aromatic ring of a substrate (*Messerschmidt et al., 2004*) (*Figure 6* and *Supplementary file 3b*). The catechol dehydroxylases are instead most closely related to acetylene hydratase, an enzyme that adds water to acetylene to provide a carbon source for the marine Proteobacterium *Pelobacter acetylenicus* (*Figure 6*) (*Tenbrink et al., 2011*; *Rosner and Schink, 1995*; *Schoepp-Cothenet et al., 2012*). A sequence similarity network (SSN) analysis using sequences of bis-MGD enzymes revealed distinct clusters of catechol dehydroxylases, further suggesting these enzymes are functionally different from known family members (*Figure 7—figure supplement 1*). The clustering of the dehydroxylases in the SSN did not simply reflect the phylogeny of the organisms because additional sequences from both *Eggerthella* and *Gordonibacter* were found in clusters containing distinct, biochemically characterized enzymes (*Figure 7—figure supplement 2*). In addition, we found that the two catechol dehydroxylase-containing clusters also harbored sequences from organisms other than *Eggerthella* and *Gordonibacter* (*Figure 7—figure supplement 2*). Based on these data, we propose that catechol dehydroxylases are a distinct group of molybdenum-dependent enzymes.

To assess the diversity of putative dehydroxylases, we queried the NCBI nucleotide database and our collection of Actinobacterial genomes for homologs of the *Eggerthella* and *Gordonibacter* enzymes. Phylogenetic analyses of the resulting sequences revealed a large diversity of putative dehydroxylases, including numerous uncharacterized enzymes encoded in individual *Gordonibacter* and *Eggerthella* genomes (*Figure 7*). This highlights that catechol dehydroxylases likely have diversified within these closely related gut Actinobacteria, that individual gut Actinobacteria can likely metabolize a range of different catechols, and that many substrate-enzyme pairs remain to be discovered. Our analysis also revealed that catechol dehydroxylases are not restricted to human-associated Actinobacteria and are instead part of a larger group of bis-MGD enzymes present in diverse bacteria and even Archaea (*Figure 7*). These organisms come from mammal-associated, plant-associated, soil, and aquatic habitats. Notable organisms encoding putative dehydroxylases include soil-dwelling Streptomycetes (*Wu et al., 2017*; *Huang et al., 2012*), the industrially important anaerobe *Clostridium ljungdahlii* (*Köpke et al., 2010*), and a large number of anaerobic bacterial genera known for their ability to degrade aromatic compounds, including *Azoarcus, Thauera, Desulfobacula, Geobacter, Desulfumonile,* and *Desulfitobacterium* (*Figure 7*) (*DeWeerd et al., 1991*; *Wöhlbrand et al., 2013*; *Butler et al., 2007*; *Wagner et al., 2012*; *Cole et al., 1995*; *Fernández et al., 2014*; *Molina-Fuentes et al., 2015*; *Villemur et al., 2006*; *Pacheco-Sánchez et al., 2019a*). The presence of similar enzymes in gut and environmental microbes likely reflects the availability of catechol substrates in many different environments (*Figure 7*).

As the vast majority of dehydroxylase homologs remain uncharacterized, it is difficult to assign the biochemical activities of the major clades and define the characteristic features of these enzymes. However, we are confident that at least some portion of the sequences captured in this analysis are true catechol dehydroxylases. First, we found that representative sequences from across our phylogenetic tree are more closely related to acetylene hydratase and the *Gordonibacter* and

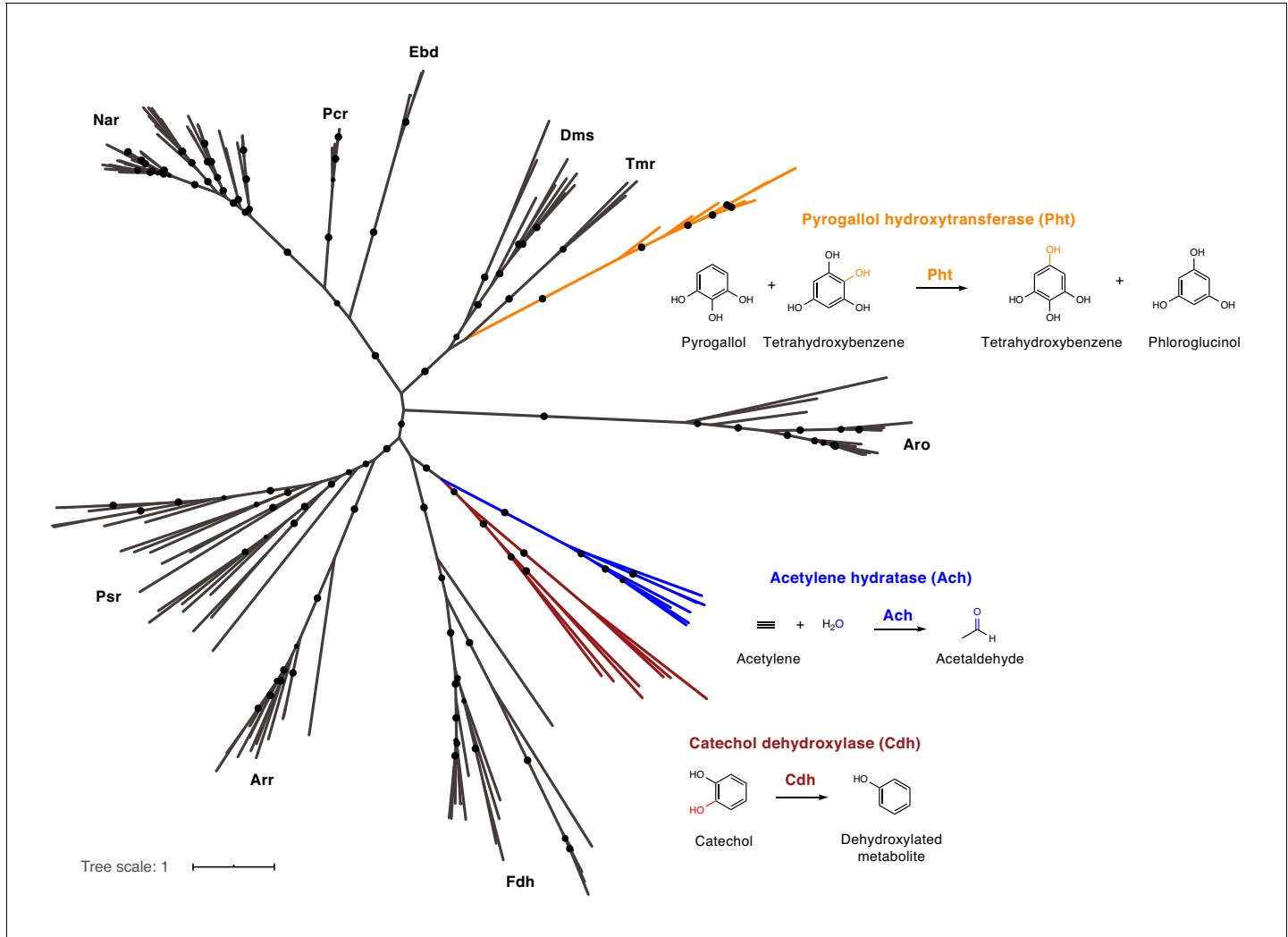

**Figure 6.** Catechol dehydroxylases are distinct from other molybdenum-dependent enzymes. Phylogenetic analysis of newly discovered catechol dehydroxylases reveals a unique evolutionary origin and relationship to acetylene hydratase. Psr = polysulfide reductase; Arr = arsenate reductase; Fdh = formate dehydrogenase; Pht = phloroglucinol transhydroxylase; Dms = DMSO reductase; Tmr = TMAO reductase; Aro = arsenite oxidase; Ebd = ethylbenzene dehydrogenase; Pcr = perchlorate reductase; Nar = nitrate reductase; Cdh = catechol dehydroxylase. The maximum likelihood tree was constructed using sequences from *Schoepp-Cothenet et al. (2012)* as well as additional family members and reproduced the previously reported phylogeny of this enzyme family. Black circles on branches indicate bootstrap values greater than 0.7. Alignment and tree files can be found in *Figure 6* alignment.fasta and *Figure 6* tree file.nex, respectively.

The online version of this article includes the following source data for figure 6:

**Source data 1.** Alignment file for bis-MGD family tree (*Figure 6*).
**Source data 2.** Tree file for bis-MGD family tree (*Figure 6*).

*Eggerthella* dehydroxylases than to any other member of the bis-MGD enzyme family, indicating shared evolutionary origins (*Supplementary file 3c* and *Figure 7—figure supplements 3* and *4*). Moreover, recent genetic studies have implicated several homologs from environmental bacteria in catechol dehydroxylation. For instance, a putative dehydroxylase is present in *Streptomyces* biosynthetic gene clusters that produce the potent anti-tumor compounds yatakemycin and CC-1065 (*Wu et al., 2017*; *Huang et al., 2012*) (*Figure 7*). Gene knock-out and complementation studies revealed this enzyme is essential for CC-1065 production and likely catalyzes reductive dehydroxylation of a late-stage biosynthetic intermediate (*Wu et al., 2017*). Another homolog is present in the 3,5-dihydroxybenzoate (3,5-DHB) degradation operon within the anaerobic soil Proteobacterium

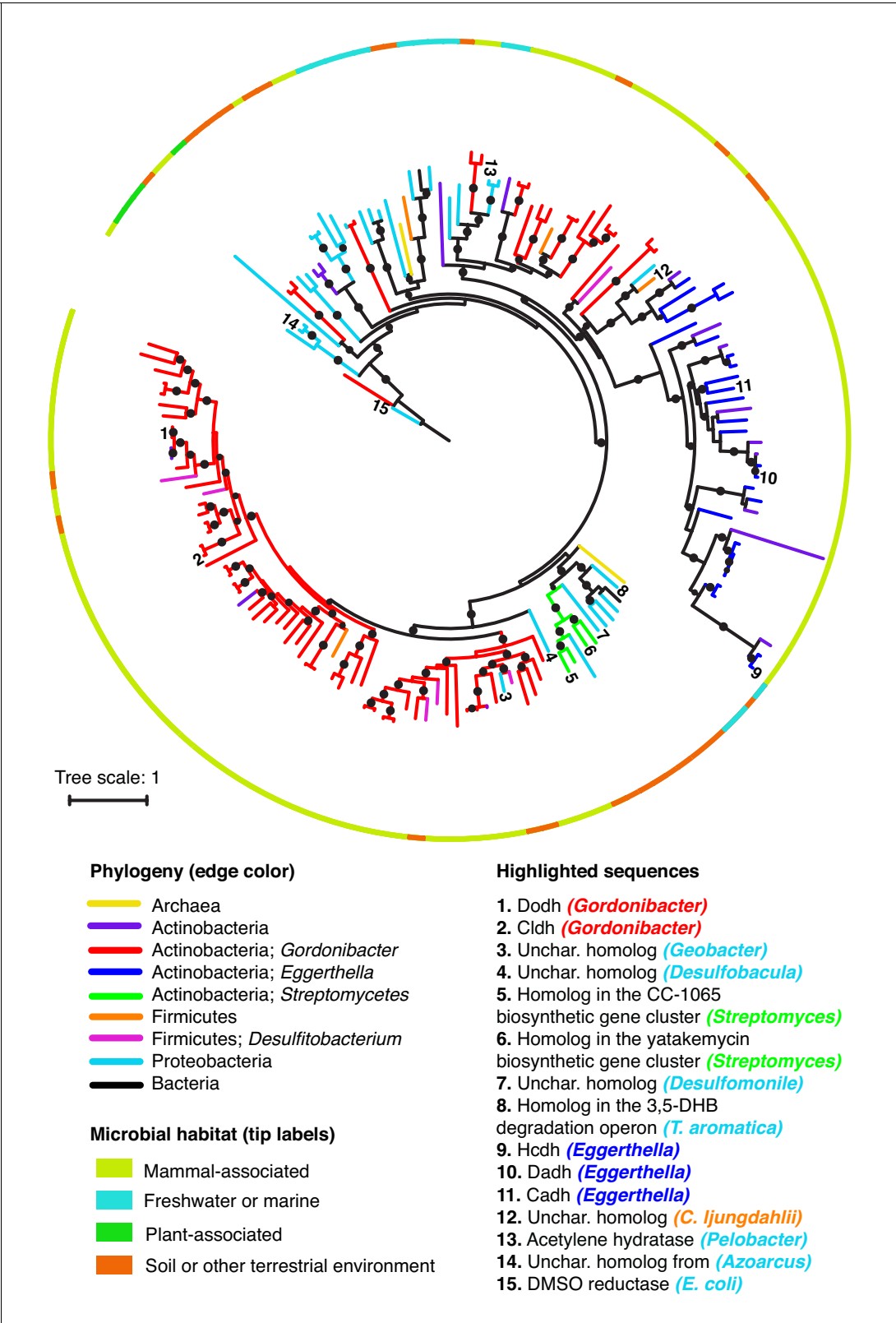

**Phylogeny (edge color)**

- Archaea
- Actinobacteria
- Actinobacteria; *Gordonibacter*
- Actinobacteria; *Eggerthella*
- Actinobacteria; *Streptomycetes*
- Firmicutes
- Firmicutes; *Desulfitobacterium*
- Proteobacteria
- Bacteria

**Microbial habitat (tip labels)**

- Mammal-associated
- Freshwater or marine
- Plant-associated
- Soil or other terrestrial environment

**Highlighted sequences**

1. Dodh *(Gordonibacter)*
2. Cldh *(Gordonibacter)*
3. Unchar. homolog *(Geobacter)*
4. Unchar. homolog *(Desulfobacula)*
5. Homolog in the CC-1065 biosynthetic gene cluster *(Streptomyces)*
6. Homolog in the yatakemycin biosynthetic gene cluster *(Streptomyces)*
7. Unchar. homolog *(Desulfomonile)*
8. Homolog in the 3,5-DHB degradation operon *(T. aromatica)*
9. Hcdh *(Eggerthella)*
10. Dadh *(Eggerthella)*
11. Cadh *(Eggerthella)*
12. Unchar. homolog *(C. ljungdahlii)*
13. Acetylene hydratase *(Pelobacter)*
14. Unchar. homolog from *(Azoarcus)*
15. DMSO reductase *(E. coli)*

**Figure 7.** Catechol dehydroxylases are widely distributed across sequenced microbes. Maximum likelihood phylogenetic tree for catechol dehydroxylase homologs identified by querying 26 gut Actinobacterial genomes (*Bisanz et al., 2018*) and the NCBI nucleotide collection for Dadh and Cldh homologs (see Materials and methods for details). The color of the lines indicates the phylogeny of the organism harboring the homolog. The color of the border indicates the primary habitat from which the organism was originally isolated. Numbers at the end of the branches indicate

*Figure 7 continued on next page*

*Figure 7 continued*

highlighted sequences, which are specified in the legend above. Unchar. stands for uncharacterized. The color of the organism matches the phylogeny of the organism. DMSO reductase from *E. coli* (sequence #15) was used as an outgroup to root the tree. All of the sequences highlighted in the figure in the figure are mentioned in the main text. Black circles on branches indicate bootstrap values greater than 0.7. Alignment and tree files can be found in *Figure 7* alignment.fasta and *Figure 7* tree file.newick, respectively.

The online version of this article includes the following source data and figure supplement(s) for figure 7:

**Source data 1.** Alignment file for tree displaying catechol dehydroxylase diversity and distributionamong sequenced microbes (*Figure 7*).
**Source data 2.** Tree file for tree displaying catechol dehydroxylase diversity and distribution among sequenced microbes (*Figure 7*).
**Figure supplement 1.** Sequence similarity network of the bis-MGD enzyme family reveals that the *Gordonibacter* and *Eggerthella* dehydroxylases belong to distinct, uncharacterized clusters.
**Figure supplement 2.** Sequence similarity network of the bis-MGD enzyme family reveals that *Gordonibacter* and *Eggerthella* bis-MGD enzymes are widely distributed across the enzyme family.
**Figure supplement 3.** Phylogenetic analysis of representative catechol dehydroxylase homologs highlighting sequences included in further analyses.
**Figure supplement 4.** Phylogenetic analysis of putative catechol dehydroxylase sequences from diverse bacterial phyla reveals a relationship with the biochemically characterized molybdenum-dependent enzyme acetylene hydratase.

*Thaeura aromatica* (*Figure 7*). Strains lacking this enzyme exhibit impaired growth on 3,5-DHB as a sole carbon source, suggesting a possible role for this enzyme in metabolizing the one of the two catecholic intermediates involved in this pathway (*Molina-Fuentes et al., 2015*; *Pacheco-Sánchez et al., 2019a*; *Pacheco-Sánchez et al., 2019b*). Based on this analysis, we conclude that the catechol dehydroxylases harbor vast uncharacterized diversity that contributes to both primary and secondary metabolic pathways in habitats beyond the human gut.

## Catechol dehydroxylase reactivity is present across the gut microbiotas of mammals representing distinct diets and phylogenetic origins

Our phylogenetic analysis suggested that catechol dehydroxylase activity is present in a range of microbial habitats, making us curious whether we could detect this metabolism in additional microbial communities. As a first step, we explored catechol dehydroxylation by gut microbiotas of non-human mammals. We assembled a panel of gut microbiota samples from 12 different mammals representing diverse phylogenetic origins and diets (three individuals per mammal) (*Reese et al., 2018*; *Reese et al., 2019*) (*Figure 8* and *Figure 8—figure supplement 1*). We cultured these gut communities anaerobically ex vivo, assessed metabolism using a colorimetric assay, and confirmed potential hits using LC-MS/MS (*Figure 8—figure supplement 1*). We observed catechol dehydroxylation across the gut microbiotas of mammals spanning different diets and phylogenies (*Figure 8*). Hydrocaffeic acid dehydroxylation occurred in >50% of species, while dopamine and (+)-catechin metabolism were observed in 5/12 and 4/12 animals, respectively (*Figure 8*). DOPAC was only metabolized by the rat gut microbiota, which was the only community that had activity towards all compounds tested. While a larger sample size is required to reach clear conclusions about possible links between metabolism of specific catechols and individual mammal gut microbiotas, our results clearly demonstrate that catechol dehydroxylation is found in distantly related mammal gut microbiotas that have large differences in species composition and gene content (*Reese et al., 2018*; *Reese et al., 2019*; *Coelho et al., 2018*). This finding further reinforces the relevance of catechol dehydroxylation to variety of different microbial habitats.

## Discussion

For many decades the human gut microbiota has been known to dehydroxylate catechols, but the molecular basis of this enigmatic transformation has remained largely unknown. In this study, we characterized the specificity and regulation of a gut bacterial enzyme that dehydroxylates dopamine (Dadh). We then used this knowledge to identify candidate enzymes that dehydroxylate additional host-and plant-derived small molecules. Together, the catechol dehydroxylases represent a previously unappreciated group of molybdenum-dependent enzymes that is present in diverse microbial phyla and environments. Our studies of Dadh revealed a high specificity for catecholamines, supporting the hypothesis that the physiological role of this enzyme is to enable neurotransmitter metabolism by *E. lenta*. This idea is also consistent with recent observations of gut bacteria using

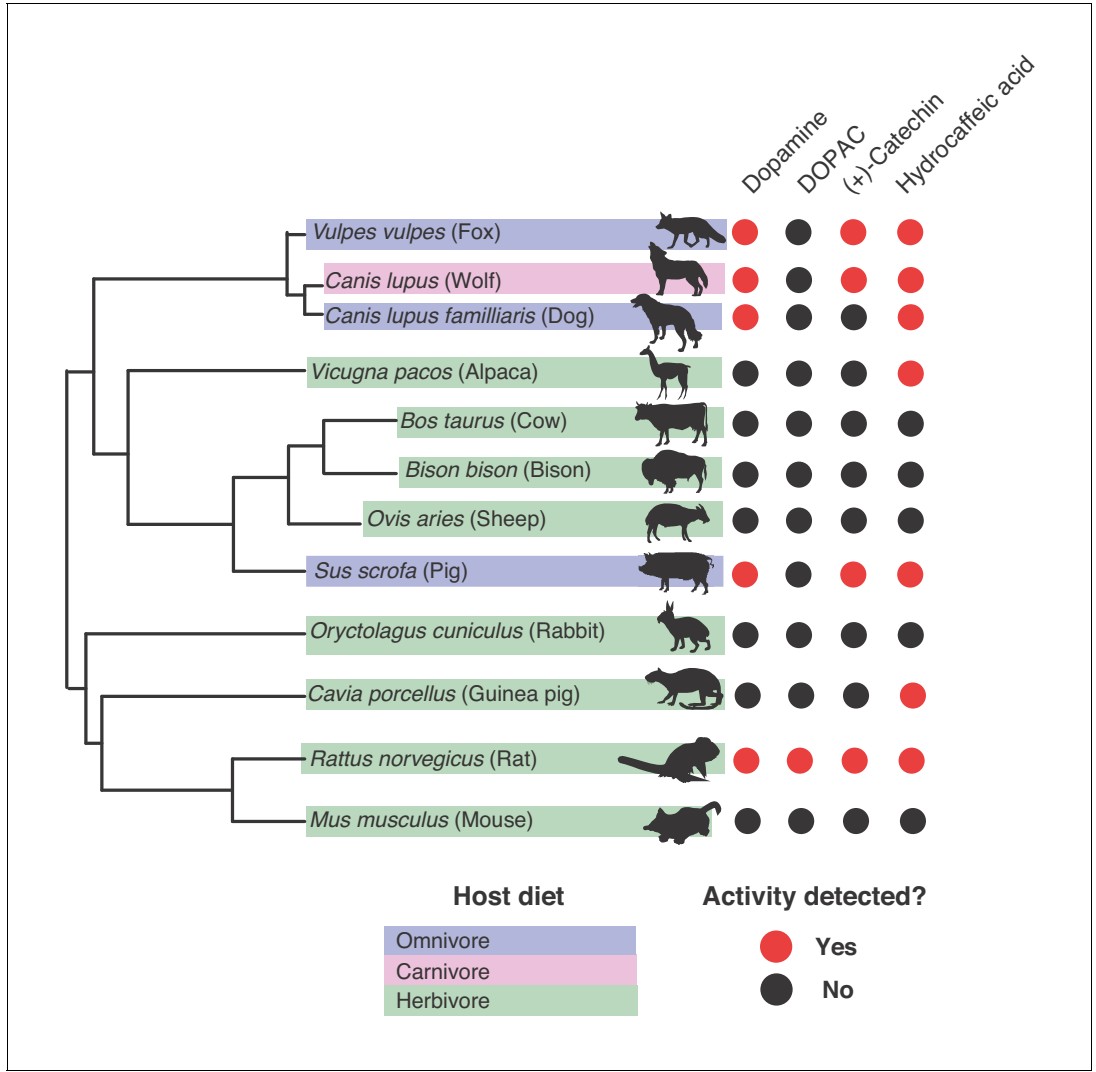

**Figure 8.** Gut microbiotas of mammals representing distinct diets and phylogenetic origins can dehydroxylate catechols. Catechol dehydroxylation of dopamine, DOPAC, (+)-catechin, and hydrocaffeic acid by gut microbiota samples from mammals spanning distinct diets and phylogenetic groups. Gut communities from 12 different mammals and three individuals per animal were cultured anaerobically for 96 hr in basal medium with 0.5 mM catechol at 37°C. The results summarize animals and individuals where the known dehydroxylation pathways examined in human gut Actinobacteria took place, as assessed by LC-MS/MS. Red indicates that metabolism took place in at least one of the individuals, and black indicates lack of metabolism, as assessed by the detection of the dehydroxylated metabolite using LC-MS/MS. The experiment was performed once. The phylogenetic tree was created using the aptg plugin in R and missing branches were added manually based on mammalian phylogeny. The icons were adapted under a Creative Commons license (https://creativecommons.org/licenses/by/3.0/) at phylopic (http://phylopic.org), including Alpaca logo (made my Steven Traver), Bison (Lukasiniho). Cow (Steven Traver), Dog (Tracy A Heath), Fox (Anthony Caravaggi), Guinea pig (Zimices), Mouse (Madeleine Price Ball), Pig (Steven Traver), Rabbit (Steven Traver), Rabbit (Steven Traver), Rat (Rebecca Groom), Sheep (Zimices), and Wolf (Tracy A Heath). All data can be found in *Figure 8—source data 1*.

The online version of this article includes the following source data and figure supplement(s) for figure 8:

**Source data 1.** Screen for catechol metbabolism by mammalian gut microbiota samples (*Figure 8*).

**Figure supplement 1.** Screen for catechol dehydroxylation by gut microbiota samples from diverse mammals.

specific neurotransmitters for growth (*Strandwitz et al., 2019*). To our knowledge, Dadh is the first catecholamine-metabolizing enzyme from a human gut commensal. However, interactions between catecholamines and intestinal pathogens are well-characterized and have long been known as key players in virulence and infection (*Freestone et al., 2007*; *Lyte and Ernst, 1992*). Whereas pathogenic organisms such as *Escherichia coli*, *Yersinia enterocolitica*, and *Salmonella enterica* require the intact catechol group of dopamine and norepinephrine to sequester iron and boost growth

(*Freestone et al., 2007*; *Lyte and Ernst, 1992*; *Rooks et al., 2017*; *Dichtl et al., 2019*), we propose that *E. lenta* uses these molecules as electron acceptors. Thus, Dadh might represent a novel strategy by which gut bacteria take advantage of catecholamines present in the gastrointestinal tract (*Eisenhofer et al., 1997*; *Eisenhofer et al., 1996*). Understanding the interplay between pathogenic and commensal interactions with catecholamines is an intriguing avenue for further research.

In addition to characterizing Dadh, we discovered candidate dehydroxylases that metabolize (+)-catechin, hydrocaffeic acid, and DOPAC. We also partially purified the hydrocaffeic acid dehydroxylase to confirm its involvement in this reaction. Further biochemical studies are important for validating the activities of the remaining enzymes, but our preliminary data support a working model in which catechol dehydroxylation is performed by distinct enzymes that are specialized for individual substrates. We identified large numbers of uncharacterized dehydroxylases encoded within individual *Eggerthella* and *Gordonibacter* genomes (*Figure 7*), hinting at an expansion of this group of enzymes among human gut Actinobacteria. While it remains to be seen whether these uncharacterized enzymes are also specific for distinct substrates, this type of diversification of closely related enzymes indicates a potentially important role for catechol dehydroxylation in the human gut microbiota. Expansion of enzyme families within specific clades of gut microbes is well-characterized in the context of polysaccharide metabolism. For example, individual human gut Bacteroides strains isolates harbor hundreds of polysaccharide utilization loci but upregulate only a subset of genes in response to distinct substrates (*Martens et al., 2011*; *Rogowski et al., 2015*; *Ndeh et al., 2017*; *Larsbrink et al., 2014*). This transcriptional regulation and biochemical specificity enables utilization of various host- or plant-derived carbon sources depending on their availability (*Hehemann et al., 2010*; *Desai et al., 2016*; *Sonnenburg et al., 2010*). The diversity of catechol dehydroxylases might have evolved in a similar manner, providing a biochemical arsenal that enables Actinobacteria to use a range of different electron acceptors whose availability depends on the diet and/or physiology of the host. Identifying the substrates of uncharacterized catechol dehydroxylases could shed light on the adaptation of gut organisms to small molecules produced and ingested by the host.

In addition to uncovering the diversity of catechol dehydroxylases, our study illustrates that the chemical strategies used to enable microbial survival and interactions in the human gut may be relevant to a broad range of species and habitats. While mammalian gut microbiomes have previously been compared in terms of gene content and species composition (*Reese et al., 2019*; *Coelho et al., 2018*; *Youngblut et al., 2019*), our study provides functional evidence for conservation of specific gut microbial metabolic pathways across distinct hosts. While this hints at potentially important roles for catechol dehydroxylation across mammalian gut communities, the distribution of putative dehydroxylases among environmental microbes suggests this chemistry is present in many additional microbial habitats. This reinforces findings from studies of additional gut microbial enzymes. For example, gut microbial carbohydrate-degrading enzymes and glycyl radical enzymes, which play important roles in degrading diet-derived polysaccharides, amino acids, and osmolytes in the human gut, are also found in environmental isolates (*Ndeh et al., 2017*; *Levin et al., 2017*; *Craciun and Balskus, 2012*; *Peck et al., 2019*). Enzyme discovery in the human gut microbiota not only has implications for improving human health and disease, but also for discovering novel catalytic functions and metabolic pathways broadly relevant to microbial life. Our study now sets the stage for further investigations of the chemical mechanisms and biological consequences of catechol dehydroxylation in the human body and beyond.

More broadly, our study underscores how enzyme discovery can help to dissect the metabolic diversity of gut microbial strains and communities. Although previous studies had linked certain dehydroxylation reactions to individual gut Actinobacteria (*Maini Rekdal et al., 2019*; *Takagaki and Nanjo, 2015*; *Selma et al., 2014*; *Bess et al., 2020*), we have found that specific catechol dehydroxylases are variably distributed among closely related strains and human gut metagenomes. These findings reinforce the idea that gut microbial phylogeny is often not predictive of functional capabilities (*Koppel et al., 2018*; *Maini Rekdal et al., 2019*; *Levin et al., 2017*; *Craciun and Balskus, 2012*; *Peck et al., 2019*; *Martínez-del Campo et al., 2015*). Additionally, we noticed that the prevalence of the different dehydroxylation reactions among human and animal gut microbiota samples reflected their distribution among individual Actinobacterial strains, with hydrocaffeic acid metabolism being the most prevalent across all strains and species, and DOPAC dehydroxylation being the least prevalent. This may suggest that the strain-level variability in dehydroxylases is important for metabolism both within humans and other mammalian species. While the evolutionary forces

shaping the distribution of specific dehydroxylases within gut bacterial strains and complex gut communities remain unknown, our data provide a starting point for further understanding the effects of catechol dehydroxylation on both gut microbiota and host.

Finally, our findings provide a framework for linking metabolic transformations performed by complex gut microbial communities to individual strains, genes, and enzymes. Our broad exploration of a class of metabolic transformations contrasts with the more common focus on metabolism of individual drugs or dietary compounds (*Maini Rekdal et al., 2019*; *Koppel et al., 2018*; *Levin et al., 2017*; *Craciun and Balskus, 2012*; *Peck et al., 2019*; *Martínez-del Campo et al., 2015*; *Williams et al., 2014*; *Yan et al., 2018*). This functional group-focused approach may greatly increase the efficiency with which we can link metabolic activities to microbial genes and enzymes. We envision that related experimental workflows could find broad utility in the discovery of gut microbial enzymes catalyzing other widespread, biologically significant reactions, including reductive metabolism of additional functional groups that are prevalent in diverse molecules encountered by the gut microbiota (*Koppel et al., 2017*).

## Materials and methods

This section includes the key resources table and materials, methods, and data for all experiments except for synthesis and characterization of dopamine. Materials, methods, and characterization data for synthesis of dopamine analogs can be found Appendix 1.

**Key resources table**

| Reagent type (species) or resource | Designation | Source or reference | Identifiers | Additional information |
|---|---|---|---|---|
| Strain, strain background (*Eggerthella lenta*) | *Eggerthella lenta* strains | REF 37 | El | See *Supplementary file 1d* |
| Strain, strain background (*Eggerthella sinensis*) | *Eggerthella sinensis* DSM16107 | REF 37 | Es | See *Supplementary file 1d* |
| Strain, strain background (*Gordonibacter*) | *Gordonibactr* strains | REF 37 | Gp, Gs | See *Supplementary file 1d* |
| Strain, strain background (*Paraeggerthella*) | *Paraeggerthella hongkongensis* | REF 37 | Ph | See *Supplementary file 1d* |
| Strain, strain background (*Eschericia coli*) | *Eschericia coli* MG1655 | Palmer lab, Newcastle University | | See *Supplementary file 1d* |
| Strain, strain background (*Bacteroides fragilis* ATCC 25285) | *Bacteroides fragilis* ATCC 25285 | ATCC | | See *Supplementary file 1d* |
| Strain, strain background (*Clostridium sporogenes* ATCC 15579) | *Clostridium sporogenes* ATCC 15579 | ATCC | | See *Supplementary file 1d* |
| Strain, strain background (*Enterococcus faecalis* OG1RF) | *Enterococcus faecalis* OG1RF | ATCC | | See *Supplementary file 1d* |
| Strain, strain background (*Edwarsiella tarda* ATCC 23685) | *Edwarsiella tarda* ATCC 23685 | ATCC | | See *Supplementary file 1d* |
| Sequence-based reagent | qPCR primers *dadh* | REF 9 | PCR primers | GAGATCTGGTCCACCGTCAT and AGTGGAAGTACACCGGGATG |
| Sequence-based reagent | qPCR primers *E. lenta* | REF 10 | PCR primers | CAGCAGGGAAGAAATTCGAC and TTGAGCCCTCGGATTAGAGA |
| Sequence-based reagent | qPCR primers *dodh* | This work | PCR primers | GFP version of pLKO.1-Puro |

*Continued on next page*

*Continued*

| Reagent type (species) or resource | Designation | Source or reference | Identifiers | Additional information |
|---|---|---|---|---|
| Sequence-based reagent | Primers for full length *dadh* | REF 9 | PCR primers | ATGGGTAACCTGACCATG and TTACTCCCTCCCTTCGTA |
| Sequence-based reagent | Sequencing primers SNP506 in *dadh* | REF 9 | PCR primers | GGGGTGTCCATGTTGCCGGT and ACCGGCTACGGCAACGGC |
| Commercial assay or kit | DNeasy UltraClean Microbial Kit | Qiagen, catalog | Cat # 12224–50 | Extraction of gDNA from bacterial cultures |
| Chemical compound, drug | Dopamine hydrochloride | Sigma-Aldrich | Cat# PHR1090-1G | |
| Chemical compound, drug | (+)-catechin hydrate | Millipore Sigma | Cat# C1251-5G | |
| Chemical compound, drug | 3,4-dihydroxyphenylacetic acid (DOPAC) | Millipore Sigma | Cat# 850217–1G | |
| Chemical compound, drug | 3,4-dihydroxyphenylpropionic acid (Hydrocaffeic acid) | Millipore Sigma | Cat# 102601–10G | |
| Other | LC-MS/MS | Agilent | Agilent:6410 Triple Quad LC/MS | |
| Other | Anaerobic chambers | Coy Laboratory products | | |
| Chemical compound, drug | BBL Brain Heart Infusion (BHI) media | Beckton Dickinson | Cat# L007440 | |
| Chemical compound, drug | L-arginine | Sigma-Aldrich | Cat# A5006-100G | |
| Chemical compound, drug | benzyl viologen | Sigma-Aldrich | Cat# 271845–250 mg | |
| Chemical compound, drug | methyl viologen | Sigma-Aldrich | Cat# 856177–1 g | |
| Chemical compound, drug | sodium dithionite | Sigma-Aldrich | Cat# 157953–5G | |
| Chemical compound, drug | diquat | Sigma-Aldrich | Cat# 45422–250 mg | |
| Chemical compound, drug | 3,4-dihydroxyphenylacetic acid (ring-d3, 2,2-d2, 98%) | Cambridge Isotope Laboratories | Cat# DLM-2499–0.01 | |
| Chemical compound, drug | dopamine HCl (1,1,2,2-d4, 97–98%) | Cambridge Isotope Laboratories | Cat# DLM-2498–0.1 | |

## General materials and methods

The following chemicals were used in this study: tetracycline (Sigma Aldrich, catalog# 87128–25G), *p*-tyramine (Sigma Aldrich, catalog# T2879-1G), DL-3,4-Dihydroxymandelic acid (Carbo Synth, catalog# FD22118), protocatechuic Acid (Millipore Sigma, catalog# 37580–25 G-F), DL norepinephrine (Millipore Sigma, catalog# A7256-1G), L-norepinephrine (Matrix Scientific, catalog# 037592–500 MG) L-epinephrine (Alfa Aesar, catalog# L04911.06), DL-epinephrine (Sigma Aldrich, catalog# E4642-5G), 3,4-dihydroxyphenylacetic acid (Millipore Sigma, catalog# 850217–1G), 3,4-dihydroxyhydrocinnamic acid (hydrocaffeic acid) (Millipore Sigma, catalog# 102601–10G), caffeic acid (Millipore Sigma, catalog# C0625-2G), (+)-catechin hydrate (Millimore Sigma, catalog# C1251-5G), (+/–)-catechin hydrate (Millipore Sigma, catalog# C1788-500MG), (–)-Epicatechin (Millipore Sigma, catalog# E1753-1G), L-(-)-a-Methyldopa (Chemcruz, catalog# sc-203092), 2,3-dihydroxybenzoic acid (Millipore Sigma, catalog# 126209–5G), *R*-(–)-apomorphine hydrochloride hemihydrate (Sigma Aldrich, catalog# A4393-100MG), hydroxytyrosol (Ava Chem Scientific, catalog# 2528), enterobactin (generous gift from Prof. Elizabeth Nolan, MIT), fenoldopam mesylate (Sigma Aldrich, catalog# SML0198-10MG), 5-hydroxydopamine (Sigma Aldrich, catalog# 151564–100G), 6-hydroxydopamine (Sigma Aldrich, catalog #H4381-100MG), 3-methoxytyramine (Sigma Aldrich, catalog# M4251-100MG), 3,4-dihydroxybenzylamine (Sigma Aldrich, catalog# 858781–250 MG), *N*-methyldopamine (Santa Cruz Biotechnology, catalog# sc-358430A), 4-(2-aminoethyl)benzene-1,3-diol (Enamine, catalog # EN300-

65185), *m*-tyramine (Chemcruz, catalog# sc-255257), 3-hydroxyphenylacetic acid (Sigma Aldrich, catalog# H49901-5G), 3-hydroxyphenylpropionic acid (Toronto Research Chemicals, catalog# H940090), L-dopa (Oakwood Chemical, catalog# 358380–25 g), dopamine (Sigma-Aldrich, catalog# PHR1090-1G, or Millipore Sigma, catalog# H8502-25G), *m*-tyramine (Santa Cruz Biotechnology, catalog# sc-255257), carbidopa (Sigma-Aldrich, catalog# PHR1655-1G), L-arginine (Sigma-Aldrich, catalog# A5006-100G), sodium molybdate (Sigma-Aldrich, catalog # 243655–100G), sodium tungstate (72069–25G), SIGMAFAST protease inhibitor tablets (Sigma-Aldrich, catalog#: S8830), benzyl viologen (Sigma-Aldrich, catalog# 271845–250 mg), methyl viologen (Sigma-Aldrich, catalog# 856177–1 g), diquat (Sigma-Aldrich, catalog# 45422–250 mg), sodium dithionite (Sigma-Aldrich, catalog# 157953–5G), 3,4-dihydroxyphenylacetic acid (ring-d3, 2,2-d2, 98%) (Cambridge Isotope Laboratories, catalog #DLM-2499–0.01), dopamine HCl (1,1,2,2-d4, 97–98%) (Cambridge Isotope Laboratories, catalog #DLM-2498–0.1). LC-MS grade acetonitrile and methanol for LC-MS analyses were purchased from Honeywell Burdick and Jackson or Sigma-Aldrich. Brain Heart Infusion (BHI) broth was purchased from Beckton Dickinson (catalog# 211060) or from VWR (catalog# 95021–488).

All bacterial culturing work was performed in an anaerobic chamber (Coy Laboratory Products) under an atmosphere of 10% hydrogen, 10% carbon dioxide, and nitrogen as the balance, unless otherwise noted. Hungate tubes were used for anaerobic culturing unless otherwise noted (Chemglass, catalog# CLS-4209–01). All lysate work and biochemical experiments were performed in an anaerobic chamber (Coy Laboratory Products) situated in a cold room at 4°C under an atmosphere of 10% hydrogen and nitrogen as the balance. Gut Actinobacterial strains were grown on BHI containing 1% arginine (w/v) to obtain isolated colonies for culturing.

All genomic DNA (gDNA) was extracted from bacterial cultures using the DNeasy UltraClean Microbial Kit (Qiagen, catalog # 12224–50) according to the manufacturer's protocol.

## LC-MS methods

Method A: Samples were analyzed using an Agilent technologies 6410 Triple Quad LC/MS and a Dikma Technologies Inspire Phenyl column (4.6 × 150 mm, 5 µm; catalog #81801). The flow rate was 0.5 mL min$^{-1}$ using 0.1% formic acid in water as mobile phase A and 0.1% formic acid in acetonitrile as mobile phase B. The column temperature was maintained at room temperature. The following gradient was applied: 0–2 min: 0% B isocratic, 2–9 min: 0–10% B, 9–11 min: 10–95% B, 11–15 min: 95% B isocratic, 15–18 min: 95–0% B, 18–21 min: 0% B isocratic. For mass spectrometry, the source temperature was 300°C, and the masses of dopamine (precursor ion *m/z* = 154.3, daughter ion *m/z* = 137.3), and tyramine (precursor ion *m/z* = 138.3, daughter ion *m/z* = 121.3) were monitored at a collision energy of 15 mV and fragmentor setting of 135 in positive MRM mode.

Method B: Samples were analyzed using an Agilent technologies 6410 Triple Quad LC/MS and a Thermo Scientific Acclaim Polar Advantage II column (3 µM, 120A, 2.1*150 mm, product #: 063187). The flow rate was 0.2 mL min$^{-1}$ using 0.1% formic acid in water as mobile phase A and methanol as mobile phase B. The following gradient was applied: 0–4 min: 50% B isocratic, 4–7 min: 50–99%, 7–9 min: 99–50%, 9–13 min: 50% B isocratic. For mass spectrometry, the source temperature was 300°C, and the masses of trihydroxydopamine (precursor ion *m/z* = 170.3, daughter ion *m/z* = 153.3), dopamine (precursor ion *m/z* = 154.3, daughter ion *m/z* = 137.3), phenylethylamine (precursor ion *m/z* = 122.3, daughter ion *m/z* = 105.2), and tyramine (precursor ion *m/z* = 138.3, daughter ion *m/z* = 121.3) were monitored at a collision energy of 15 mV and fragmentor setting of 135 in positive MRM mode.

Method C: Samples were analyzed using an Agilent technologies 6530 Accurate-Mass Q-TOF LC/MS and a Dikma Technologies Inspire Phenyl column (4.6 × 150 mm, 5 µm; catalog #81801). The flow rate was 0.4 mL min$^{-1}$ using 0.1% formic acid in water as mobile phase A and 0.1% formic acid in acetonitrile as mobile phase B. The column temperature was maintained at room temperature. The following gradient was applied: 0–2 min: 5% B isocratic, 2–25 min: 0–95% B, 25–30 min: 95% B isocratic, 30–40 min: 95–5% B. For the MS detection, the ESI mass spectra data were recorded in positive mode for a mass range of *m/z* 50 to 3000. A mass window of ±0.005 Da was used to extract the ion of [M+H].

Method D: Samples were analyzed using an Agilent technologies 6530 Accurate-Mass Q-TOF LC/MS and a Dikma Technologies Inspire Phenyl column (4.6 × 150 mm, 5 µm; catalog #81801). The flow rate was 0.4 mL min$^{-1}$ using 0.1% formic acid in water as mobile phase A and 0.1% formic acid in acetonitrile as mobile phase B. The column temperature was maintained at room temperature.

The following gradient was applied: 0–2 min: 5% B isocratic, 2–25 min: 0–95% B, 25–30 min: 95% B isocratic, 30–40 min: 95–5% B. For the MS detection, the ESI mass spectra data were recorded in negative mode for a mass range of $m/z$ 50 to 3000. A mass window of ±0.005 Da was used to extract the ion of [M+H].

Method E: Samples were analyzed using an Agilent technologies 6410 Triple Quad LC/MS and a Thermo Scientific Acclaim Polar Advantage II column (3 µM, 120A, 2.1*150 mm, product #: 063187). The flow rate was 0.2 mL min$^{-1}$ using 0.1% formic acid in water as mobile phase A and methanol as mobile phase B. The following gradient was applied: 0–4 min: 50% B isocratic, 4–7 min: 50–99%, 7–9 min: 99–50%, 9–13 min: 50% B isocratic. For mass spectrometry, the source temperature was 300°C, and the masses of catechin (precursor ion $m/z$ = 289.2, daughter ion $m/z$ = 109.1), benzyl ether reduced catechin (precursor ion $m/z$ = 291.2, daughter ion $m/z$ = 123.1), benzyl ether reduced, dehydroxylated catechin (precursor ion $m/z$ = 275.2, daughter ion $m/z$ = 107.1) were monitored at a collision energy of 15 mV and fragmentor setting of 135 in negative MRM mode.

Method F: Samples were analyzed using an Agilent technologies 6410 Triple Quad LC/MS and a Thermo Scientific Acclaim Polar Advantage II column (3 µM, 120A, 2.1*150 mm, product #: 063187). The flow rate was 0.2 mL min$^{-1}$ using 0.1% formic acid in water as mobile phase A and methanol as mobile phase B. The following gradient was applied: 0–4 min: 50% B isocratic, 4–7 min: 50–99%, 7–9 min: 99–50%, 9–13 min: 50% B isocratic. For mass spectrometry, the source temperature was 300°C, and the masses of hydrocaffeic acid (precursor ion $m/z$ = 181.2, daughter ion $m/z$ = 137.2), hydroxy-phenylpropionic acid (precursor ion $m/z$ = 165.1, daughter ion $m/z$ = 121.2), DOPAC (precursor ion $m/z$ = 167.2, daughter ion $m/z$ = 123.2), and hydroxyphenylacetic acid (precursor ion $m/z$ = 151.3, daughter ion $m/z$ = 107.3) were monitored at a collision energy of 15 mV and fragmentor setting of 135 in negative MRM mode.

Method G: Samples were analyzed using an Agilent technologies 6410 Triple Quad LC/MS and a Thermo Scientific Acclaim polar advantage II column (3 µM, 120A, 2.1*150 mm, product #: 063187). The flow rate was 0.2 mL min$^{-1}$ using 0.1% formic acid in water as mobile phase A and methanol as mobile phase B. The following gradient was applied: 0–4 min: 50% B isocratic, 4–7 min: 50–99%, 7–9 min: 99–50%, 9–13 min: 50% B isocratic. For mass spectrometry, the source temperature was 275°C, and the masses of norepinephrine (precursor ion $m/z$ = 170.1, daughter ion $m/z$ = 152.1) and octop-amine (precursor ion $m/z$ = 154.2, daughter ion $m/z$ = 136.1) were monitored at a collision energy of 5 mV and fragmentor setting of 135 in positive MRM mode.

Method H: Samples were analyzed using an Agilent technologies 6410 Triple Quad LC/MS and a Thermo Scientific Acclaim Polar Advantage II column (3 µM, 120A, 2.1*150 mm, product #: 063187). The flow rate was 0.2 mL min$^{-1}$ using 0.1% formic acid in water as mobile phase A and methanol as mobile phase B. The following gradient was applied: 0–4 min: 50% B isocratic, 4–7 min: 50–99%, 7–9 min: 99–50%, 9–13 min: 50% B isocratic. For mass spectrometry, the source temperature was 275°C, and the masses of caffeic acid (precursor ion $m/z$ = 179.2, daughter ion $m/z$ = 135.2) and coumaric acid (precursor ion $m/z$ = 163.3, daughter ion $m/z$ = 119.2) were monitored at a collision energy of 5 mV and fragmentor setting of 135 in negative MRM mode.

Method I: Samples were analyzed using an Agilent technologies 6410 Triple Quad LC/MS and a Thermo Scientific Acclaim Polar Advantage II column (3 µM, 120A, 2.1*150 mm, product #: 063187). The flow rate was 0.2 mL min$^{-1}$ using 0.1% formic acid in water as mobile phase A and methanol as mobile phase B. The following gradient was applied: 0–4 min: 50% B isocratic, 4–7 min: 50–99%, 7–9 min: 99–50%, 9–13 min: 50% B isocratic. For mass spectrometry, the source temperature was 300°C, and the masses of dihydroxybenzoic acid (precursor ion $m/z$ = 153.1, daughter ion $m/z$ = 137.1) and hydroxybenzoic acid (precursor ion $m/z$ = 137.1, daughter ion $m/z$ = 93.2) were monitored at a colli-sion energy of 15 mV and fragmentor setting of 135 in negative MRM mode.

Method J: Samples were analyzed using an Agilent technologies 6410 Triple Quad LC/MS and a Thermo Scientific Acclaim Polar Advantage II column (3 µM, 120A, 2.1*150 mm, product #: 063187). The flow rate was 0.2 mL min$^{-1}$ using 0.1% formic acid in water as mobile phase A and methanol as mobile phase B. The following gradient was applied: 0–4 min: 50% B isocratic, 4–7 min: 50–99%, 7–9 min: 99–50%, 9–13 min: 50% B isocratic. For mass spectrometry, the source temperature was 300°C, and the masses of norepinephrine (precursor ion $m/z$ = 184.1, daughter ion $m/z$ = 166.1) and dehy-droxynorepinephrine (precursor ion $m/z$ = 168.1, daughter ion $m/z$ = 150.1) were monitored at a col-lision energy of 15 mV and fragmentor setting of 135 in positive MRM mode.

Method K: Samples were analyzed using an Agilent technologies 6410 Triple Quad LC/MS and a Thermo Scientific Acclaim Polar Advantage II column (3 μM, 120A, 2.1*150 mm, product #: 063187). The flow rate was 0.2 mL min$^{-1}$ using 0.1% formic acid in water as mobile phase A and methanol as mobile phase B. The following gradient was applied: 0–4 min: 50% B isocratic, 4–7 min: 50–99%, 7–9 min: 99–50%, 9–13 min: 50% B isocratic. For mass spectrometry, the source temperature was 300°C, and the masses of dihydroxybenzylamine (precursor ion $m/z$ = 140.3, daughter ion $m/z$ = 123.2) and hydroxybenzylamine (precursor ion $m/z$ = 124.3, daughter ion $m/z$ = 107.2) were monitored at a collision energy of 15 mV and fragmentor setting of 135 in positive MRM mode.

Method L: Samples were analyzed using an Agilent technologies 6410 Triple Quad LC/MS and a Thermo Scientific Acclaim Polar Advantage II column (3 μM, 120A, 2.1*150 mm, product #: 063187). The flow rate was 0.2 mL min$^{-1}$ using 0.1% formic acid in water as mobile phase A and methanol as mobile phase B. The following gradient was applied: 0–4 min: 50% B isocratic, 4–7 min: 50–99%, 7–9 min: 99–50%, 9–13 min: 50% B isocratic. For mass spectrometry, the source temperature was 300°C, and the masses of 3-aminotyramine (precursor ion $m/z$ = 153.3, daughter ion $m/z$ = 136.2) and 3-aminophenylethylamine (precursor ion $m/z$ = 137.3, daughter ion $m/z$ = 120.2) were monitored at a collision energy of 15 mV and fragmentor setting of 135 in positive MRM mode.

Method M: Samples were analyzed using an Agilent technologies 6410 Triple Quad LC/MS and a Thermo Scientific Acclaim Polar Advantage II column (3 μM, 120A, 2.1*150 mm, product #: 063187). The flow rate was 0.2 mL min$^{-1}$ using 0.1% formic acid in water as mobile phase A and methanol as mobile phase B. The following gradient was applied: 0–4 min: 50% B isocratic, 4–7 min: 50–99%, 7–9 min: 99–50%, 9–13 min: 50% B isocratic. For mass spectrometry, the source temperature was 300°C, and the masses 3-methoxytyramine (precursor ion $m/z$ = 151.1, daughter ion $m/z$ = 91.1) and 3-methoxyphenylethylamine (precursor ion $m/z$ = 135.1, daughter ion $m/z$ = 75.1) were monitored at a collision energy of 15 mV and fragmentor setting of 135 in positive MRM mode.

Method N: Samples were analyzed using an Agilent technologies 6410 Triple Quad LC/MS and a Thermo Scientific Acclaim Polar Advantage II column (3 μM, 120A, 2.1*150 mm, product #: 063187). The flow rate was 0.2 mL min$^{-1}$ using 0.1% formic acid in water as mobile phase A and methanol as mobile phase B. The following gradient was applied: 0–4 min: 50% B isocratic, 4–7 min: 50–99%, 7–9 min: 99–50%, 9–13 min: 50% B isocratic. For mass spectrometry, the source temperature was 300°C, and the masses 3-hydroxytyrosol (precursor ion $m/z$ = 153.2, daughter ion $m/z$ = 123.1) and tyrosol (precursor ion $m/z$ = 137.2, daughter ion $m/z$ = 107.1) were monitored at a collision energy of 15 mV and fragmentor setting of 135 in negative MRM mode.

Method O: Samples were analyzed using an Agilent technologies 6410 Triple Quad LC/MS and a Thermo Scientific Acclaim Polar Advantage II column (3 μM, 120A, 2.1*150 mm, product #: 063187). The flow rate was 0.15 mL min$^{-1}$ using 0.1% formic acid in water as mobile phase A and methanol as mobile phase B. The following gradient was applied: 0–4 min: 50% B isocratic, 4–7 min: 50–99%, 7–9 min: 99–50%, 9–13 min: 50% B isocratic. For mass spectrometry, the source temperature was 300°C, and the masses of d5-DOPAC (precursor ion m/z = 172.2, daughter ion m/z = 128.2) and d5-hydroxyphenylacetic acid (precursor ion $m/z$ = 156.3, daughter ion $m/z$ = 113.3) were monitored at a collision energy of 15 mV and fragmentor setting of 135 in negative MRM mode.

Method P: Samples were analyzed using an Agilent technologies 6410 Triple Quad LC/MS and a Thermo Scientific Acclaim Polar Advantage II column (3 μM, 120A, 2.1*150 mm, product #: 063187). The flow rate was 0.2 mL min$^{-1}$ using 0.1% formic acid in water as mobile phase A and methanol as mobile phase B. The following gradient was applied: 0–4 min: 50% B isocratic, 4–7 min: 50–99%, 7–9 min: 99–50%, 9–13 min: 50% B isocratic. For mass spectrometry, the source temperature was 300°C, and the masses of d4-dopamine (precursor ion $m/z$ = 158.3, daughter ion $m/z$ = 141.3), and d4-tyramine (precursor ion $m/z$ = 142.3, daughter ion $m/z$ = 125.3) were monitored at a collision energy of 15 mV and fragmentor setting of 135 in positive MRM mode.

## Colorimetric assay for catechol detection

The colorimetric assay for dopamine dehydroxylation was based on the Arnow test (*Arnow, 1937*). Briefly, 50 μL of 0.5 M aqueous HCl was added to 50 μL of culture supernatant. After mixing, 50 μL of an aqueous solution containing both sodium molybdate and sodium nitrite (0.1 g/mL each) was added, which produced a yellow color. Finally, 50 μL of 1 M aqueous NaOH was added followed by pipetting up and down to mix. This allowed the characteristic pink color to develop. Absorbance

was measured at 500 nm immediately using a Synergy HTX Multi-Mode Microplate Reader (BioTek) or SPECTROstar Nano (BMG LABTECH).

## Anaerobic activity-based purification of *E. lenta* A2 dopamine dehydroxylase

### Protein purification

Experiments were performed as described previously (*Maini Rekdal et al., 2019*), with minor modifications. All procedures were carried out under strictly anaerobic conditions at 4°C. Procedures outside the anaerobic chamber were performed in tightly sealed containers to prevent oxygen contamination. First, *E. lenta* A2 starter cultures were inoculated from single colonies into liquid BHI medium and were grown for 30 hr. Starter cultures were diluted 1:100 into 5 L of BHI medium containing 1% arginine and 10 mM formate and grown anaerobically at 37°C for 16 hr. Dopamine was added as a solid to a final concentration of 0.5 mM in the cultures. Cells were pelleted in 5 separate 1 L bottles by centrifugation (6000 rpm, 15 mins), and each pellet was resuspended in 20 mL of 20 mM Tris pH 8 containing 4 mg/mL SIGMAFAST protease inhibitor cocktail. Resuspended cells were then lysed using two rounds of sonication in an anaerobic chamber (Branson Sonifier 450, 2 min total, 10 s on, 40 s off, 25% amplitude). The lysates were then clarified by centrifugation (10800 rpm, 15 mins), and the soluble fractions were subjected to two rounds of ammonium sulfate precipitation. During the precipitation, three different tubes each containing 40 mL total clarified lysate were precipitated in parallel. Solid ammonium sulfate was first dissolved in these clarified lysates to a final concentration of 30% (w/v), and lysates were left for 1 hr and 20 min followed by centrifugation to pellet the precipitates (4000 rpm, 15 mins). The supernatant was saved, and the pellet was discarded. The supernatant was mixed with additional solid ammonium sulfate to achieve a final concentration of 40% (w/v) and left for 1 hr and 20 min. Following centrifugation (4000 rpm, 15 mins) and removal of supernatant, each pellet containing the precipitated proteins was re-dissolved in 20 mL 20 mM Tris pH 8 containing 0.5 M ammonium sulfate. The re-dissolved pellets were combined and centrifuged to remove particulates (10800 rpm, 15 mins). The resulting 60 mL solution was injected onto an FPLC (Bio-Rad BioLogic DuoFlow System equipped with GE Life Sciences DynaLoop90) for hydrophobic interaction chromatography (HIC) using 5 × 1 mL HiTrap Phenyl HP columns (GE Life Sciences, catalog# 17135101). Fractions were eluted with a gradient of 0.5 M to 0 M ammonium sulfate (in 20 mM Tris pH 8) at a flow rate of 1 mL/min and were tested for activity using the assay described below. The majority of the dopamine dehydroxylase activity eluted around 0.05 M-0.1 M ammonium sulfate. Active fractions displaying >50% conversion of dopamine were combined and injected onto the FPLC system described above for anion exchange chromatography using a UNO Q1 column (Bio-Rad, catalog# 720–0001) at a flow-rate of 1 mL/min. Fractions were eluted using a gradient of 0 to 1 M NaCl in 20 mM Tris pH eight and were tested for activity. The majority of the dopamine dehydroxylase activity eluted around 250 mM NaCl. Active fractions were combined and concentrated 20-fold using a spin concentrator with a 5 kDa cutoff (4000 rpm centrifugation speed). 250 µL of the concentrate was injected onto FPLC for size exclusion chromatography using an Enrich 24 mL column (Enrich SEC 650, 10*300 column, Bio-Rad, catalog# 780–1650). Fractions were eluted over a 26 mL volume run isocratically in 20 mM Tris pH 8 containing 250 mM NaCl and were subjected to activity assays. Active fractions were then combined and used for enzyme assays and were run on SDS-PAGE to assess the presence of protein. Absorbance at 280 nm was used to determine the protein concentration, using a predicted extinction coefficient of 317735 $M^{-1}$ $cm^{-1}$ for the dopamine dehydroxylase.

### Activity assays during protein purification

50 µL aliquots of fractions from FPLC runs were mixed, in the following order, with 1 µL electron donors (final concentration 1 mM each of methyl viologen, 1 mM diquat dibromide, 1 mM benzyl viologen, all dissolved in water), 2 µL sodium dithionite (2 mM final concentration, dissolved in water), and 1 µL substrate (500 µM final concentration, dissolved in water). The assay mixtures were left at room temperature in an anaerobic chamber for 12–14 hr to allow dopamine dehydroxylation to proceed, followed by assessment of activity using the colorimetric assay for catechol detection. Due to the inability of Dadh to survive freeze-thawing even in the presence of glycerol, the natively purified enzyme was always immediately used for enzyme assays.

## Assays of the *E. lenta* A2 dopamine dehydroxylase substrate scope

Active fractions from the size exclusion chromatography described above were combined and then diluted in 20 mM Tris pH 8 containing 250 mM NaCl to a final enzyme concentration of 0.1 μM. The enzyme mixture was transferred the wells of a 96 well plate, for a final volume of 50 μL in each well (VWR, catalog# 82006–636). 1 μL of substrate (in water, or 50:50 water:DMF for caffeic acid and catechin substrates) was then added at a final concentration of 500 μM. Following this, 1 μL of a solution containing electron donors (final concentration 1 mM each of methyl viologen, 1 mM diquat dibromide, 1 mM benzyl viologen, all dissolved in water) and 2 μL of sodium dithionite (2 mM final concentration, dissolved in water) were added. The resulting solution was mixed by pipetting and the 96-well plate was then sealed tightly with an aluminum seal. The enzyme assay mixtures were left at room temperature in an anaerobic chamber for 22 hr to allow dehydroxylation to proceed. The enzyme reaction mixtures were quenched by bringing the samples out of the anaerobic chamber and freezing at –20°C. These mixtures were then diluted 1:10 with LC-MS grade methanol and analyzed by LC-MS/MS. For the screen with physiologically relevant catechol substrates, samples containing caffeic acid were analyzed using Method H, hydrocaffeic acid and DOPAC were analyzed using Method F, catechin was analyzed using Method E, protocathecuic acid was analyzed using Method I, epinephrine was analyzed using Method J, norepinephrine was analyzed using Method G, and ellagic acid was analyzed using Method D. For the screen with dopamine analogs, all monohydroxylated, dihydroxylated, and trihydroxylated phenylethylamine analogs were analyzed using method B, *N*-methyldopamine was analyzed using Method C, methoxytyramine was analyzed using Method M, dihydroxybenzylamine was analyzed using Method K, hydroxytyrosol was analyzed using Method N, and aminotyramine was analyzed using Method L.

## Metabolism of dopamine analogs by *E. lenta* A2 cells

Cells were cultured in 96-well plates and all experiments were performed anaerobically. The strains screened for dopamine dehydroxylation have been previously described (*Koppel et al., 2018*; *Bisanz et al., 2018*). *E. lenta* A2 was inoculated from a single colony into 10 mL of BHI liquid medium and grown for 48 hr at 37°C to provide turbid starter cultures. These were diluted 1:10 in triplicate into 200 μL of fresh BHI medium containing 500 μM substrate (*p*-tyramine, dopamine, 3,4-dihydroxybenzylamine, or DL-norepinephrine). These cultures were grown for 48 hr at 37°C. Cultures were harvested by centrifugation at 4000 rpm for 10 min, and the supernatants were diluted 1:10 with LC-MS grade methanol. Samples containing dopamine or *p*-tyramine were analyzed using Method B, norepinephrine was analyzed using Method G, dihydroxybenzylamine was analyzed using Method K.

## RNA-sequencing experiments with *E. lenta* A2

We repeated the setup previously used in the RNA-sequencing experiment with dopamine (*Maini Rekdal et al., 2019*). Turbid 48 hr starter cultures of *E. lenta* in BHI medium were inoculated 1:100 into 5 mL of BHI medium containing 1% arginine and 10 mM formate, and cultures were grown at 37°C anaerobically. When the cultures reached $OD_{600}$ = 0.200, hydrocaffeic acid, (+)-catechin, *p*-tyramine, 3,4-dihydroxybenzylamine, DL-norepinephrine, or *N*-methyldopamine were added at final concentrations of 500 μM to triplicate cultures. All compounds except for (+)-catechin were dissolved in water; (+)-catechin was dissolved in DMF. Control cultures contained vehicle (water or DMF). Cultures were harvested when they reached $OD_{600}$ = 0.500. They were centrifuged for 15 min at 4000 rpm, and cell pellets were re-suspended in 500 μL Trizol reagent (ThermoFisher, catalog#: 15596026). Total RNA was isolated by first bead beating to lyse cells and then using the Zymo Research Direct-Zol RNA MiniPrep Plus kit (Catalog # R2070) according to the manufacturer's protocol. Illumina cDNA libraries were generated using a modified version of the RNAtag-Seq protocol (*Shishkin et al., 2015*). Briefly, 500 ng of total RNA was fragmented, depleted of genomic DNA, and dephosphorylated prior to its ligation to DNA adapters carrying 5'-AN8-3' barcodes with a 5' phosphate and a 3' blocking group. Barcoded RNAs were pooled and depleted of rRNA using the RiboZero rRNA depletion kit (Epicentre). These pools of barcoded RNAs were converted to Illumina cDNA libraries in three main steps: (i) reverse transcription of the RNA using a primer designed to the constant region of the barcoded adaptor; (ii) addition of a second adapter on the 3' end of the cDNA during reverse transcription using SmartScribe RT (Clonetech) as described (*Shishkin et al.,*

2015); (iii) PCR amplification using primers that target the constant regions of the 3' and 5' ligated adaptors and contain the full sequence of the Illumina sequencing adaptors. cDNA libraries were sequenced on Illumina HiSeq 2500. For the analysis of RNAtag-Seq data, reads from each sample in the pool were identified based on their associated barcode using custom scripts, and up to one mismatch in the barcode was allowed with the caveat that it did not enable assignment to more than one barcode. Barcode sequences were removed from the first read as were terminal G's from the second read that may have been added by SMARTScribe during template switching. Reads were aligned to the *Eggerthella lenta* A2 genome using BWA (*Li and Durbin, 2009*) and read counts were assigned to genes and other genomic features using custom scripts. Differential expression analysis was conducted with DESeq2 (*Love et al., 2014*) and/or edgeR (*Robinson et al., 2010*).

## RNA-sequencing experiments with *G. pamelaeae* 3C

Method 1 (compound added at mid-exponential phase): Turbid 48 hr starter cultures of *G. pamelaeae* 3C grown in BHI medium were inoculated 1:100 into triplicate Hungate tubes containing 20 mL BHI medium with 10 mM formate. When cultures reached $OD_{600}$ = 0.110, DOPAC (0.5 mM final) or vehicle (water) was added to the cultures. The cultures were then grown at 37°C anaerobically and harvested when they reached $OD_{600}$ = 0.185. They were centrifuged, and cell pellets were re-suspended in 500 µL Trizol reagent (ThermoFisher, catalog#: 15596026).

Method 2 (compound added at the beginning of growth): Turbid 48 hr starter cultures of *G. pamelaeae* 3C grown in BHI medium were inoculated 1:100 into triplicate hungate tubes containing 20 mL BHI with 10 mM formate and DOPAC (0.5 mM final) or vehicle (water). These cultures were then left to grow at 37°C anaerobically. When cultures reached $OD_{600}$ = 0.110, they were harvested. They were centrifuged, and cell pellets were re-suspended in 500 µL Trizol reagent (ThermoFisher, catalog#: 15596026).

RNA extraction and sequencing: this was performed using the exactly same setup as described above, except the reads were aligned to the genome of *Gordonibacter pamelaeae* 3C.

## Growth of *E. lenta* A2 in BHI with and without dopamine

Cells were cultured in Hungate tubes and all experiments were performed anaerobically. *E. lenta* A2 was inoculated from a single colony into 10 mL BHI liquid medium and grown for 48 hr at 37°C to provide turbid starter cultures. These were diluted 1:100 in triplicate into 5 mL BHI medium containing either 0.5 mM dopamine or vehicle. Growth was assessed by measuring the optical density at 600 nm using a Genesys 20 spectrophotometer (Thermo Scientific).

## Preparation of basal medium lacking electron acceptors

The medium was prepared as described previously, with minor modifications (*Maini Rekdal et al., 2019*). A 100-fold stock solution of salts was first prepared by dissolving 100 g NaCl, 50 g $MgCl_2 \cdot 6H_2O$, 20 g $KH_2PO_4$, 30 g $NH_4Cl$, 30 g KCl, 1.5 g $CaCl_2 \times 2H_2O$ in 1 L of water. Then, 10 mL of this solution was added to 1 L of water containing 1 g yeast extract (Beckton Dickinson #288260), 1 g tryptone (Beckton Dickinson #21175), and 0.25 mL of 0.1% resazurin (dissolved in MilliQ water). This medium was autoclaved. Following autoclaving, the medium was left to cool for 15 min in an atmosphere of air (outside the anaerobic chamber). After cooling, the following components were added using sterile technique: 10 mL of ATCC Trace element mix (ATCC, catalog# MD-TMS), 10 mL of Vitamin Supplement (ATCC, catalog# MD-VS), solid $NaHCO_3$ (SIGMA, 2.52 g, to give 30 mM) and solid L-cysteine HCl (SIGMA, 63 mg, to give 0.4 mM). The medium had a final pH of 7.2–7.3. The medium was then sparged with nitrogen gas (for how long) and was brought into the anaerobic chamber to equilibrate for at least 30 hr prior to use. In all experiments utilizing the basal medium, except for those experiments performed with *Gordonibacter pamelaeae* 3C or the screen for catechol metabolism by mammalian gut microbiota samples, sodium acetate was added at a final concentration of 10 mM at the time of bacterial inoculation. In experiments performed with *Gordonibacter pamelaeae* 3C, sodium formate was added at a final concentration of 10 mM. In the ex vivo experiments with the mammalian gut microbiota, neither acetate nor formate were added to the basal medium.

## Growth of single *E. lenta* strains in basal medium

Cells were cultured in hungate tubes and all experiments were performed anaerobically. *E. lenta* strains were inoculated from single colonies into 10 mL of BHI liquid medium and grown for 48–72 hr at 37°C to provide turbid starter cultures. These were diluted 1:100 in triplicate into 5 mL of basal medium containing 10 mM acetate and either 1 mM dopamine (in water) or vehicle (water). If applicable. molybdate (0.5 mM), tungstate (0.5 mM), DMSO (14 mM), or nitrate (1 mM) were added at the time of inoculation. Cultures were grown anaerobically for 36–72 hr at 37°C. Endpoint growth was assessed by measuring the optical density at 600 nm using a Genesys 20 spectrophotometer (Thermo Scientific). Catechol dehydroxylation was assessed at the end of growth in culture supernatants using the colorimetric method.

## Competition of *E. lenta* strains in basal medium

Cells were cultured in hungate tubes and all experiments were performed anaerobically. *E. lenta* strains W1BHI6 (Tet resistant non-metabolizer, and Valencia (Tet sensitive metabolizer) were inoculated from single colonies into individual tubes containing 10 mL of BHI liquid medium and grown for 48 hr at 37°C to provide turbid starter cultures. For the competition experiment, 50 µL of each starter culture of the two competing strains was combined in triplicate in 5 mL of basal medium containing 10 mM acetate and either 1 mM dopamine or vehicle (water). Following inoculation, cultures were grown anaerobically for 72 hr at 37°C. At the end of the incubation, growth of *E. lenta* was assessed. Cultures were serially diluted in PBS under anaerobic conditions, and 8 µL of each serial dilution ($10^{-1}$ through $10^{-7}$) was plated onto BHI plates containing 1% arginine (w/v) with and without 10 µg/mL Tetracycline using a spot plating method. Plates were grown at 37°C for 72 hr following by counting of colonies. To calculate the proportion of metabolizer in the W1BHI6/Valencia competition experiment, we selected a dilution where distinct colonies were clearly visible ($10^{-4}$-$10^{-5}$) and counted the number of colonies growing on the BHI 1% arginine Tetracycline plates (W1BHI6) as well as the colonies growing on the BHI 1% arginine plates (Both Valencia and W1BHI6). To get the number of metabolizer (Valencia) colonies, we subtracted the number of Tetraycline resistant colonies from the colonies on the no Tetracycline plate.

## Growth of *E. lenta* strains in the presence of a defined community

Cells were cultured in hungate tubes and all experiments were performed anaerobically. *E. lenta* strains, as well as *Enterococcus faecalis* OGR1F, *Escherichia coli* MG1655, *Bacteroides fragilis* ATCC 25285, *Clostridium sporogenes* ATCC 15579, *Edwarsiella tarda* ATCC 23685, were inoculated from single colonies into individual tubes containing 10 mL of BHI liquid medium and grown for 48–72 hr at 37°C to provide turbid starter cultures. Growth was assessed by measuring the optical density at 600 nm using a Genesys 20 spectrophotometer (Thermo Scientifc). These starter cultures were then diluted to a final $OD_{600}$ of 0.100 in BHI medium anaerobically. The defined community was created by combining equal volumes of all strains (after diluting each culture to $OD_{600}$ = 0.100) except for *E. lenta*. The community was then inoculated 1:100 in triplicate into 5 mL basal medium containing 10 mM acetate and either 1 mM dopamine or vehicle. *E. lenta* strains were then added by diluting the *E. lenta* starter cultures (normalized to $OD_{600}$ = 0.100) 1:50 into the tubes containing the defined community. Cultures were then grown anaerobically for 72 hr at 37°C. At the end of the incubation, growth of *E. lenta* was assessed. Cultures were serially diluted in PBS under anaerobic conditions, and 8 µL of each serial dilution ($10^{-1}$ through $10^{-7}$) was plated onto BHI plates containing 1% arginine (w/v) and 10 µg/mL Tetracycline (spot plating method). Plates were grown at 37°C for 72 hr following by counting of colonies.

## Human fecal samples used in this study

The human fecal samples used in this study have been previously described (*Maini Rekdal et al., 2019*). To prepare them for culturing, all samples were resuspended anaerobically in anaerobic PBS at a final concentration of 0.1 g/mL. The mixture was vortexed to produce a homogenous slurry and was then left for 30 min to let particulates settle. Aliquots of the supernatant were dissolved 50:50 with 40% glycerol and flash-frozen in liquid nitrogen, creating slurries that were used for anaerobic culturing of human fecal samples. Slurries were stored at –80°C and were thawed anaerobically at room temperature at the time of use.

## Growth of human fecal samples in basal medium with dopamine

### Culturing

Fecal slurries from n = 24 unrelated humans were diluted 1:100 into two different hungate tubes containing 5 mL of basal medium with 10 mM acetate and either 1 mM dopamine or vehicle (water). These fecal microbiota cultures were grown anaerobically for 72 hr at 37°C. Metabolism was then assessed in culture supernatants using the colorimetric method. In addition, cultures were spun down and the total community gDNA was extracted from the entire 5 mL of culture for downstream PCR and qPCR assays as detailed below. qPCR assays for *E. lenta* and *dadh* abundance in human fecal samples grown in basal medium with and without dopamine.

### qPCR assays

Assays were performed as previously described (*Maini Rekdal et al., 2019*). gDNA was extracted from the culture pellets generated in the experiments described above ('Growth of fecal samples in basal medium with dopamine') using the DNeasy UltraClean Microbial Kit. The extracted DNA from each culture was used for qPCR assays containing 10 μL of iTaq Universal SYBRgreen Supermix (Bio-rad, catalog 3: 1725121), 7 μL of water, and 10 μM each of forward and reverse primers. PCR was performed on a CFX96 Thermocycler (Bio-Rad), using the following program: initial denaturation at 95°C for 5 min 34 cycles of 95°C for 1 min, 60°C for 1 min, 72°C for 1 min. The program ended with a final extension at 34°C for five mins. The primers used were: 16S primers for *E. lenta* (*Haiser et al., 2013*): CAGCAGGGAAGAAATTCGAC and TTGAGCCCTCGGATTAGAGA; primers for dopamine dehydroxylase: GAGATCTGGTCCACCGTCAT and AGTGGAAGTACACCGGGATG (*Maini Rekdal et al., 2019*).

### Amplification of *dadh* and sequencing of SNP506

Amplification of full-length *dadh* and sequencing of the SNP at position 506 from human fecal samples grown in basal medium with and without dopamine gDNA was extracted from the culture pellets generated in the experiments described above ('Growth of fecal samples in basal medium with dopamine') using the DNeasy UltraClean Microbial Kit. The extracted DNA from each culture was used for PCR assays containing 10 μL of Phusion High-Fidelity PCR Master mix with HF buffer (NEB, catalog# M0531L), 7 μL of water, and 10 μM each of forward and reverse primers. The primers used to amplify the full-length dopamine dehydroxylase from these samples were ATGGGTAACC TGACCATG and TTACTCCCTCCCTTCGTA. PCR was performed on a C1000 Touch Thermocycler (Bio-Rad), using the following program: initial denaturation at 98°C for 30 s, 34 cycles of 98°C for 10 s, 61°C for 15 s, 72°C for 2.5 mins. The program ended with a final extension at 72°C for five mins. Amplicons were purified using the Illustra GFX PCR DNA and Gel Band Purification Kit (GE Health-care, catalog# 28-9034-70) and were sequenced using Sanger sequencing (Eton Biosciences) for the region containing the SNP at position 506 using primers GGGGTGTCCATGTTGCCGGT and ACCGGCTACGGCAACGGC. Sequence chromatograms were analyzed in Ape Plasmid Editor (version 2.0.47), and the single nucleotide polymorphism (SNP) at position 506 was called by visual inspection compared to results obtained from control cultures of *E. lenta* strains.

## Screen of gut Actinobacteria for metabolism of catechols

This procedure was performed in an anaerobic chamber (Coy Laboratory Products, atmospheric conditions: 20% $CO_2$, 2–2.5% $H_2$, and the balance $N_2$)—equilibrating media and consumables to the atmosphere prior to use—until centrifugation, which was performed using a benchtop centrifuge. The 96-well plates used in this experiment were purchased from VWR (catalog# 10861–562). Into the wells of flat-bottom 96-well plates, 100 μL of BHI medium supplemented with L-cysteine-HCl (0.05%, w/v), *L*-arginine (1%, w/v), and sodium formate (10 mM) (referred to here as BHI++) were aliquoted. Seed cultures were prepared by inoculating wells, in triplicate, with Actinobacterial strains that were cultured on BHI++ agar plates. Additional wells served as sterile controls. Plates were sealed with tape and incubated at 37°C for 12 to 18 hr to afford dense cultures. Next, 99 μL of BHI++ medium containing 500 μM of compound were aliquoted into the wells of a 96-well plate. To these wells, 1 μL of dense seed culture (or sterile control) was added. Plates were sealed and incubated at 37°C for 24 or 48 hr. Plates were then centrifuged at 2000 rpm for 10 min at 4°C, and the supernatant was aspirated and transferred to a fresh 96-well plate. An aliquot (35 μL) of supernatant was then

immediately screened via the catechol colorimetric assay (described above in 'Colorimetric assay for catechol detection'). Absorbance was immediately measured at 500 nm using a plate reader (Spectrostar Nano, BMG LABTECH). A standard curve (2-fold serial dilutions, 1000–15.6 μM in BHI++) was simultaneously prepared, developed, and analyzed using the conditions listed above. The catechol concentrations in bacterial cultures were normalized to the sterile control. To confirm metabolism of (+)-catechin, DOPAC, and hydrocaffeic acid, the incubations were repeated following the same procedure with minor modifications. Strains were grown in BHI for 48 hr anaerobically at 37°C. Cultures were harvested by centrifugation and were then analyzed by LC-MS. To prepare samples for LC-MS, 20 μL of the culture supernatant was diluted 1:10 with 180 μL of methanol, followed by centrifugation at 4000 rpm for 10 min to pellet particulates, salts, and proteins. 50 μL of the resulting supernatant was then transferred to a 96-well plate and 5 μL of the supernatant was injected onto the instrument using Method E for catechin and Method F for hydrocaffeic acid and DOPAC. Following this screen, select strains were re-cultured to confirm absence/presence of metabolism. The following stock solutions were used in the screens: dihydroxymandelic acid (50 mM in water), dopamine (50 mM in water), protocatechuic acid (50 mM in ethanol), L-dopa (50 mM in 0.5 M HCl), norepinephrine (50 mM in 0.5 M HCl), epinephrine (50 mM in 0.5 M HCl), DOPAC (50 mM in 0.5 M HCl), Hydrocaffeic acid (50 mM in ethanol), caffeic acid (50 mM in ethanol), (+)-catechin (50 mM in ethanol), (+/−)-catechin (50 mM in ethanol), (−)-epicatechin (50 mM in DMSO), methyldopa (solid dissolved directly into the media at 0.5 mM final concentration), carbidopa (solid dissolved directly into the media at 0.5 mM final concentration), dihydroxybenzoic acid (50 mM in methanol). Hydroxytyrosol (50 mM in water), enterobactin (10 mM in DMSO), apomorphine (50 mM in DMSO).

## Confirmation of dehydroxylation of (+)-catechin and hydrocaffeic acid by *E. lenta* A2

Cells were cultured in hungate tubes and all experiments were performed anaerobically. *E. lenta* A2 was inoculated from a single colony into 10 mL BHI liquid medium and grown for 48 hr at 37°C to provide turbid starter cultures. These were diluted 1:100 in triplicate into 5 mL of BHI medium containing either 0.5 mM hydrocaffeic acid (in water), 0.5 mM (+)-catechin (in DMF), or vehicle (water or DMF). After 48 hr of anaerobic growth at 37°C, cultures were harvested by centrifugation and were then analyzed by LC-MS. To prepare samples for LC-MS, 20 μL of the culture supernatant was diluted 1:10 with 180 μL of methanol, followed by centrifugation at 4000 rpm for 10 min to pellet particulates, salts, and proteins. 50 μL of the resulting supernatant was then transferred to a 96-well plate and 5 μL of the supernatant was injected onto the instrument using Method E for catechin and Method F for hydrocaffeic acid.

## Confirmation of dehydroxylation of DOPAC by *Gordonibacter pamelaeae* 3C

Cells were cultured in hungate tubes and all experiments were performed anaerobically. *G. pamelaeae* 3C was inoculated from a single colony into 10 mL of BHI liquid medium and grown for 48 hr at 37°C to provide turbid starter cultures. These were diluted 1:100 in triplicate into 5 mL of BHI medium containing 10 mM formate and 0.5 mM DOPAC or vehicle (water). After 72 hr of anaerobic growth at 37°C, the cultures were harvested by centrifugation and were then analyzed by LC-MS. To prepare samples for LC-MS, 20 μL of the culture supernatant was diluted 1:10 with 180 μL of methanol, followed by centrifugation at 4000 rpm for 10 min to pellet particulates, salts, and proteins. 50 μL of the resulting supernatant was then transferred to the LC-MS 96-well plate and 5 μL of the supernatant was injected onto the instrument using Method F for DOPAC.

## PBS resuspension assays for inducibility and oxygen sensitivity of catechol dehydroxylases

### Assays with *E. lenta* A2

Cells were cultured in hungate tubes and all experiments were performed anaerobically. *E. lenta* A2 was inoculated from a single colony into 10 mL of BHI liquid medium and grown for 48 hr at 37°C to provide turbid starter cultures. These were diluted 1:100 in triplicate into 10 mL of BHI medium containing 1% arginine and 10 mM formate and either 0.5 mM dopamine (in water), 0.5 mM hydrocaffeic acid (in water), 0.5 mM (+)-catechin (in DMF), or vehicle (water or DMF). After 18 hr of anaerobic

growth at 37°C, cultures had reached an $OD_{600}$ of 0.700 and were harvested by centrifugation (4000 rpm, 15 min). The bacterial pellets were resuspended anaerobically in 10 mL of pre-reduced PBS to wash the cells, followed by an additional round of centrifugation to pellet the washed cells (4000 rpm, 15 min). The cells were then resuspended in 5 mL of pre-reduced PBS. 0.1 mL aliquots of this resuspension was transferred to Eppendorf tubes containing either vehicle or 0.5 mM catechol substrate. The samples were vortexed briefly and incubated anaerobically at room temperature for 20 hr to allow for metabolism to proceed. To assess the impact of oxygen on the metabolism of catechols, 0.1 mL of the PBS resuspension in an Eppendorf tube was brought outside the anaerobic chamber, followed by addition of 0.5 mM substrate in the presence of atmospheric oxygen. The samples were vortexed briefly and were incubated at room temperature for 20 hr. Samples were then analyzed by LC-MS. To prepare samples for LC-MS, 20 µL of the culture supernatant was diluted 1:10 with 180 µL of methanol, followed by centrifugation at 4000 rpm for 10 min to pellet particulates, salts, and proteins. 50 µL of the resulting supernatant was then transferred to the LC-MS 96-well plate and 5 µL of the supernatant was injected onto the instrument using Method B for dopamine, Method E for catechin and Method F for hydrocaffeic acid.

### Assays with *Gordonibacter pamelaeae* 3C

Cells were cultured in hungate tubes and all experiments were performed anaerobically. *G. pamelaeae* 3C was inoculated from a single colony into 10 mL of BHI liquid medium and grown for 48 hr at 37°C to provide turbid starter cultures. These were diluted 1:100 in triplicate into 10 mL of BHI medium containing 10 mM formate and either 0.5 mM DOPAC or vehicle. After 18 hr of anaerobic growth at 37°C, cultures had reached an $OD_{600}$ of 0.180 and were harvested by centrifugation (4000 rpm, 15 min). The bacterial pellets were resuspended anaerobically in 10 mL of pre-reduced PBS to wash the cells, followed by an additional round of centrifugation to pellet the washed cells. The cells were then resuspended in 5 mL of pre-reduced PBS. 0.1 mL aliquots of this resuspension were transferred to Eppendorf tubes containing either vehicle or 0.5 mM catechol substrate. The samples were vortexed briefly and incubated anaerobically at room temperature for 20 hr to allow for metabolism to proceed. To assess the impact of oxygen on DOPAC metabolism, 0.1 mL of the PBS resuspension in an Eppendorf tube was brought outside the anaerobic chamber, followed by addition of 0.5 mM substrate in the presence of atmospheric oxygen. The samples were vortexed briefly and were incubated at room temperature for 20 hr. Samples were then analyzed by LC-MS. To prepare samples for LC-MS, 20 µL of the culture supernatant was diluted 1:10 with 180 µL of methanol, followed by centrifugation at 4000 rpm for 10 min to pellet particulates, salts, and proteins. 50 µL of the resulting supernatant was then transferred to the LC-MS 96-well plate and 5 µL of the supernatant was injected onto the instrument using Method F.

## Effect of tungstate on growth and catechol dehydroxylation by *E. lenta* A2 and *G. pamelaeae* 3C

### Assays with *E. lenta* A2

Starter cultures of *E. lenta* A2 were grown over 48 hr in 10 mL of BHI medium and then inoculated 1:100 into 200 µL of BHI medium containing either 500 µM dopamine (in water), 500 µM (+)-catechin (in DMF), or 500 µM hydrocaffeic acid (in water), and either sodium tungstate (0.5 mM, in water), sodium molybdate (0.5 mM, in water), or vehicle (water or DMF). Cultures were grown for 48 hr anaerobically at 37°C and were harvested by centrifugation. Supernatants were dissolved 1:10 in LC-MS grade methanol and analyzed using LC-MS/MS Methods B, E, or F described above. Experiments were performed anaerobically, and cultures were grown in 96-well plates (VWR, catalog# 29442–054).

### Assays with *Gordonibacter pamelaeae* 3C

Starter cultures of *G. pamelaeae* 3C were grown over 48 hr in 10 mL of BHI medium and then inoculated 1:100 into 200 µL of BHI medium containing 500 µM DOPAC and either sodium tungstate (0.5 mM, in water), sodium molybdate (0.5 mM, in water), or vehicle (water). Cultures were grown for 48 hr anaerobically at 37°C and were harvested by centrifugation. Supernatants were dissolved 1:10 in LC-MS grade methanol and analyzed using LC-MS/MS Method F described above. Experiments

were performed anaerobically, and cultures were grown in 96-well plates (VWR, catalog# 29442–054).

## Lysate assays for transcriptional and biochemical specificity of dehydroxylases from *G. pamelaeae* 3C and *E. lenta* A2

### Assays in *E. lenta* A2

Bacterial cultures were grown in hungate tubes. All bacterial growth and lysate experiments were performed in an anaerobic chamber. Lysis and sample processing took place in an anaerobic chamber kept at 4°C. *E. lenta* A2 was inoculated from a single colony into 10 mL of BHI liquid medium and grown for 48 hr at 37°C to provide turbid starter cultures. These were diluted 1:100 in triplicate into 50 mL of BHI medium containing 1% arginine and 10 mM formate and either 1 mM dopamine, 1 mM hydrocaffeic acid, 1 mM (+)-catechin, or vehicle. After 18 hr of anaerobic growth at 37°C, cultures had reached $OD_{600}$ of 0.700 and were harvested by centrifugation. The bacterial pellets were resuspended anaerobically in 10 mL of cold, pre-reduced PBS to wash the cells, followed by an additional round of centrifugation to pellet the washed cells. The washed cells from each culture were then transferred to an Eppendorf tube and resuspended in 1.4 mL of lysis buffer (20 mM Tris pH 8 containing 4 mg/mL SIGMAFAST protease inhibitor cocktail). The cells were lysed using sonication in an anaerobic chamber. 50 µL of this lysate was transferred in triplicate to a 96 well plate (VWR, catalog# 82006–636). 1 µL of substrate was then added to each of the replicates at a final concentration of 0.5 mM. These samples were incubated anaerobically at room temperature for 28 hr to allow for metabolism to proceed. Samples were then analyzed by LC-MS. To prepare samples for LC-MS, 20 µL of the culture supernatant was diluted 1:10 with 180 µL of methanol, followed by centrifugation at 4000 rpm for 10 min to pellet particulates, salts, and proteins. 50 µL of the resulting supernatant was then transferred to the LC-MS 96-well plate and 5 µL of the supernatant was injected onto the instrument using method B for dopamine, method E for catechin and method F for hydrocaffeic acid.

### Assays in *Gordonibacter pamelaeae* 3C

Bacterial cultures were grown in hungate tubes. All bacterial growth and lysate experiments were performed in an anaerobic chamber. Lysis and sample processing took place in an anaerobic chamber kept at 4°C. *G. pamelaeae* 3C was inoculated from a single colony into 10 mL of BHI liquid medium and grown for 48 hr at 37°C to provide turbid starter cultures. These were diluted 1:100 in triplicate into 50 mL of BHI medium containing 10 mM formate and 1 mM DOPAC or vehicle. After 18 hr of anaerobic growth at 37°C, cultures had reached $OD_{600}$ of 0.180 and were harvested by centrifugation. The bacterial pellets were resuspended anaerobically in 10 mL of cold, pre-reduced PBS to wash the cells, followed by an additional round of centrifugation to pellet the washed cells. The washed cells from each culture were then transferred to an Eppendorf tube and resuspended in 1.4 mL of lysis buffer (20 mM Tris pH 8 containing 4 mg/mL SIGMAFAST protease inhibitor cocktail). The cells were lysed using sonication in an anaerobic chamber. 50 µL of this lysate was transferred in triplicate to a 96 well plate (VWR, catalog# 82006–636). 1 µL substrate was then added to each of the replicates, for a final concentration of 0.5 mM. These samples were left anaerobically at room temperature for 28 hr to allow for metabolism to proceed. Samples were then analyzed by LC-MS. To prepare samples for LC-MS, 20 µL of the culture supernatant was diluted 1:10 with 180 µL of methanol, followed by centrifugation at 4000 rpm for 10 min to pellet particulates, salts, and proteins. 50 µL of the resulting supernatant was then transferred to the LC-MS 96-well plate and 5 µL of the supernatant was injected onto the instrument using LC-MS/MS Method F described above.

## Comparative genomics among human gut Actinobacteria

To characterize the distribution of Cadh, Hcdh, Dodh among our gut Actinobacterial strain library, we performed a tBLASTn search. We queried the genomes for Cadh, Hcdh, Dodh and used 90% coverage, 75% amino acid identity, and e-value = 0 as the cutoff for assessing the presence of each dehydroxylase.

## Anaerobic activity-based purification of *E. lenta* A2 hydrocaffeic acid dehydroxylase

### Protein purification

All procedures were carried out under strictly anaerobic conditions at 4°C. Procedures outside the anaerobic chamber were performed in tightly sealed containers to prevent oxygen contamination. First, *E. lenta* A2 starter cultures were inoculated from single colonies into liquid BHI medium and were grown for 30 hr. Starter cultures were diluted 1:100 into 8 L of BHI medium containing 1% arginine and 10 mM formate and grown anaerobically at 37°C for 17 hr. Hydrocaffeic acid (1M stock solution in water) was added to a final concentration of 0.5 mM in the cultures. Cells were pelleted in 8 separate 1 L bottles by centrifugation (6000 rpm, 15 mins), and each pellet was resuspended in 12 mL of 20 mM Tris pH 8 containing 4 mg/mL SIGMAFAST protease inhibitor cocktail. Resuspended cells were then lysed using two rounds of sonication in an anaerobic chamber (Branson Sonifier 450, 2 min total, 10 s on, 40 s off, 25% amplitude). The lysates were then clarified by centrifugation (10800 rpm, 15 mins), and the soluble fractions were subjected to two rounds of ammonium sulfate precipitation. During the precipitation, two different tubes each containing 40 mL total clarified lysate were precipitated in parallel. Solid ammonium sulfate was first dissolved in these clarified lysates to a final concentration of 30% (w/v), and lysates were left for 1 hr and 30 min followed by centrifugation to pellet the precipitates (3300 rcf, 15 mins). The supernatant was saved, and the pellet was discarded. The supernatant was mixed with additional solid ammonium sulfate to achieve a final concentration of 40% (w/v) and left for 1 hr and 30 min. Following centrifugation (3300 rcf rpm, 15 mins) and removal of supernatant, each pellet containing the precipitated proteins was re-dissolved in 20 mL 20 mM Tris pH 8 containing 0.5 M ammonium sulfate. The re-dissolved pellets were combined and centrifuged to remove particulates (10800 rpm, 15 mins). The resulting 80 mL solution was injected onto an FPLC (Bio-Rad BioLogic DuoFlow System equipped with GE Life Sciences DynaLoop90) for hydrophobic interaction chromatography (HIC) using 4 × 1 mL HiTrap Phenyl HP columns strung together (GE Life Sciences, catalog# 17135101). Fractions were eluted with a gradient of 0.5 M to 0 M ammonium sulfate (in 20 mM Tris pH 8) at a flow rate of 1 mL/min and were tested for activity using the assay described below. The majority of the hydrocaffeic acid dehydroxylase activity eluted around 0.2 M-0.5M ammonium sulfate. Active fractions displaying >50% conversion of hydrocaffeic acid were combined and injected onto the FPLC system described above for anion exchange chromatography using a UNO Q1 column (Bio-Rad, catalog# 720–0001) at a flow-rate of 1 mL/min. Fractions were eluted using a gradient of 0 to 1 M NaCl in 20 mM Tris pH eight and were tested for activity. The majority of the hydrocaffeic acid dehydroxylase activity eluted around 200–400 mM NaCl. Active fractions were combined and concentrated 20-fold using a spin concentrator with a 5 kDa cutoff (3300 rcf centrifugation speed). 500 µL of the concentrate was injected onto FPLC for size exclusion chromatography using an Enrich 24 mL column (Enrich SEC 650, 10*300 column, Bio-Rad, catalog# 780–1650). Fractions were eluted over a 26 mL volume run isocratically in 20 mM Tris pH 8 containing 250 mM NaCl and were subjected to activity assays. Active fractions were then combined and used for enzyme assays and were run on SDS-PAGE to assess the presence of protein.

### Activity assays during protein purification

100 µL aliquots of fractions from FPLC runs were mixed, in the following order, with 1 µL of methyl viologen (0.5 mM final concentration), 2 µL sodium dithionite (1 mM final concentration, dissolved in water), and 1 µL substrate (500 µM final concentration, dissolved in water). The assay mixtures were left at room temperature in an anaerobic chamber for 12–14 hr to allow dehydroxylation to proceed, followed by assessment of activity using the colorimetric assay for catechol detection.

## Proteomics of *E. lenta* A2 hydrocaffeic acid dehydroxylase

### Sample preparation

The cut-out gel band was washed twice with 50% aqueous acetonitrile for five minutes followed by drying in a SpeedVac. The gel was then reduced with a volume sufficient to completely cover the gel pieces (100 µL) of 20 mM TCEP in 25 mM TEAB at 37°C for 45 min. After cooling to room temp, the TCEP solution was removed and replaced with the same volume of 10 mM iodoacetamide Ultra (Sigma) in 25 mM TEAB and kept in the dark at room temperature for 45 min. Gel pieces were

washed with 200 µL of 100 mM TEAB (10 min). The gel pieces were then shrunk with acetonitrile. The liquid was then removed followed by swelling with the 100 mM TEAB again and dehydration/shrinking with the same volume of acetonitrile. All of the liquid was removed, and the gel was completely dried in a SpeedVac for ~20 min. 0.06 µg/5 µL of trypsin in 50 mM TEAB was added to the gel pieces and the mixture was placed in a thermomixer at 37°C for about 15 min. 50 µL of 50 mM TEAB was added to the gel slices. The samples were vortexed, centrifuged, and placed back in the thermomixer overnight. Samples were digested overnight at 37°C. Peptides were extracted with 50 µL 20 mM TEAB for 20 min and 1 change of 50 µL 5% formic acid in 50% acetonitrile at room temp for 20 min while in a sonicator. All extracts obtained were pooled into an HPLC vial and were dried using a SpeedVac to the desired volume (~50 µL). This sample was used for protein identification by LC-MS/MS, as described below.

## Mass spectrometry

Each sample was submitted for a single LC-MS/MS experiment that was performed on an LTQ Orbitrap Elite (Thermo Fischer) equipped with a Waters (Milford, MA) NanoAcquity HPLC pump. Peptides were separated using a 100 µm inner diameter microcapillary trapping column packed first with approximately 5 cm of C18 Reprosil resin (5 µm, 100 Å, Dr. Maisch GmbH, Germany) followed by ~20 cm of Reprosil resin (1.8 µm, 200 Å, Dr. Maisch GmbH, Germany). Separation was achieved through applying a gradient of 5–27% ACN in 0.1% formic acid over 90 min at 200 nL min$^{-1}$. Electrospray ionization was enabled through applying a voltage of 1.8 kV using a home-made electrode junction at the end of the microcapillary column and sprayed from fused silica pico tips (New Objective, MA). The LTQ Orbitrap Elite was operated in data-dependent mode for the mass spectrometry methods. The mass spectrometry survey scan was performed in the Orbitrap in the range of 395–1,800 m/z at a resolution of $6 \times 10^4$, followed by the selection of the twenty most intense ions (TOP20) for CID-MS2 fragmentation in the Ion trap using a precursor isolation width window of 2 m/z, AGC setting of 10,000, and a maximum ion accumulation of 200 ms. Singly charged ion species were not subjected to CID fragmentation. Normalized collision energy was set to 35 V and an activation time of 10 ms. Ions in a 10 ppm m/z window around ions selected for MS2 were excluded from further selection for fragmentation for 60 s. The same TOP20 ions were subjected to HCD MS2 event in Orbitrap part of the instrument. The fragment ion isolation width was set to 0.7 m/z, AGC was set to 50,000, the maximum ion time was 200 ms, normalized collision energy was set to 27 V and an activation time of 1 ms for each HCD MS2 scan.

## Mass spectrometry data analysis

Raw data were submitted for analysis in Proteome Discoverer 2.1.0.81 (Thermo Scientific) software. Assignment of MS/MS spectra were performed using the Sequest HT algorithm by searching the data against a protein sequence database including a custom database from *Eggerthella lenta* A2 and other sequences such as human keratins and common lab contaminants. Sequest HT searches were performed using a 20 ppm precursor ion tolerance and requiring each peptide's N-/C-termini to adhere with trypsin protease specificity, while allowing up to two missed cleavages. Cysteine carbamidomethyl (+57.021) was set as a static modification while methionine oxidation (+15.99492 Da) was set as a variable modification. A MS2 spectra assignment false discovery rate (FDR) of 1% on protein level was achieved by applying the target-decoy database search. Filtering was performed using a Percolator (64bit version). For quantification, a 0.02 m/z window centered on the theoretical m/z value of each the six reporter ions and the intensity of the signal closest to the theoretical m/z value was recorded. Reporter ion intensities were exported in result file of Proteome Discoverer 2.1 search engine as an excel tables.

## Assays of partially purified *E. lenta* A2 hydrocaffeic acid dehydroxylase substrate scope

Active fractions from the size exclusion chromatography described above were combined and then diluted 1:5 in 20 mM Tris pH 8 containing 250 mM NaCl. The enzyme mixture was transferred the wells of a 96 well plate, for a final volume of 50 µL in each well (VWR, catalog# 82006–636). 1 µL of substrate (in water for hydrocaffeic acid, dopamine, and DOPAC, or 50:50 water:DMF for (+)-catechin) was then added at a final concentration of 500 µM. Following this, 1 µL of a solution containing

methyl viologen (1 mM final concentration) and 2 µL of sodium dithionite (2 mM final concentration, dissolved in water) were added. The resulting solution was mixed by pipetting and the 96-well plate was then sealed tightly with an aluminum seal. The enzyme assay mixtures were left at room temperature in an anaerobic chamber for 26 hr to allow dehydroxylation to proceed. The enzyme reaction mixtures were quenched by bringing the samples out of the anaerobic chamber and freezing at –20°C. These mixtures were then diluted 1:10 with LC-MS grade methanol and analyzed by LC-MS/MS. Samples containing hydrocaffeic acid and DOPAC were analyzed using Method F, catechin was analyzed using Method E, and dopamine was analyzed using Method C.

## Metagenomic analysis of catechol dehydroxylase abundance and prevalence across human patients

To generate estimates of the enzyme prevalence in human populations, the amino acid sequences of Cldh, Dodh, Dadh, Hcdh, and Cadh were searched against a non-redundant gut microbiome gene catalogue using BLASTP with a minimum 70% percent identity, query coverage and target coverage and used to extract per-sample gene abundances from a collection of human metagenomes (10.1093/bioinformatics/btv382). High identity matches were obtained for all queries (81.7%, 81.7%, 100%, 99.5%, and 99.9% respectively over >99% target coverage) with a second lower-identity match observed for Hcdh (75.7%) for which abundances were summed with the higher identity hit. Next, to account for repeated sampling of individuals, the median gene abundance across a subject's samples was calculated and carried forward for prevalence estimates leading to a total of 1872 human subjects considered (*Nayfach et al., 2015*). Prevalence was then calculated as a rolling function of minimum abundance.

## Incubation of human fecal samples with $d_4$-dopamine, hydrocaffeic acid, (+)-catechin, and $d_5$-DOPAC

### Culturing

20 µL of fecal slurries from n = 12 unrelated humans were diluted 1:50 into three independent cultures containing 980 µL of BHI and 0.5 mM substrate. As positive controls, saturated cultures of 20 µL *E. lenta* A2 or *G. pamelaeae* 3C (grown from a glycerol stock for 48 hr in BHI) were added in triplicate to 980 µL BHI and 0.5 mM hydrocaffeic acid, (+)-catechin, $d_4$-dopamine for *E. lenta* A2, and $d_5$-DOPAC for *G. pamelaeae* 3C. These cultures were grown anaerobically for 72 hr at 37°C. The cultures were harvested by centrifugation and 20 µL was then diluted 1:10 with LC-MS grade methanol and analyzed by LC-MS/MS using Method F for samples grown with hydrocaffeic acid, Method P for $d_4$-dopamine, Method O for $d_5$-DOPAC, and Method E for catechin. In addition, the total community gDNA was extracted from the remaining 980 µL of culture growth with $d_5$-DOPAC for downstream qPCR assays as detailed below. The experiment was performed in a 96 well plate (Agilent, catalog# A696001000).

### qPCR assays

gDNA was extracted from the culture pellets generated in the experiments described above ('Incubation of human fecal samples with $d_4$-dopamine, hydrocaffeic acid, (+)-catechin, and $d_5$-DOPAC') using the DNeasy UltraClean Microbial Kit. 40 ng of the extracted DNA (2 µL) from each culture was used for qPCR assays containing 10 µL of iTaq Universal SYBRgreen Supermix (Bio-rad, catalog 3: 1725121), 7 µL of water, and 10 µM each of forward and reverse primers. PCR was performed on a CFX96 Thermocycler (Bio-Rad), using the following program: initial denaturation at 95°C for 5 min 34 cycles of 95°C for 1 min, 60°C for 1 min, 72°C for 1 min. The program ended with a final extension at 34°C for five mins. The primers used for *dodh* were TACGCCTACAACAGCTCCAA and ACATCATCTGGGGCGGATAC. Data analysis was performed in graphpad prism (version 8).

## Phylogenetic analysis of relationship between catechol dehydroxylases and other characterized members of the bis-molybdopterin guanine dinucleotide enzyme family

For phylogenetic analysis of the bis-MGD enzymes, we gathered sequences that have been previously used to study the evolution of bis-MGD enzymes (*Schoepp-Cothenet et al., 2012*). However, we also added sequences to capture additional diversity of biochemically characterized bis-MGD

enzymes that were not included in the original tree described in *Schoepp-Cothenet et al. (2012)*. In particular, we performed a pBLAST search in Uniprot using perchlorate reductase (Uniprot ID# PCRA_DECAR), ethylbenzene dehydrogenase (Uniprot ID# Q5NZV2_AROAE), acetylene hydratase (Uniprot ID# AHY_PELAE), and pyrogallol transhydroxylase (Uniprot ID# PGTL_PELAC) as the queries, and collected sequences with 85–90% amino acid ID. In addition, we added the sequences of Dadh, Hcdh, Cadh from *E. lenta* A2, and Dodh and Cldh (*Bess et al., 2020*) from G. *pamelaeae* 3C. The sequences were combined with those reported in *Schoepp-Cothenet et al. (2012)* and were aligned in Geneious (version 11) using MUSCLE. We subsequently used FastTree (standard settings, 20 rate categories of sites) to create a maximum likelihood tree. The tree files were uploaded to the Interactive Tree of Life web server (https://itol.embl.de/) to annotate the trees (*Letunic and Bork, 2016*).

## Construction of sequence similarity network of the bis-molybdopterin guanine dinucleotide enzyme family

A SSN was generated using the EFI-EST tool (http://efi.igb.illinois.edu/efi-est/) on July 15 2017 (*Gerlt et al., 2015*). In particular, we generated an SSN of the molybdopterin dinucleotide binding domain enzyme superfamily (PF01568), including sequences between 600 and 1400 amino acids in length and using an initial alignment score of e-150. Nodes represented sequences with 75% amino acid identity. The SSN was imported into Cytoscope v 3.2.1 and visualized with the 'Organic layout' setting. The alignment score cutoff was increased to e-167 until the groups of biochemically characterized enzymes included in our phylogenetic analysis (*Figure 6*) separated from each other into putatively isofunctional clusters.

## Phylogenetic analysis of catechol dehydroxylases encoded by gut Actinobacteria and environmental isolates

To identify additional diversity beyond the newly identified putative dehydroxylases from this study, we created a database containing putative homologs from a collection of 26 previously sequenced Actinobacterial genomes (*Bisanz et al., 2018*), as well as from genomes publicly available through NCBI. First, the *Eggerthella lenta* A2 dopamine dehydroxylase (Dadh) protein sequence was used as the query sequence for a tBLASTn search of 26 previously sequenced Actinobacterial genomes (*Bisanz et al., 2018*) (April 23, 2019). The genomes were loaded in Geneious (version 11) and hits with an amino acid ID of >30% and e-value of e-34 were considered potential dehydroxylase hits and were saved. This cutoff was chosen because sequences captured within this window more closely resembled the acetylene hydratase and Dadh than to any other biochemically characterized moco enzyme, as assessed by percent amino acid identity. In addition, we used the representative *Gordonibacter* enzyme Cldh as a separate query to identify the more distantly related, smaller enzymes from *Gordonibacter* that were not detected when using the large, multi-subunit Dadh as the query. Specifically, we used tBLASTn to search the 26 Actinobacterial genomes for the *Gordonibacter pamelaeae* 3C Cldh protein sequence. Hits from *Paraeggerthella hongongensis*, *Gordonibacter pamelaeae* 3C and *Gordonibacter sp.* 28C, the only organisms containing these smaller Cldh-like enzymes in our collection, were saved. Again, amino acid ID of >30% and e-value of e-34 were considered potential hits because sequences captured within this window more closely resembled the acetylene hydratase and Dadh than to any other biochemically characterized moco enzyme, as assessed by percent amino acid identity. The hits from our searches with Cldh and Dadh were combined into a preliminary database in Geneious. To expand the sequence diversity within this database, we used Cldh and Dadh as queries for two separate tBLASTn searches in NCBI (nucleotide collection). To ensure that we captured diversity beyond human gut microbes, we excluded *Gordonibacter* and *Eggerthella* as organisms in the tBLASTn searches for Cldh and Dadh queries, respectively. For the two searches, sequences of >29% amino acid ID and e-value of e-55 were considered potential dehydroxylase hits. This was a more conservative cutoff than we used with human Actinobacteria and was selected based on the observation that the pBLAST alignment of Dadh and Cldh has an e value of e-45% and 29% amino acid ID. The sequences retrieved from NCBI were added to the database already containing the hits from searches of the 26 Actinobacterial genomes. In addition, we added the biochemically characterized *E. coli* bis-MGD enzyme DMSO reductase (DmsA, Uniprot ID#P18775) to this database as the outgroup. This sequence was also used as the root of

the tree. For phylogenetic analysis of these sequences, we first aligned sequences in Geneious using MUSCLE and removed sequences that were 95% identical to each other (considered duplicates). After deleting these duplicate sequences, we re-aligned the sequences using MUSCLE (standard settings) and subsequently used FastTree (standard settings, 20 rate categories of sites) to create a maximum likelihood tree. The tree files were uploaded to the Interactive Tree of Life web server (https://itol.embl.de/) to annotate the trees (*Letunic and Bork, 2016*).

## Phylogenetic analysis of relationship between representative dehydroxylase homologs (from *Figure 7—figure supplement 3* and *Supplementary file 3b*) and other characterized members of the bis-molybdopterin guanine dinucleotide enzyme family

Once we had constructed the two trees described above (*Figures 6* and *7* in the main text) and uncovered dehydroxylase homologs in gut and environmental bacteria, we wanted to explore the phylogenetic relationship between these enzymes and the broader bis-molybdopterin guanine dinucleotide (bis-MGD) enzyme family. To do this, we added the representative sequences from *Figure 7—figure supplement 3* and *Supplementary file 3c* to the sequence database already described in 'Phylogenetic analysis of relationship between catechol dehydroxylases and other characterized members of the bis-molybdopterin guanine dinucleotide enzyme family'. Using MUSCLE, we aligned the newly added sequences with the sequences represented on the tree (*Figure 6* in main text). We then used FastTree (standard settings, 20 rate categories of sites) in Geneious (version 11) to generate the tree seen in *Figure 7—figure supplement 4*. The tree files were uploaded to the Interactive Tree of Life web server (https://itol.embl.de/) to annotate the trees (*Letunic and Bork, 2016*).

## Mammalian fecal samples used in this study

The collection of fecal samples from mammals (n = 12 different species, n = 3 individuals per species) has been previously described (*Reese et al., 2018*). To prepare these samples for culturing, all samples were resuspended anaerobically in pre-reduced PBS at a final concentration of 0.1 g/mL. The mixture was vortexed to produce a homogenous slurry and was then left for 30 min to let particulates settle. Aliquots of the supernatant were dissolved 50:50 with 40% glycerol in water and flash-frozen in liquid nitrogen, creating slurries. These slurries were stored at –80°C and were defrosted anaerobically at room temperature at the time of use.

## Screen for catechol dehydroxylation by mammalian gut microbiota samples

Mammalian fecal slurries were prepared as described above and were defrosted by incubation at room temperature at the time of use. 20 µL of each slurry was then combined with 980 µL of basal medium containing 500 µM each of dopamine, (+) catechin, DOPAC, or hydrocaffeic acid. Each individual sample was grown in one well, with the n = 3 individual samples for each animal serving as the biological replicates. Control wells contained compound but no bacteria. Samples were grown anaerobically at 37°C for 96 hr in a 96-well plate (Agilent Technology, catalog# A696001000). Following growth, we first assessed the total microbial growth by measuring the $OD_{600}$ in a plate reader (BioTek Synergy HTX). Cultures were then harvested by centrifugation and 50 µL of supernatant was transferred to a new 96-well plate, at which time the catechol colorimetric assay was used to assess total dehydroxylation by the complex microbial community. Samples that had potential catechol depletion as assessed by the colorimetric assay were then further analyzed by LC-MS. To prepare samples for LC-MS, 20 µL of the culture supernatant was diluted 1:10 with 180 µL of methanol, followed by centrifugation at 4000 rpm for 10 min to pellet particulates, salts, and proteins. 50 µL of the resulting supernatant was then transferred to the LC-MS 96-well plate and 5 µL of the supernatant was injected onto the instrument using Method A for dopamine, Method E for catechin, and Method F for DOPAC and hydrocaffeic acid.

## Construction of a mammalian phylogenetic tree

The mammalian phylogenetic tree was generated using the Automatic Phylogenetic Tree Generator (aptg, version 0.1.0) script in R (version 3.5.1). Mammals not part of the aptg database were added

manually to the tree using additional information about the mammalian phylogeny as a reference (*Reese et al., 2019*). The mammalian icons were adapted under a Creative Commons license (https://creativecommons.org/licenses/by/3.0/) at phylopic (http://phylopic.org), including Alpaca logo (made by Steven Traver), Bison (Lukasiniho). Cow (Steven Traver), Dog (Tracy A Heath), Fox (Anthony Caravaggi), Guinea pig (Zimices), Mouse (Madeleine Price Ball), Pig (Steven Traver), Rabbit (Steven Traver), Rabbit (Steven Traver), Rat (Rebecca Groom), Sheep (Zimices), and Wolf (Tracy A. Heath).

## Acknowledgements

We acknowledge the Broad Institute Microbial Omics Core (MOC) for assistance with RNA sequencing analysis and experimental design, Dr. Rachel Carmody, Dr. Richard Losick, and Dr. Rachelle Gaudet (all at Harvard University) for helpful discussions. We thank Dr. Lihan Zhang (Westlake University) for help with phylogenetic analysis. We thank Dr. Liz Ortiz (UC Irvine) for critical experimental feedback. We thank Dr. Elizabeth Nolan (MIT) for providing enterobactin. We thank Dr. Rachel Carmody and Dr. Aspen Reese for kindly providing the mammalian fecal samples.

## Additional information

### Competing interests

Peter J Turnbaugh: Reviewing editor, *eLife*. Emily P Balskus: has consulted for Merck, Novartis, and Kintai Therapeutics; is on the Scientific Advisory Boards of Kintai Therapeutics and Caribou Biosciences. The other authors declare that no competing interests exist.

### Funding

| Funder | Grant reference number | Author |
|---|---|---|
| David and Lucile Packard Foundation | 2013-39267 | Emily P Balskus |
| Bill and Melinda Gates Foundation | OPP1158186 | Emily P Balskus |
| University of California, Irvine | | Elizabeth N Bess |
| National Science Foundation | Graduate research fellowship | Vayu Maini Rekdal |
| Howard Hughes Medical Institute | Gilliam Fellowship | Vayu Maini Rekdal |
| Harvard University | Ardis and Robert James Graduate Research Fellowship | Vayu Maini Rekdal |
| National Institutes of Health | 5T32GM007598-38 | Vayu Maini Rekdal |
| Human Frontier Science Program | Postdoctoral research fellowship, LT001561/2017-C | Michael U Luescher |
| Harvard University | Undergraduate research funding | Sina Kiamehr |

The funders had no role in study design, data collection and interpretation, or the decision to submit the work for publication.

### Author contributions

Vayu Maini Rekdal, Conceptualization, Data curation, Formal analysis, Validation, Investigation, Visualization, Methodology, Project administration; Paola Nol Bernadino, Conceptualization, Data curation, Formal analysis, Validation, Investigation, Visualization, Methodology; Michael U Luescher, Conceptualization, Data curation, Formal analysis, Investigation, Visualization, Methodology; Sina Kiamehr, Data curation, Formal analysis, Validation, Investigation, Visualization; Chip Le,

Investigation; Jordan E Bisanz, Formal analysis, Investigation; Peter J Turnbaugh, Conceptualization, Resources; Elizabeth N Bess, Conceptualization, Resources, Supervision, Funding acquisition, Methodology, Project administration; Emily P Balskus, Conceptualization, Supervision, Funding acquisition, Project administration

### Author ORCIDs
Vayu Maini Rekdal (iD) https://orcid.org/0000-0002-8699-8902
Jordan E Bisanz (iD) https://orcid.org/0000-0002-8649-1706
Peter J Turnbaugh (iD) https://orcid.org/0000-0002-0888-2875
Elizabeth N Bess (iD) https://orcid.org/0000-0003-0349-0423
Emily P Balskus (iD) https://orcid.org/0000-0001-5985-5714

### Decision letter and Author response
Decision letter https://doi.org/10.7554/eLife.50845.sa1
Author response https://doi.org/10.7554/eLife.50845.sa2

## Additional files

### Supplementary files
• Supplementary file 1. Tables detailing the bacterial strains used in this study, the chemical structures and sources of compounds used in this study, and the RNA-sequencing results for experiments with DL-norepinephrine. Supplementary file 1a. Physiologically relevant catechol substrates used in assays with natively purified dopamine dehydroxylase from *E. lenta* A2. Supplementary file 1b. Commercially available and synthesized dopamine analogs used in assays with natively purified dopamine dehydroxylase from *E. lenta* A2. Supplementary file 1c. Differentially expressed genes upon exposure of *Eggerthella lenta* A2 to 0.5 mM DL-norepinephrine relative to vehicle (>|2|-fold difference, FDR < 0.1). The dopamine dehydroxylase (dadh) in *E. lenta* A2 is highlighted in red. Data are from n = 3 bacterial cultures for each condition. Supplementary file 1d. Bacterial strains used in this study Supplementary file 1e. Catechol substrates used in screen of gut Actinobacteria for catechol metabolism. 3,4-DHMA stands for 3,4-dihydroxymandelic acid. 2,3-DHBA stands for 2,3-dihydroxybenzoic acid.

• Supplementary file 2. Tables containing the proteomics data supporting the identification of hydrocaffeic acid dehydroxylase, the percent homology between dehydroxylases, and the RNA-sequencing results for experiments with DOPAC, hydrocaffeic acid, and (+)-catechin. Supplementary file 2a. Differentially expressed genes upon exposure of *Eggerthella lenta* A2 to 0.5 mM hydrocaffeic acid relative to vehicle (>|2|-fold difference, FDR < 0.1). *E. lenta* A2 was grown with 500 μM hydrocaffeic acid or a vehicle control in BHI medium containing 1% (w/v) arginine and 10 mM formate. The catalytic subunit of the putative hydrocaffeic acid dehydroxylase (hcdh), as well as its predicted 4Fe-4S and membrane anchor partners, are highlighted in red. Data are from n = 4 cultures for each condition. The experiment was performed once. Supplementary file 2b. Differentially expressed genes upon exposure of *Eggerthella lenta* A2 to 0.5 mM (+)-catechin relative to vehicle (>|2|-fold difference, FDR < 0.1). *E. lenta* A2 was grown with 500 μM (+)-catechin or a vehicle control in BHI medium containing 1% (w/v) arginine and 10 mM formate. The catalytic subunit of the putative catechin dehydroxylase (cadh), as well as its predicted 4Fe-4S and membrane anchor partners, are highlighted in red. Data are from n = 3 cultures from each condition. The experiment was performed once. Supplementary file 2c. Percent amino acid identity between the putative dehydroxylases from *G. pamelaeae* 3C and *E. lenta* A2. Ach stands for acetylene hydratase from *P. acetylenicus*. This protein is the closest biochemically characterized homolog to the newly identified catechol dehydroxylases. Colors represent % identity, with identity going from low (blue) to high (red). Supplementary file 2d. Differentially expressed genes upon exposure of *G. pamelaeae* 3C to 0.5 mM DOPAC relative to vehicle (>|2|-fold difference, FDR < 0.1) when catechol was added during exponential phase. *G. pamelaeae* 3C was grown with 500 μM DOPAC or a vehicle control in BHI medium containing 10 mM formate. The catalytic subunit of the putative DOPAC dehydroxylase (dodh), as well as its predicted 4Fe-4S partner, are highlighted in red. The data are from n = 3 cultures for each condition. The experiment was performed once. Supplementary file 2e. Differentially expressed genes upon

exposure of *G. pamelaeae* 3C to 0.5 mM DOPAC relative to vehicle (>|2|-fold difference, FDR < 0.1) when catechol was added during the beginning of growth. *G. pamelaeae* 3C was grown with 500 µM DOPAC or a vehicle control in BHI medium containing 10 mM formate. Cells were harvested in mid-exponential phase when metabolism appeared. The catalytic subunit of the putative DOPAC dehydroxylase (*dodh*), as well as its predicted 4Fe-4S partner, are highlighted in red. Data are from n = 3 cultures for each condition. The experiment was performed once. Supplementary file 2f. Proteomics identification of proteins in the band of interest during the activity-based native purification of the hydrocaffeic acid dehydroxylase from *E. lenta* A2. The band highlighted with a red asterisk in *Figure 4—figure supplement 4* was cut out and subjected to proteomics. This band was confirmed to contain the catalytic subunit of the hydrocaffeic acid dehydroxylase (highlighted in red). The predicted size of 133.9 kDa (a) is for the full peptide containing the Twin Arginine Translocation (TAT) signal sequence. The predicted size of 130.5 kDa (b) is for the processed, mature peptide where the TAT signal sequence has been removed, as is suggested from the coverage map in *Figure 4—figure supplement 5*.

• Supplementary file 3. Tables containing accession numbers for protein sequences used in bioinformatics analyses of catechol dehydroxylase enzymes. Supplementary file 3a. Accession numbers (Uniprot and Genbank) of putative Eggerthella and Gordonibacter dehydroxylases identified in this study. Supplementary file 3b. Accession numbers of bis-MGD enzymes used to generate the phylogenetic tree of the bis-MGD enzyme family (*Figure 6*). Supplementary file 3c. Accession numbers (Uniprot) and organismal origin of representative dehydroxylases identified from phylogenetic tree in *Figure 7—figure supplement 3*. Accession numbers (Uniprot and Genbank) of putative Eggerthella and Gordonibacter dehydroxylases identified in this study.

• Transparent reporting form

### Data availability

All data generated or analysed during this study are included in the manuscript and supporting files. Source data files have been provided for Figures 1-4 and Figure 6. RNA-Seq data has been deposited into the Sequence Read Archive available by way of BioProject PRJNA557713 for Eggerthella lenta A2 and PRJNA557714 for Gordonibacter pamelaeae 3C.

The following datasets were generated:

| Author(s) | Year | Dataset title | Dataset URL | Database and Identifier |
|---|---|---|---|---|
| Vayu Maini Rekdal, Emily P Balskus | 2019 | RNA-sequencing in E. lenta A2 | https://www.ncbi.nlm.nih.gov/bioproject/PRJNA557713 | NCBI BioProject, PRJNA557713 |
| Vayu Maini Rekdal, Emily P Balskus | 2019 | RNA-sequencing in G. pamelaeae 3C | https://www.ncbi.nlm.nih.gov/bioproject/PRJNA557714 | NCBI BioProject, PRJNA557714 |

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

**Appendix 1**

# Materials, methods, references, and characterization data for synthesis of dopamine analogs

## General materials and methods

All reactions were performed in dried glassware under an atmosphere of dry $N_2$. Reaction mixtures were stirred magnetically unless otherwise indicated and monitored by thin layer chromatography (TLC) on Merck precoated glass-backed silica gel 60 F-254 0.25 mm plates with visualization by fluorescence quenching at 254 nm. TLC plates were stained using a potassium permanganate solution. Chromatographic purification of products (flash column chromatography) was performed on Silicycle Silica Flash F60 (230–400 Mesh) silica gel using a forced flow of eluent at 0.3–0.5 bar. Concentration of reaction product solutions and chromatography fractions under reduced pressure was performed by rotary evaporation at 35–40°C at the appropriate pressure and then at rt, ca. 0.1 mmHg (vacuum pump) unless otherwise indicated.

All chemicals were purchased from Acros, Aldrich, Fluka, Merck, ABCR, TCI, Alfa Aesar or Strem and used as such unless stated otherwise. Commercial grade reagents and solvents were used without further purification except as indicated below. Toluene, diethylether ($Et_2O$), tetrahydrofuran (THF) and dichloromethane ($CH_2Cl_2$) were purified by pressure filtration through activated alumina. N,N-Dimethylformamide (DMF), acetonitrile ($CH_3CN$), and ethanol (EtOH) were used as purchased. Yields given refer to chromatographically purified and spectroscopically pure compounds unless otherwise stated.

Infrared (IR) spectra were recorded on a Bruker ALPHA FT-IR spectrophotometer and reported as wavenumber ($cm^{-1}$) of the absorption maxima for the range between 4000 $cm^{-1}$ and 750 $cm^{-1}$ with only major peaks reported. [1]H NMR and [13]C NMR spectra were recorded on a Varian-Inova-500 500 MHz, 125 MHz spectrometer. [1]H NMR chemical shifts are expressed in parts per million ($\delta$) downfield from tetramethylsilane (with the $CHCl_3$ peak at 7.26 ppm, MeOH peak at 3.31, DMSO peak at 2.50, and acetone peak at 2.05 used as a standard). [13]C NMR chemical shifts are expressed in parts per million ($\delta$) downfield from tetramethylsilane (with the central peak of $CHCl_3$ at 77.16 ppm, MeOH peak 49.00, DMSO peak at 39.52, and acetone peak at 29.84 used as a standard). All [13]C spectra were measured with complete proton decoupling. NMR coupling constants (J) are reported in Hertz (Hz), and splitting patterns are indicated as follows: br, broad; s, singlet; d, doublet; dd, doublet of doublet; t, triplet; m, multiplet. High-resolution mass spectrometric measurements (HRMS) were performed on an Accurate-Mass 6530 Q-TOF LC/MS (Agilent) using dual electrospray ionization (ESI).

## Preparation of 2-aminoethylbenzenediol /- triol derivatives

All reactions were carried out with degassed solvents under a positive pressure of nitrogen

## 2-PHENYLACETIC ACID DERIVATIVES AS STARTING MATERIALS

**Appendix 1—scheme 1.** Preparation of 2-phenylacetic acid derivatives.

**Appendix 1—scheme 2.** Reduction of benzyl carboxylic acids.

## General carboxylicacid reduction procedure

$BH_3 \bullet SMe_2$ (2.0 M in THF; 1.30 equiv) was added dropwise to a solution of commercially available di-/trimethoxyphenylacetic acid (1.00 equiv) in THF (0.25 M) at 0°C. The resulting mixture was allowed to warm to rt over 3 hr and stirring was continued for 14 hr while a colorless solid formed. The obtained suspension was cooled to 0°C and carefully quenched with the dropwise addition of saturated aqueous $NaHCO_3$. The layers were separated, and the aqueous layer was extracted with EtOAc (3 × 50 mL). The combined organic layers were washed with $H_2O$ (2 × 20 mL), brine (2 × 20 mL), dried over anhydrous $Na_2SO_4$, filtered, and concentrated under reduced pressure to yield analytically pure alcohol **S1–S5** that was used in the next step without further purification.

**Appendix 1—chemical structure 1.** Dimethoxyphenethyl alcohol (S1).

**2-(2,3-Dimethoxyphenyl)ethan-1-ol (S1).** Following the general carboxylic acid reduction procedure using 2-(2,3-dimethoxyphenyl)acetic acid (2.00 g, 10.2 mmol), alcohol **S1** was obtained as a colorless oil (1.85 g, quant.). [1]H NMR (500 MHz, $CDCl_3$): δ 7.01 (t, $J$ = 7.9 Hz, 1H), 6.88–6.75 (m, 2H), 3.87 (s, 3H), 3.84 (s, 3H), 3.84 (t, $J$ = 6.5 Hz, 2H), 2.91 (t, $J$ = 6.5 Hz, 2H), 1.76 (br s, OH); [13]C NMR (125 MHz, $CDCl_3$): δ 152.8, 147.4, 132.6, 124.2, 122.6, 111.0, 63.4, 60.7, 55.7, 33.8. The spectral characteristics were identical to those reported in the current literature, which fails to report the signals for the OMe and one of the $CH_2$ groups (*Macchia et al., 1993*).

**Appendix 1—chemical structure 2.** Dimethoxyphenethyl alcohol (S2).

**2-(3,5-Dimethoxyphenyl)ethan-1-ol (S2).** Following the general carboxylic acid reduction procedure using 2-(3,5-dimethoxyphenyl)acetic acid (1.00 g, 5.10 mmol), alcohol **S2** was obtained as a colorless oil (920 mg, quant.). [1]H NMR (500 MHz, $CDCl_3$): δ 6.39 (d, $J$ = 2.4 Hz,

2H), 6.36–6.33 (m, 1H), 3.86 (t, $J$ = 6.4 Hz, 2H), 3.79 (s, 6H), 2.82 (t, $J$ = 6.4 Hz, 2H), 1.45 (br s, OH). The spectral characteristics were identical to those reported in the current literature (*Yoshida et al., 2016*).

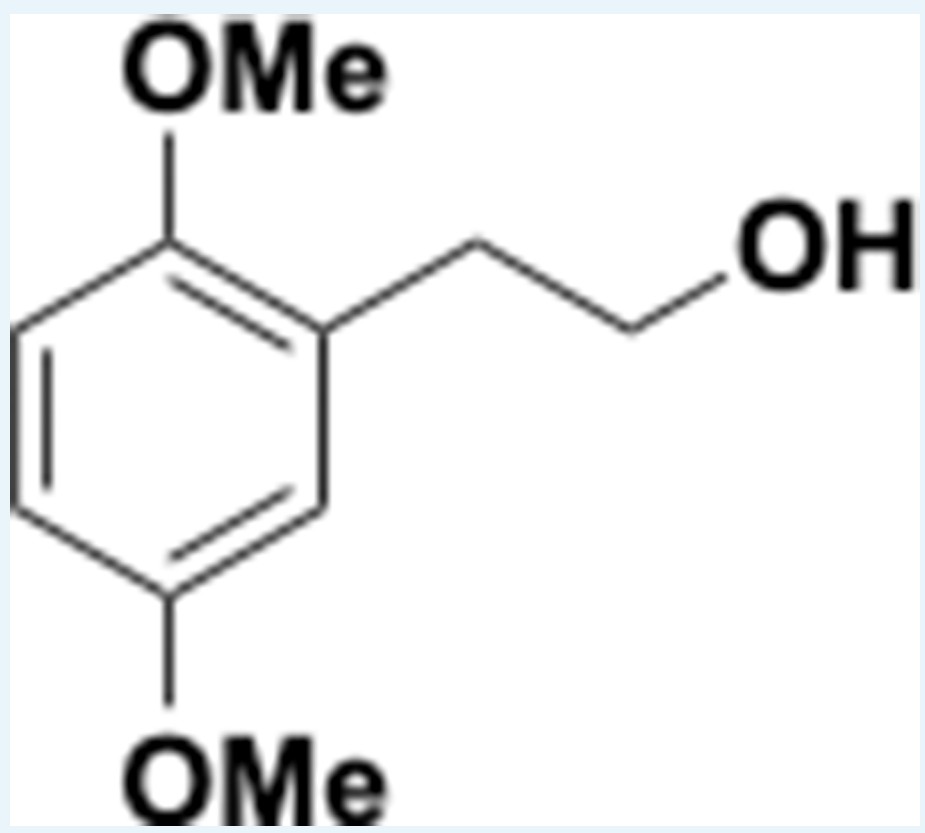

**Appendix 1—chemical structure 3.** Dimethoxyphenethyl alcohol (S3).

**2-(2,5-Dimethoxyphenyl)ethan-1-ol** (**S3**). Following the general carboxylic acid reduction procedure using 2-(2,5-dimethoxyphenyl)acetic acid (2.00 g, 10.2 mmol), alcohol **S3** was obtained as a colorless oil (1.70 g, 92% yield). [1]H NMR (500 MHz, CDCl$_3$): δ 6.80 (d, $J$ = 8.6 Hz, 1H), 6.77–6.71 (m, 2H), 3.83 (t, $J$ = 6.4 Hz, 2H), 3.79 (s, 3H), 3.76 (s, 3H), 2.88 (t, $J$ = 6.4 Hz, 2H), 1.72 (br s, OH). The spectral characteristics were identical to those reported in the current literature (*Gerdes et al., 1996*).

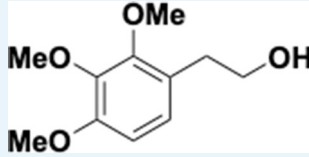

**Appendix 1—chemical structure 4.** Trimethoxyphenethyl alcohol (S4).

**2-(2,3,4-Trimethoxyphenyl)ethan-1-ol** (**S4**). Following the general carboxylic acid reduction procedure using 2-(2,3,4-trimethoxyphenyl)acetic acid (2.00 g, 8.84 mmol), alcohol **S4** was obtained as a colorless oil (1.87 g, quant.). IR (thin film) ν 3348, 2926, 2850, 1769, 1658, 1602, 1498, 1395, 1091, 840 cm$^{-1}$; [1]H NMR (500 MHz, CDCl$_3$): δ 6.86 (d, $J$ = 8.4 Hz, 1H), 6.63 (d, $J$ = 8.4 Hz, 1H), 3.90 (s, 3H), 3.87 (s, 3H), 3.84 (s, 3H), 3.80 (t, $J$ = 6.4 Hz, 2H), 2.83 (t, $J$ = 6.4 Hz, 2H), 1.79 (br s, OH); [13]C NMR (125 MHz, CDCl$_3$): δ 152.6, 152.1, 142.4, 124.7, 124.7, 107.5, 63.5, 61.0, 60.8, 56.1, 33.6; ESI-HRMS calcd for C$_{11}$H$_{17}$O$_4$ [M+H] 213.1121, found 213.1128.

**Appendix 1—chemical structure 5.** Trimethoxyphenethyl alcohol (S5).

**2-(2,4,6-Trimethoxyphenyl)ethan-1-ol (S5).** Following the general carboxylic acid reduction procedure using 2-(2,4,6-trimethoxyphenyl)acetic acid (1.00 g, 4.42 mmol), alcohol **S5** was obtained as a colorless oil (940 mg, quant.). [1]H NMR (500 MHz, CDCl$_3$): δ 6.14 (s, 2H), 3.80 (s, 3H), 3.80 (s, 6H), 3.71 (t, J = 6.4 Hz, 3H), 2.89 (t, J = 6.4 Hz, 2H), 1.96 (br s, OH); [13]C NMR (125 MHz, CDCl$_3$): δ 159.8, 159.2, 107.6, 90.7, 63.1, 55.8, 55.5, 26.2. The spectral characteristics were identical to those reported in the current literature (*Lal et al., 1987*).

**Appendix 1—scheme 3.** Mitsunobu reaction on phenethyl alcohols.

## General Mitsunobu reaction protocol

Diethyl azodicarboxylate (DEAD; 40% in toluene; 1.10 equiv) was added dropwise over 5–10 min to a solution of PPh$_3$ (1.15 equiv), phthalimide (1.15 equiv) and the corresponding alcohol **S1–S5** (1.00 equiv) in THF (0.15 M) at 0°C. The resulting mixture was allowed to warm to rt over 3 hr and stirring was continued for 14 hr. The resulting pale-yellow solution was concentrated under reduced pressure and purified by flash column chromatography to afford the desired phthalimide protected amine **S6–S10**.

**Appendix 1—chemical structure 6.** Dimethoxyphenethyl phthalimide (S6).

**2-(2,3-Dimethoxyphenethyl)isoindoline-1,3-dione (S6).** Following the general Mitsunobu reaction protocol, purification by flash column chromatography (hexanes:EtOAc 5:1) afforded phthalimide protected amine **S6** as a colorless solid (2.90 g, 85% yield) using alcohol **S1** (2.00 g, 11.0 mmol) as starting material. [1]H NMR (500 MHz, CDCl$_3$): δ 7.85–7.78 (m, 2H), 7.73–7.66 (m, 2H), 6.94 (t, J = 7.9 Hz, 1H), 6.83–6.74 (m, 2H), 3.95–3.91 (m, 2H), 3.89 (s, 3H), 3.83 (s, 3H), 3.07–2.97 (m, 2H); [13]C NMR (125 MHz, CDCl$_3$): δ 168.2, 152.8, 147.7, 133.8, 132.2, 131.9, 123.8, 123.1, 122.3, 111.3, 60.8, 55.7, 38.5, 29.1. The spectral characteristics were identical to those reported in the current literature (*Selvakumar and Ramanathan, 2011*).

**Appendix 1—chemical structure 7.** Dimethoxyphenethyl phthalimide (S7).

**2-(3,5-Dimethoxyphenethyl)isoindoline-1,3-dione (S7).** Following the general Mitsunobu reaction protocol, purification by flash column chromatography (hexanes:EtOAc 3:1) afforded phthalimide protected amine **S7** as a colorless solid (1.70 g, 99% yield) using alcohol **S2** (1.00

g, 5.50 mmol) as starting material. [1]H NMR (500 MHz, CDCl$_3$): δ 7.84 (dd, $J$ = 5.4, 3.1 Hz, 2H), 7.71 (dd, $J$ = 5.4, 3.1 Hz, 2H), 6.45–6.38 (m, 2H), 6.32 (t, $J$ = 2.2 Hz, 1H), 3.96–3.89 (m, 2H), 3.75 (s, 6H), 2.96–2.91 (m, 2H); [13]C NMR (125 MHz, CDCl$_3$): δ 168.3, 161.0, 140.4, 134.0, 132.2, 123.3, 106.8, 99.0, 55.4, 39.2, 35.0; $R_f$ = 0.33 (hexanes:EtOAc 3:1). The spectral characteristics were identical to those reported in the current literature (*Selvakumar and Ramanathan, 2011*).

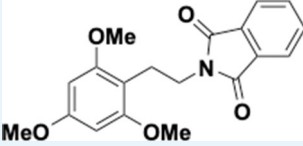

**Appendix 1—chemical structure 8.** Dimethoxyphenethyl phthalimide (S8).

**2-(2,5-Dimethoxyphenethyl)isoindoline-1,3-dione** (**S8**). Following the general Mitsunobu reaction protocol, purification by flash column chromatography (hexanes:EtOAc 5:1) afforded phthalimide protected amine **S8** as a colorless solid (3.15 g, 98% yield) using alcohol **S3** (1.85 g, 10.2 mmol) as starting material. [1]H NMR (500 MHz, CDCl$_3$): δ 7.79–7.71 (m, 2H), 7.68–7.59 (m, 2H), 6.70–6.63 (m, 3H), 3.90 (t, $J$ = 7.1 Hz, 2H), 3.66 (s, 3H), 3.62 (s, 3H), 2.93 (t, $J$ = 7.1 Hz, 2H); [13]C NMR (125 MHz, CDCl$_3$): δ 168.1, 153.3, 152.0, 133.7, 132.1, 127.6, 123.0, 116.6, 112.2, 111.1, 55.7, 55.6, 37.8, 29.6; $R_f$ = 0.28 (hexanes:EtOAc 5:1). The spectral characteristics were identical to those reported in the current literature (*Selvakumar and Ramanathan, 2011*).

**Appendix 1—chemical structure 9.** Trimethoxyphenethyl phthalimide (S9).

**2-(2,3,4-Trimethoxyphenethyl)isoindoline-1,3-dione** (**S9**). Following the general Mitsunobu reaction protocol, purification by flash column chromatography (hexanes:EtOAc 5:1) afforded phthalimide protected amine **S9** as a colorless solid (1.55 g, 96% yield) using alcohol **S4** (1.00 g, 4.71 mmol) as starting material. [1]H NMR (500 MHz, CD$_3$OD): δ 7.85–7.70 (m, 4H), 6.78 (d, $J$ = 8.5 Hz, 1H), 6.60 (d, $J$ = 8.5 Hz, 1H), 3.86 (t, $J$ = 6.8 Hz, 2H), 3.82 (s, 3H), 3.77 (s, 3H), 3.65 (s, 3H), 2.93–2.87 (m, 2H); [13]C NMR (125 MHz, CD$_3$OD): δ 169.7, 154.1, 153.5, 143.4, 135.3, 135.2, 126.0, 124.1, 123.9, 108.7, 61.3, 60.9, 56.4, 39.7, 29.8; $R_f$ = 0.18 (hexanes:EtOAc 3:1). The spectral characteristics were identical to those reported in the current literature (*Selvakumar and Ramanathan, 2011*).

**Appendix 1—chemical structure 10.** Trimethoxyphenethyl phthalimide (S10).

**2-(2,4,6-Trimethoxyphenethyl)isoindoline-1,3-dione** (**S10**). Following the general Mitsunobu reaction protocol, purification by flash column chromatography (hexanes:EtOAc 3:1) afforded phthalimide protected amine **S9** as a colorless solid (1.60 g, 99% yield) using alcohol **S5** (1.00 g, 4.71 mmol) as starting material. IR (thin film) ν 3248, 2928, 2850, 1772, 1715, 1634, 1607, 1439, 1205, 1139, 809 cm$^{-1}$; [1]H NMR (500 MHz, CDCl$_3$): δ 7.84–7.73 (m,

2H), 7.71–7.62 (m, 2H), 6.00 (s, 2H), 3.85 (t, $J$ = 6.3 Hz, 2H), 3.76 (s, 3H), 3.59 (s, 6H), 2.97 (t, $J$ = 6.3 Hz, 2H); $^{13}$C NMR (125 MHz, CDCl$_3$): δ 168.4, 160.0, 159.2, 133.6, 132.5, 122.9, 107.4, 90.2, 55.5, 55.3, 37.6, 21.6; R$_f$ = 0.20 (hexanes:EtOAc 3:1); ESI-HRMS calcd for C$_{18}$H$_{18}$NO$_4$ [M+H] 312.1230, found 312.1223.

**Appendix 1—scheme 4.** Hydrazinolysis of phenethyl phthalimides.

## General phthalimide deprotection protocol

Hydrazine monohydrate (10.0 equiv) was added to a suspension of the phthalimide protected amine **S6 – S10** in EtOH (0.15 M). The resulting solution was heated to reflux for 1.5 hr while colorless solids crashed out. The resulting suspension was allowed to cool to rt before H$_2$O (20–50 mL) was added in one portion. Stirring was continued to afford a clear solution that was extracted with EtOAc (3 × 50 mL). The combined organic layers were washed with H$_2$O (3 × 20 mL), brine (2 × 20 mL), dried over anhydrous Na$_2$SO$_4$, filtered and concentrated under reduced pressure to yield analytically pure amine **S11–S15** that was used in the next step without further purification.

**Appendix 1—chemical structure 11.** Dimethoxyphenethyl amine (S11).

   **2-(2,3-Dimethoxyphenyl)ethan-1-amine (S11).** Following the general phthalimide deprotection protocol using phthalimide protected amine **S6** (1.00 g, 3.21 mmol), primary amine **S11** was obtained as a colorless oil (500 mg, 86% yield). $^1$H NMR (500 MHz, CD$_3$OD): δ 6.99 (dd, $J$ = 8.0, 7.7 Hz, 1H), 6.88 (dd, $J$ = 8.0, 1.5 Hz, 1H), 6.78 (dd, $J$ = 7.7, 1.6 Hz, 1H), 3.84 (s, 3H), 3.79 (s, 3H), 2.88–2.80 (m, 2H), 2.80–2.73 (m, 2H); NH$_2$-group is not visible; $^{13}$C NMR (125 MHz, CD$_3$OD): δ 154.2, 148.6, 134.3, 125.1, 123.4, 112.1, 61.0, 56.2, 43.5, 34.5. The spectral characteristics were identical to those reported in the current literature (*Yang et al., 2013*).

**Appendix 1—chemical structure 12.** Dimethoxyphenethyl amine (S12).

   **2-(3,5-Dimethoxyphenyl)ethan-1-amine (S12).** Following the general phthalimide deprotection protocol using phthalimide protected amine **S7** (400 mg, 1.28 mmol), primary amine **S12** was obtained as a colorless oil (140 mg, 60% yield). $^1$H NMR (500 MHz, CD$_3$OD): δ 6.41–6.36 (m, 2H), 6.35 (t, $J$ = 2.3 Hz, 1H), 3.76 (s, 6H), 2.92 (t, $J$ = 7.3 Hz, 2H), 2.72 (t, $J$ = 7.3 Hz, 2H), NH$_2$-group is not visible; $^{13}$C NMR (125 MHz, CD$_3$OD): δ 162.5, 142.6, 107.8, 99.3, 55.7, 43.6, 39.4. The spectral characteristics were identical to those reported in the current literature (*Kinney et al., 2016*).

**Appendix 1—chemical structure 13.** Dimethoxyphenethyl amine (S13).

**2-(2,5-Dimethoxyphenyl)ethan-1-amine (S13)**. Following the general phthalimide deprotection protocol using phthalimide protected amine **S8** (500 mg, 1.61 mmol), primary amine **S13** was obtained as a colorless oil (250 mg, 86% yield). [1]H NMR (500 MHz, CDCl$_3$): δ 6.96–6.57 (m, 3H), 3.78 (s, 3H), 3.76 (s, 3H), 3.04–2.85 (m, 3H), 2.75 (t, J = 6.9 Hz, 2H), 1.90 (br s, NH$_2$); [13]C NMR (125 MHz, CDCl$_3$): δ 153.5, 152.0, 129.2, 117.0, 111.4, 111.3, 55.9, 55.7, 42.1, 34.6. The spectral characteristics were identical to those reported in the current literature (*Šuláková et al., 2019*).

**Appendix 1—chemical structure 14.** Trimethoxyphenethyl amine (S14).

**2-(2,3,4-Trimethoxyphenyl)ethan-1-amine (S14)**. Following the general phthalimide deprotection protocol using phthalimide protected amine **S9** (1.40 g, 4.10 mmol), primary amine **S14** was obtained as a colorless oil (600 mg, 69% yield). [1]H NMR (500 MHz, CDCl$_3$): δ 6.83 (d, J = 8.5 Hz, 1H), 6.61 (d, J = 8.5 Hz, 1H), 3.87 (s, 3H), 3.86 (s, 3H), 3.83 (s, 3H), 2.90 (t, J = 7.0 Hz, 2H), 2.70 (t, J = 7.0 Hz, 2H), 1.51 (br s, NH$_2$); [13]C NMR (125 MHz, CDCl$_3$): δ 152.4, 152.2, 142.5, 125.8, 124.5, 107.3, 61.1, 60.8, 56.1, 43.1, 34.2. The spectral characteristics were identical to those reported in the current literature (*Yang et al., 2013*).

**Appendix 1—chemical structure 15.** Trimethoxyphenethyl amine (S15).

**2-(2,4,6-Trimethoxyphenyl)ethan-1-amine (S15)**. Following the general phthalimide deprotection protocol using phthalimide protected amine **S10** (1.60 g, 4.69 mmol), primary amine **S15** was obtained as a colorless oil (693 mg, 70% yield). IR (thin film) ν 3426, 2938, 2838, 1593, 1498, 1455, 1417, 1204, 1148 cm$^{-1}$; [1]H NMR (500 MHz, CD$_3$OD): δ 6.20 (s, 2H), 3.79 (s, 6H), 3.79 (s, 3H), 2.79–2.71 (m, 4H), NH$_2$-group is not visible; [13]C NMR (125 MHz, CD$_3$OD): δ 161.5, 160.4, 108.1, 91.5, 56.0, 56.0, 55.7, 41.9, 25.8; ESI-HRMS calcd for C$_{11}$H$_{18}$NO$_3$ [M+H] 212.1281, found 212.1280.

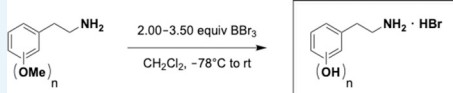

**Appendix 1—scheme 5.** Practical demethylation of aryl methyl ethers.

## General methyl ether cleavage protocol

BBr$_3$ (1.0 M in CH$_2$Cl$_2$; 1.15 equiv for each OMe group) was added dropwise to a solution of the 2-(methoxyphenyl)ethyl amine **S11–S15** (1.00 equiv) in CH$_2$Cl$_2$ (0.025 M) at –78°C. The resulting mixture was allowed to warm to rt over 3 hr. Stirring was continued for 14 hr. The resulting suspension was cooled to 0°C and quenched with the dropwise addition of MeOH (ca. 5 mL). Stirring at rt was continued for 1 hr. The resulting suspension was concentrated under reduced pressure to afford a pale-brown oil. The obtained residue was dissolved in a small amount of MeOH and again concentrated under reduced pressure; this step was repeated 3–4 times to remove all of the trimethyl borate side product and obtain analytically pure 2-(2-aminoethyl)benzenediol /-triol derivatives **S16–S20** as HBr salts.

**Appendix 1—chemical structure 16.** Aminoethylbenzene diol hydrobromide (S16).

**3-(2-Aminoethyl)benzene-1,2-diol hydrobromide** (**S16**). Following the general phenol ether cleavage protocol using phenol ether **S11** (200 mg, 1.10 mmol), catechol amine **S16** was obtained as a brown oil (250 mg, 97% yield). $^1$H NMR (500 MHz, CD$_3$OD): δ 6.73 (dd, $J$ = 6.7, 2.7 Hz, 1H), 6.69–6.56 (m, 2H), 3.17 (t, $J$ = 7.4 Hz, 2H), 2.95 (t, $J$ = 7.4 Hz, 2H), OH- and NH- protons are not visible; $^{13}$C NMR (125 MHz, CD$_3$OD): δ 146.0, 144.6, 124.4, 122.3, 120.8, 115.4, 40.8, 29.4. The spectral characteristics were identical to those reported in the current literature (*Short et al., 1973*).

**Appendix 1—chemical structure 17.** Aminoethylbenzene diol hydrobromide (S17).

**5-(2-Aminoethyl)benzene-1,3-diol hydrobromide** (**S17**). Following the general phenol ether cleavage protocol using phenol ether **S12** (20 mg, 0.110 mmol), resorcinol amine **S17** was obtained as a brown oil (25.0 mg, 97% yield). IR (thin film) ν 3358, 2928, 2853, 1771, 1597, 1495, 1418, 1091, 844 cm$^{-1}$; $^1$H NMR (500 MHz, CD$_3$OD): δ 6.33–6.08 (m, 2H), 3.13 (t, $J$ = 7.6 Hz, 2H), 2.80 (t, $J$ = 7.6 Hz, 2H), OH- and NH-protons are not visible; $^{13}$C NMR (125 MHz, CD$_3$OD): δ 160.0, 139.9, 108.1, 102.4, 41.9, 34.5; ESI-HRMS calcd for C$_8$H$_{12}$NO$_2$ [M+H] 154.0863, found 154.0860.

**Appendix 1—chemical structure 18.** 5-(2-Aminoethyl)benzene-1,4-diol hydrobromide (S18).

**5-(2-Aminoethyl)benzene-1,4-diol hydrobromide** (**S18**). Following the general phenol ether cleavage protocol using phenol ether **S13** (100 mg, 0.552 mmol), diol amine **S18** was obtained as a brown oil (118 mg, 91% yield). IR (thin film) ν 3352, 2927, 2858, 1621, 1505, 1455, 1344, 1202, 1152, 1212 cm$^{-1}$; $^1$H NMR (500 MHz, CD$_3$OD): δ 6.66 (d, $J$ = 8.5 Hz, 1H), 6.63–6.46 (m, 2H), 3.15 (t, $J$ = 7.2 Hz, 2H), 2.88 (t, $J$ = 7.2 Hz, 2H), OH- and NH-protons are

not visible; [13]C NMR (125 MHz, CD₃OD): δ 151.1, 149.4, 124.9, 118.2, 116.9, 115.8, 40.9, 29.9; ESI-HRMS calcd for $C_8H_{12}NO_2$ [M+H] 154.0863, found 154.0852.

**Appendix 1—chemical structure 19.** Aminoethylbenzene triol hydrobromide (**S19**).

**4-(2-Aminoethyl)benzene-1,2,3-triol hydrobromide (S19).** Following the general phenol ether cleavage protocol using phenol ether **S14** (200 mg, 0.948 mmol), triol amine **S19** was obtained as a brown oil (240 mg, quant.). IR (thin film) ν 3357, 3222, 2537, 1620, 1484, 1282, 1230, 1182, 1100, 1053, 1016 cm$^{-1}$; [1]H NMR (500 MHz, CD₃OD): δ 6.48 (d, *J* = 8.2 Hz, 1H), 6.32 (d, *J* = 8.2 Hz, 1H), 3.12 (t, *J* = 7.3 Hz, 2H), 2.87 (t, *J* = 7.3 Hz, 2H), OH- and NH-protons are not visible; [13]C NMR (125 MHz, CD₃OD): δ 146.6, 145.7, 134.4, 121.4, 115.9, 108.0, 41.2, 29.5; ESI-HRMS calcd for $C_8H_{12}NO_3$ [M+H] 170.0812, found 170.0805.

**Appendix 1—chemical structure 20.** Aminoethylbenzene triol hydrobromide (**S20**).

**2-(2-Aminoethyl)benzene-1,3,5-triol hydrobromide (S20).** Following the general phenol ether cleavage protocol using phenol ether **S15**, triol amine **S20** was obtained, according to MS identification, in low quantities along with brominated species and various methoxybenzene-diols in an inseparable mixture.

Initial attempts to alter reaction temperature or the number of equivalents of BBr₃ resulted in low conversion. Heating phenol ether **S15** in the presence of iodo(trimethyl)silane (**Anderson et al., 2005**) or sodium ethanethiolate (**Lutz et al., 1996**) afforded mono-deprotected material in a cleaner reaction, but the desired triol amine **S20** was not observed. Due to our inability to access this substrate, we did not evaluate this substrate in any enzyme reactions in our study.

## Benzaldehydes as starting materials

**Appendix 1—scheme 6.** Preparation of aminoethylbenzene triol hydrobromide from aryl nitriles.

**Appendix 1—chemical structure 21.** Trimethoxybenzonitrile (S21).

**2,3,5-Trimethoxybenzonitrile (S21).** K$_2$CO$_3$ (1.90 g, 13.8 mmol, 1.50 equiv) and dimethyl sulfate (0.960 mL, 1.28 g, 10.1 mmol, 1.10 equiv) were added to a solution of 5-hydroxy-2,3-dimethoxybenzonitrile (***Fürstner et al., 2002***) (1.65 g, 9.21 mmol, 1.00 equiv) in acetone (30 mL) at rt. Stirring was continued for 18 hr to afford a pale beige suspension. The solvent was removed under reduced pressure and the resulting crude material was diluted with a mixture of EtOAc–H$_2$O (1:1; 100 mL). The obtained layers were separated and the aqueous layer was extracted with EtOAc (3 × 20 mL). The combined organic layers were washed with 5% aqueous NaOH (20 mL) and brine (2 × 20 mL), dried over anhydrous Na$_2$SO$_4$, filtered, and concentrated under reduced pressure to afford analytically pure trimethoxybenzonitrile **S21** (1.78 g, quant.) as a pale beige solid. $^1$H NMR (500 MHz, CDCl$_3$): δ 6.68 (d, *J* = 2.8 Hz, 1H), 6.56 (d, *J* = 2.8 Hz, 1H), 3.94 (s, 3H), 3.86 (s, 3H), 3.79 (s, 3H). The spectral characteristics were identical to those reported in the current literature (***Rizzacasa and Sargent, 1988***).

**Appendix 1—chemical structure 22.** Trimethoxybenzaldehyde (S22).

**2,3,5-Trimethoxybenzaldehyde (S22)**. DIBAL-H (1.0 M in $CH_2Cl_2$; 12.4 mL, 12.4 mmol, 1.50 equiv) was added dropwise to a solution of nitrile **S21** (1.60 g, 8.28 mmol, 1.00 equiv) in $CH_2Cl_2$ (33 mL) at 0°C. The resulting mixture was allowed to warm to rt over 3 hr after which stirring was continued for 8 hr. The reaction was cooled to 0°C and HCl (1.0 M in $H_2O$; 10.0 mL) was added dropwise over 10 min. The mixture was allowed to warm to rt and stirring was continued for 2 hr. The layers were separated, and the aqueous layer was extracted with $CH_2Cl_2$ (2 × 15 mL). The combined organic layers were washed with $H_2O$ (2 × 15 mL) and brine (20 mL), dried over anhydrous $Na_2SO_4$, filtered, and concentrated under reduced pressure to afford analytically pure benzaldehyde **S22** (845 mg, 52%) as a beige solid. [1]H NMR (500 MHz, $CDCl_3$): δ 10.40 (s, 1H), 6.86 (d, $J$ = 2.9 Hz, 1H), 6.74 (d, $J$ = 2.9 Hz, 1H), 3.93 (s, 3H), 3.89 (s, 3H), 3.82 (s, 3H); [13]C NMR (125 MHz, $CDCl_3$): δ 190.2, 156.4, 154.4, 148.2, 129.9, 107.7, 99.7, 63.1, 56.4, 56.1. The spectral characteristics were identical to those reported in the current literature (**Singh et al., 1995**).

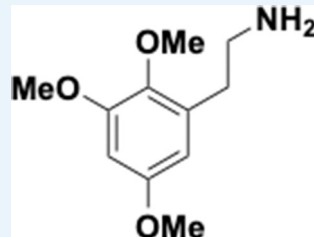

**Appendix 1—chemical structure 23.** Nitrovinyl benzene (S23).

**(E)−1,2,5-Trimethoxy-3-(2-nitrovinyl)benzene (S23)**. A mixture of benzaldehyde **S22** (845 mg, 4.31 mmol, 1.00 equiv) and ammonium acetate (500 mg, 6.46 mmol, 1.50 equiv) in nitromethane (40 mL) was heated to reflux for 18 hr, after which the reaction was found to be complete according to TLC ($R_f$ = 0.38 starting material; $R_f$ = 0.35 product; hexanes:EtOAc 4:1). The resulting mixture was concentrated under reduced pressure and purified by flash column chromatography (hexanes:EtOAc 6:1) to afford nitrovinyl benzene **S23** (750 mg, 73%) as a yellow solid. IR (thin film) ν 2959, 2846, 1717, 1633, 1601, 1492, 1465, 1332, 1282, 1206, 1176, 1151 $cm^{-1}$; [1]H NMR (500 MHz, $CDCl_3$): δ 8.16 (d, $J$ = 13.8 Hz, 1H), 7.71 (d, $J$ = 13.8 Hz, 1H), 6.61 (d, $J$ = 2.6 Hz, 1H), 6.48 (d, $J$ = 2.6 Hz, 1H), 3.86 (s, 3H), 3.83 (s, 3H), 3.80 (s, 3H); [13]C NMR (125 MHz, $CDCl_3$): δ 156.4, 154.1, 144.2, 138.7, 134.8, 124.0, 104.5, 102.9, 61.6, 56.1, 55.9; $R_f$ = 0.35 (hexanes:EtOAc 4:1); ESI-HRMS calcd for $C_{11}H_{14}NO_5$ [M+H] 240.0866, found 240.0871.

**Appendix 1—chemical structure 24.** Trimethoxyphenethyl amine (S24).

**2-(2,3,5-Trimethoxyphenyl)ethan-1-amine (S24)**. LiAlH$_4$ (2.0 M in THF; 1.83 mL, 3.66 mmol, 3.50 equiv) was added dropwise over 10 min to a solution of nitrovinyl benzene **S23** (250 mg, 1.05 mmol, 1.00 equiv) in THF (6 mL) at 0°C. The resulting mixture was allowed to warm to rt and stirring was continued for 24 hr. The reaction mixture was then cooled to 0°C and 10% aqueous NaOH (5.0 mL) was added dropwise over 10 min, resulting in an exothermic reaction. Stirring was continued for 1 hr, and the resulting suspension was diluted with EtOAc (20 mL) and filtered over a plug of Celite (EtOAc rinse). The filtrate was dried over anhydrous $Na_2SO_4$, filtered, and concentrated under reduced pressure to afford crude amine **S24**. Purification by flash column chromatography (EtOAc:MeOH 85:15 + 0.1% $Et_3N$) afforded

amine **S24** (150 mg, 68%) as a pale yellow oil. IR (thin film) ν 3363, 2937, 2839, 1599, 1492, 1465, 1427, 1380, 1220, 1175, 1150, 1089, 830 cm$^{-1}$; $^1$H NMR (500 MHz, CDCl$_3$): δ 6.39 (d, $J$ = 2.5 Hz, 1H), 6.29 (d, $J$ = 2.5 Hz, 1H), 3.84 (s, 3H), 3.77 (s, 3H), 3.76 (s, 3H), 2.97 (t, $J$ = 7.0 Hz, 2H), 2.78 (t, $J$ = 7.0 Hz, 2H), 2.27 (br s, NH$_2$); $^{13}$C NMR (125 MHz, CDCl$_3$): δ 156.0, 153.5, 141.5, 133.4, 105.5, 98.5, 60.9, 55.7, 55.6, 42.7, 34.1; R$_f$ = 0.08 (EtOAc:MeOH 85:15); ESI-HRMS calcd for C$_{11}$H$_{18}$NO$_3$ [M+H] 212.1281, found 212.1275.

**Appendix 1—chemical structure 25.** Aminoethylbenzene triol hydrobromide (S25).

3-(2-Aminoethyl)benzene-1,2,5-triol hydrobromide (**S25**). BBr$_3$ (1.0 M in CH$_2$Cl$_2$; 1.41 mL, 1.41 mmol, 3.30 equiv) was added dropwise over 10 min to a solution of phenol ether **S24** (90.0 mg, 0.425 mmol, 1.00 equiv) in CH$_2$Cl$_2$ (0.033 M) at –78°C. The resulting mixture was allowed to warm to rt over 3 hr. Stirring was continued for 18 hr. The resulting suspension was cooled to 0°C and quenched with the dropwise addition of MeOH (ca. 5 mL). Stirring at rt was continued for 1 hr. The resulting solution was concentrated under reduced pressure to afford a pale-brown oil. The obtained residue was dissolved in a small amount of MeOH and again concentrated under reduced pressure; this step was repeated 3–4 times to remove all of the trimethyl borate side product and obtain analytically pure triol amine **S25** as the HBr salt. IR (thin film) ν 3358, 3223, 1604, 1452, 1359, 1291, 1108, 1044 cm$^{-1}$; $^1$H NMR (500 MHz, CD$_3$OD): δ 6.29 (d, $J$ = 2.6 Hz, 1H), 6.11 (d, $J$ = 2.8 Hz, 1H), 3.15 (t, $J$ = 7.4 Hz, 2H), 2.88 (t, $J$ = 7.3 Hz, 2H), OH- and NH-protons are not visible; $^{13}$C NMR (125 MHz, CD$_3$OD): δ 151.4, 147.1, 137.7, 125.0, 108.2, 103.3, 41.0, 29.8; ESI-HRMS calcd for C$_8$H$_{12}$NO$_3$ [M+H] 170.0812, found 170.0812.

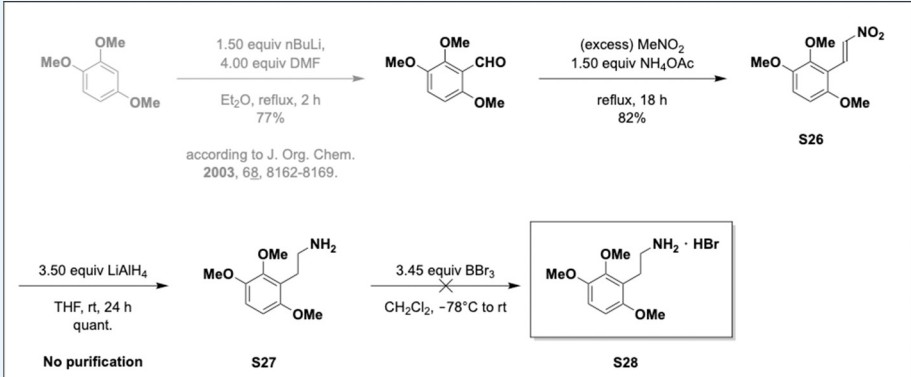

**Appendix 1—scheme 7.** Preparation of aminoethylbenzene triol hydrobromide derivatives from trimethoxybenzene.

**Appendix 1—chemical structure 26.** Trimethoxy nitrovinyl benzene (S26).

(*E*)−1,2,4-Trimethoxy-3-(2-nitrovinyl)benzene (**S26**). A mixture of 2,3,6-trimethoxybenzaldehyde[15] (500 mg, 2.55 mmol, 1.00 equiv) and ammonium acetate (295 mg, 3.82 mmol, 1.50 equiv) in nitromethane (23 mL) was heated to reflux for 18 hr after which the reaction was found to be according to TLC ($R_f$ = 0.30 starting material; $R_f$ = 0.38 product; hexanes:EtOAc 2:1). The resulting mixture was concentrated under reduced pressure and purified by flash column chromatography (hexanes:EtOAc 5:1) to afford nitrovinyl benzene **S26** (500 mg, 82%) as a yellow solid. IR (thin film) ν 2939, 2854, 1625, 1583, 1507, 1496, 1330, 1284, 1116, 1009 cm$^{-1}$; $^1$H NMR (500 MHz, CD$_3$OD): δ 8.39 (d, *J* = 13.7 Hz, 1H), 8.09 (d, *J* = 13.7 Hz, 1H), 7.17 (d, *J* = 9.2 Hz, 1H), 6.80 (d, *J* = 9.2 Hz, 1H), 3.91 (s, 3H), 3.91 (s, 3H), 3.85 (s, 3H); $^{13}$C NMR (125 MHz, CD$_3$OD): δ 155.5, 151.6, 148.2, 140.8, 130.7, 118.7, 114.7, 107.1, 61.6, 57.0, 56.6; $R_f$ = 0.38 (hexanes:EtOAc 2:1); ESI-HRMS calcd for C$_{11}$H$_{14}$NO$_5$ [M+H] 240.0866, found 240.0860.

**Appendix 1—chemical structure 27.** Trimethoxyphenethyl amine (**S27**).

**2-(2,3,6-Trimethoxyphenyl)ethan-1-amine (S27).** LiAlH$_4$ (2.0 M in THF; 3.66 mL, 7.32 mmol, 3.50 equiv) was added dropwise over 10 min to a solution of nitrovinyl benzene **S26** (200 mg, 2.09 mmol, 1.00 equiv) in THF (12 mL) at 0°C. The resulting mixture was allowed to warm to rt and stirring was continued for 24 hr. The reaction mixture was cooled to 0°C and 10% aqueous NaOH (10.0 mL) was added dropwise over 10 min, resulting in an exothermic reaction. Stirring was stirred continued for 1 hr. The resulting suspension was diluted with EtOAc (20 mL) and filtered over a plug of Celite (EtOAc rinse). The filtrate was dried over anhydrous Na$_2$SO$_4$, filtered, and concentrated under reduced pressure to afford analytically pure amine **S27** (438 mg, quant.) as a pale yellow oil. IR (thin film) ν 3363, 2936, 2833, 1648, 1485, 1463, 1253, 1085, 793, 627 cm$^{-1}$; $^1$H NMR (500 MHz, CDCl$_3$): δ 6.73 (d, *J* = 8.8 Hz, 1H), 6.55 (d, *J* = 8.8 Hz, 1H), 3.82 (s, 3H), 3.82 (s, 3H), 3.76 (s, 3H), 2.88 (d, *J* = 6.0 Hz, 2H), 2.83 (d, *J* = 6.0 Hz, 2H), 2.47 (br s, NH$_2$); $^{13}$C NMR (125 MHz, CDCl$_3$): δ 152.5, 148.4, 147.2, 122.7, 110.2, 105.4, 60.8, 56.2, 55.9, 42.2, 28.5; $R_f$ = 0.10 (EtOAc:MeOH 85:15); ESI-HRMS calcd for C$_{11}$H$_{18}$NO$_3$ [M+H] 212.1281, found 212.1273.

**Appendix 1—chemical structure 28.** Aminoethylbenzene triol hydrobromide (**S28**).

**3-(2-Aminoethyl)benzene-1,2,4-triol hydrobromide (S28).** Following the general phenol ether cleavage protocol described for the preparation of amine hydrobromide **S25** using phenol ether **S27** as starting material, triol amine **S28** was obtained, according to MS identification, in low quantities along with brominated species. Initial attempts in changing the reaction temperature or the number of equivalents of BBr$_3$ resulted in low conversion and the desired product could not be isolated in pure form. Therefore it was not used in any enzyme assays.

## Preparation of 2-amino-4-(2-aminoethyl)phenol
All reactions were carried out with degassed solvents under a positive pressure of nitrogen.

**Appendix 1—scheme 8.** 2-Amino-4-(2-aminoethyl)phenol via Mitsunobu strategy.

**Appendix 1—chemical structure 29.** Benzyloxy nitrophenethyl alcohol (S29).

**2-(4-(Benzyloxy)−3-nitrophenyl)ethan-1-ol (S29).** (Bromomethyl)benzene (3.02 mL, 4.34 g, 25.4 mmol, 2.50 equiv) was added dropwise to a suspension of commercially available 2-(4-hydroxy-3-nitrophenyl)acetic acid (2.00 g, 10.1 mmol, 1.00 equiv), anhydrous potassium carbonate (4.21 g, 30.4 mmol, 3.00 equiv), and anhydrous potassium iodide (674 mg, 4.06 mmol, 0.400 equiv) in acetone (34 mL) at rt. Vigorously stirring was continued for 48 hr. The resulting suspension was diluted with a mixture of EtOAc-$H_2O$ (1:1; 100 mL), cooled to 0°C, and adjusted to pH = 1 using aqueous 1M HCl. This resulted in an exothermic reaction. The layers were separated, and the aqueous layer was extracted with EtOAc (3 × 15 mL). The combined organic layers were washed with $H_2O$ (2 × 15 mL), brine (30 mL), dried over anhydrous $Na_2SO_4$, filtered, and concentrated under reduced pressure to afford crude 2-(4-(benzyloxy)−3-nitrophenyl)acetic acid, which was immediately used in the next step without further purification.

BH$_3$ • SMe$_2$ (2.0 M in THF; 6.57 mL, 13.1 mmol, 1.30 equiv) was added dropwise to a solution of the crude 2-(4-(benzyloxy)−3-nitrophenyl)acetic acid in THF (110 mL) at 0°C. The resulting mixture was allowed to warm to rt over 3 hr and stirring was continued for 14 hr. The resulting suspension was cooled to 0°C and carefully quenched with the dropwise addition of saturated aqueous NaHCO$_3$. The layers were separated, and the aqueous layer was extracted with EtOAc (3 × 50 mL). The combined organic layers were washed with $H_2O$ (2 × 20 mL), brine (2 × 20 mL), dried over anhydrous $Na_2SO_4$, filtered, and concentrated under reduced pressure to afford crude alcohol **S29** as a brown oil. Purification by flash column chromatography (hexanes:EtOAc 1:1) afforded analytically pure alcohol (1.98 g, 72%) as a pale yellow oil. $^1$H NMR (500 MHz, CDCl$_3$): δ 7.71 (d, $J$ = 2.3 Hz, 1H), 7.46–7.29 (m, 6H), 7.05 (d, $J$ = 8.6 Hz, 1H), 5.19 (s, 2H), 3.82 (t, $J$ = 6.5 Hz, 2H), 2.81 (t, $J$ = 6.5 Hz, 2H), 1.90 (br s, OH); $^{13}$C NMR (125 MHz, CDCl$_3$): δ 150.5, 140.0, 135.8, 134.8, 131.8, 128.7, 128.2, 127.0, 125.9, 115.4, 71.3, 63.0, 37.7; R$_f$ = 0.18 (hexanes:EtOAc 1:1). The spectral characteristics were identical to those reported in the current literature (**Lin et al., 2005**).

**Appendix 1—chemical structure 30.** Benzyloxy nitrophenethyl phthalimide (S30).

**2-(4-(Benzyloxy)−3-nitrophenethyl)isoindoline-1,3-dione (S30).** Diethyl azodicarboxylate (DEAD; 40% in toluene; 3.95 mL, 8.05 mmol, 1.10 equiv) was added dropwise over 10 min to a solution of $PPh_3$ (2.21 g, 8.42 mmol, 1.15 equiv), phthalimide (1.24 g, 8.42 mmol, 1.15 equiv) and the alcohol **S29** (1.98 g, 7.32 mmol, 1.00 equiv) in THF (50 mL) at 0°C. The resulting mixture was allowed to warm to rt over 3 hr and stirring was continued for 14 hr. The resulting pale-yellow solution was concentrated under reduced pressure and purified by flash column chromatography (hexanes:EtOAc 4:1) to afford the desired phthalimide protected amine **S30** (2.28 g, 78%) as a colorless solid. IR (thin film) ν 2985, 2871, 2783, 1774, 1750, 1640, 1387, 1307, 717 cm$^{-1}$; $^1$H NMR (500 MHz, CDCl$_3$): δ 7.94–7.80 (m, 2H), 7.78–7.67 (m, 3H), 7.50–7.30 (m, 6H), 7.05 (d, $J$ = 8.6 Hz, 1H), 5.20 (s, 2H), 3.91 (t, $J$ = 7.5 Hz, 2H), 2.99 (t, $J$ = 7.5 Hz, 2H); $^{13}$C NMR (125 MHz, CDCl$_3$): δ 168.2, 150.8, 140.1, 135.7, 134.5, 134.2, 132.0, 130.9, 128.8, 128.3, 127.1, 126.0, 123.5, 115.6, 71.3, 38.8, 33.4; R$_f$ = 0.18 (hexanes:EtOAc 1:1); ESI-HRMS calcd for C$_{23}$H$_{19}$N$_2$O$_5$ [M+H] 403.1288, found 403.1266 and C$_{22}$H$_{19}$N$_2$O$_5$Na [M+Na] 425.1113, found 425.1111.

**Appendix 1—chemical structure 31.** Amino alcohol (S31).

**2-(3-Amino-4-hydroxyphenethyl)isoindoline-1,3-dione (S31).** A flame dried round-bottomed flask was charged with nitroarene **S30** (100 mg, 0.250 mmol, 1.00 equiv) in a mixture of EtOH-CH$_2$Cl$_2$ (1:1; 18 mL) at rt. Pd-C (10% on activated charcoal; 15 mg) was added to the clear solution, which was purged with H$_2$ for 15 min with the H$_2$ inlet needle below the solvent surface. The H$_2$ inlet needle was raised above the solvent surface and stirring was continued for 18 hr. The resulting black suspension was filtered over a short plug of Celite (CH$_2$Cl$_2$ rinse). The filtrate was concentrated under reduced pressure to afford a brown oil that was purified by flash column chromatography (hexanes:EtOAc 1:1) to afford the desired amino alcohol **S31** (70 mg, quant.) as a yellow solid. IR (thin film) ν 3373, 2941, 2824, 1410, 1022, 1005, 822, 760, 617 cm$^{-1}$; $^1$H NMR (500 MHz, DMSO-d$_6$): δ 8.77 (br s, OH), 7.96–7.72 (m, 4H), 6.50 (d, $J$ = 7.9 Hz, 1H), 6.46 (d, $J$ = 2.0 Hz, 1H), 6.20 (dd, $J$ = 7.9, 2.1 Hz, 1H), 4.45 (br s, NH$_2$), 3.69 (t, $J$ = 7.6, 2H), 2.67 (t, $J$ = 7.6, 2H); $^{13}$C NMR (125 MHz, DMSO-d$_6$): δ 167.7, 142.5, 136.5, 134.4, 131.6, 128.9, 123.0, 116.3, 114.6, 114.3, 39.5, 33.4; R$_f$ = 0.20 (hexanes:EtOAc 1:1); ESI-HRMS calcd for C$_{16}$H$_{15}$N$_2$O$_3$ [M+H] 283.1077 found 283.1066 and C$_{16}$H$_{14}$N$_2$O$_3$Na [M+Na] 305.0902, found 305.0880.

**Appendix 1—chemical structure 32.** 2-Amino-4-(2-aminoethyl)phenol (S32).

**2-Amino-4-(2-aminoethyl)phenol (S32).** Hydrazine monohydrate (120 µL, 2.48 mmol, 10.0 equiv) was added to a suspension of the phthalimide protected amine **S31** (70.0 mg, 0.248 mmol, 1.00 equiv) in EtOH (1.5 mL). The resulting solution was heated to reflux for 1.5 hr while colorless solids crashed out. The resulting suspension was allowed to cool to rt before H$_2$O (10 mL) was added in one portion. Stirring was continued to afford a clear solution that was extracted with EtOAc (3 × 5 mL). The combined organic layers were washed with H$_2$O (3 × 5 mL), brine (3 × 5 mL), dried over anhydrous Na$_2$SO$_4$, filtered and concentrated under reduced pressure to yield analytically pure amine **S32** (18 mg, 48%). IR (thin film) ν 3384, 2947, 2822, 1580, 1239, 1049, 1021, 837 cm$^{-1}$; $^1$H NMR (500 MHz, CD$_3$OD): δ 6.66 (d, $J$ = 8.0 Hz, 1H), 6.63 (d, $J$ = 2.2 Hz, 1H), 6.47 (dd, $J$ = 8.0, 2.2 Hz, 1H), 3.05 (t, $J$ = 7.5 Hz, 2H), 2.75 (t, $J$ = 7.5 Hz, 2H), OH- and NH-protons are not visible; $^{13}$C NMR (125 MHz, CD$_3$OD): δ 145.2, 136.4, 131.2,

120.2, 117.7, 115.7, 43.6, 37.7; ESI-HRMS calcd for $C_8H_{13}N_2O$ [M+H] 153.1028 found 153.1019.

# Preparation of hydroxytyrosol

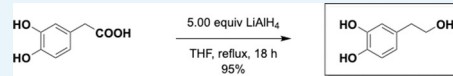

**Appendix 1—scheme 9.** Reduction of a dihydroxyphenyl acetic acid.

4-(2-Hydroxyethyl)benzene-1,2-diol(S33). $LiAlH_4$ (340 mg, 8.92 mmol, 5.00 equiv) was added in small portions to a solution of commercially available 3,4-dihydroxyphenylacetic acid (300 mg, 1.78 mmol, 1.00 equiv) in THF (35 mL) at 0°C. The suspension was allowed to warm to rt over 30 min before being heated to reflux for 18 hr. The resulting mixture was cooled to 0°C and quenched with the slow addition of aqueous 0.5 M HCl (30 mL). The layers were separated, and the aqueous layer was extracted with EtOAc (3 × 10 mL). The combined organic layers were washed with $H_2O$ (2 × 10 mL), brine (2 × 10 mL), dried over anhydrous $Na_2SO_4$, filtered, and concentrated under reduced pressure to afford an orange oil. Purification by flash column chromatography (hexanes:EtOAc 1:1) afforded triol **S33** (260 mg, 95%) as a pale red oil. [1]H NMR (500 MHz, $CD_3OD$): δ 6.74–6.60 (m, 2H), 6.58–6.48 (m, 1H), 3.67 (t, J = 7.2 Hz, 2H), 2.66 (t, J = 7.2 Hz, 2H), OH-protons are not visible; $R_f$ = 0.20 (hexanes: EtOAc 1:1). The spectral characteristics were identical to those reported in the current literature (*Napora-Wijata et al., 2014*).

## Additional information

- 2-Aminoethylbenzenediol /- triol derivatives (S16–S20, S25) as well as 2-amino-4-(2-amino-ethyl)phenol S32 are sensitive towards oxidation and turn black within hours if stored in the presence of $O_2$. No noticeable change in their composition is observed, according to [1]H NMR, when stored in the absence of $O_2$ at 3–4°C.
- Salt formation of the oxygen sensitive alkylamines (S16–S20, S25, S32) leads to more stable compounds as no decomposition was observed when stored in the presence of $O_2$ for several days.
- Upon quenching of the phenol ether cleavage reaction with MeOH, a reaction that contains $BBr_3$, volatile $B(OMe)_3$ is formed as sole side product. This can be removed under reduced pressure to afford pure products. It is important to stir the reaction mixture for approximately 1 hr upon the addition of MeOH to allow for the full conversion of $BBr_3$ to $B(OMe)_3$. The obtained residue can be dissolved in additional MeOH and again concentrated under reduced pressure to ensure the complete removal of $B(OMe)_3$.

## [1]H / [13]C SPECTRA

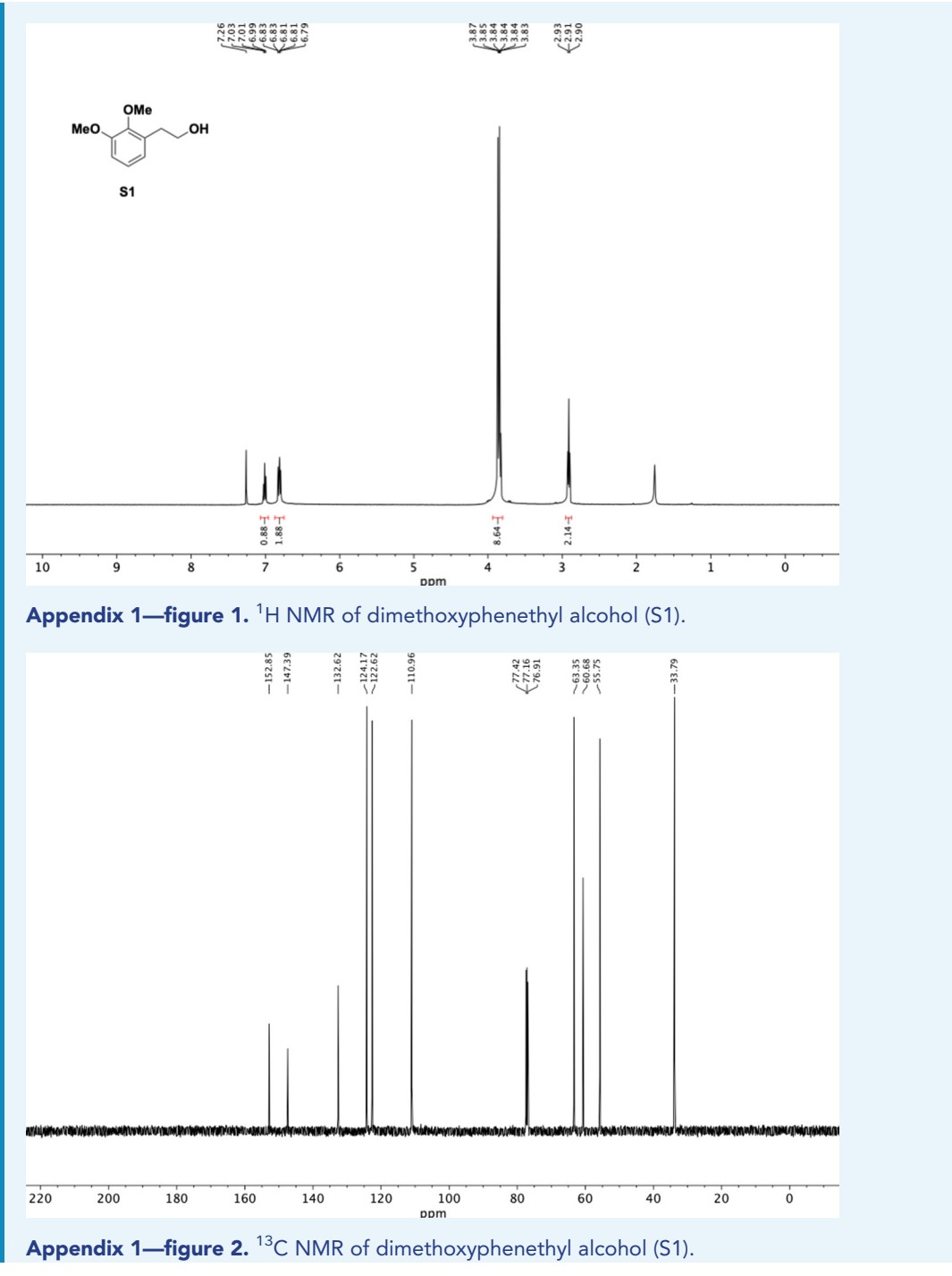

**Appendix 1—figure 1.** $^1$H NMR of dimethoxyphenethyl alcohol (S1).

**Appendix 1—figure 2.** $^{13}$C NMR of dimethoxyphenethyl alcohol (S1).

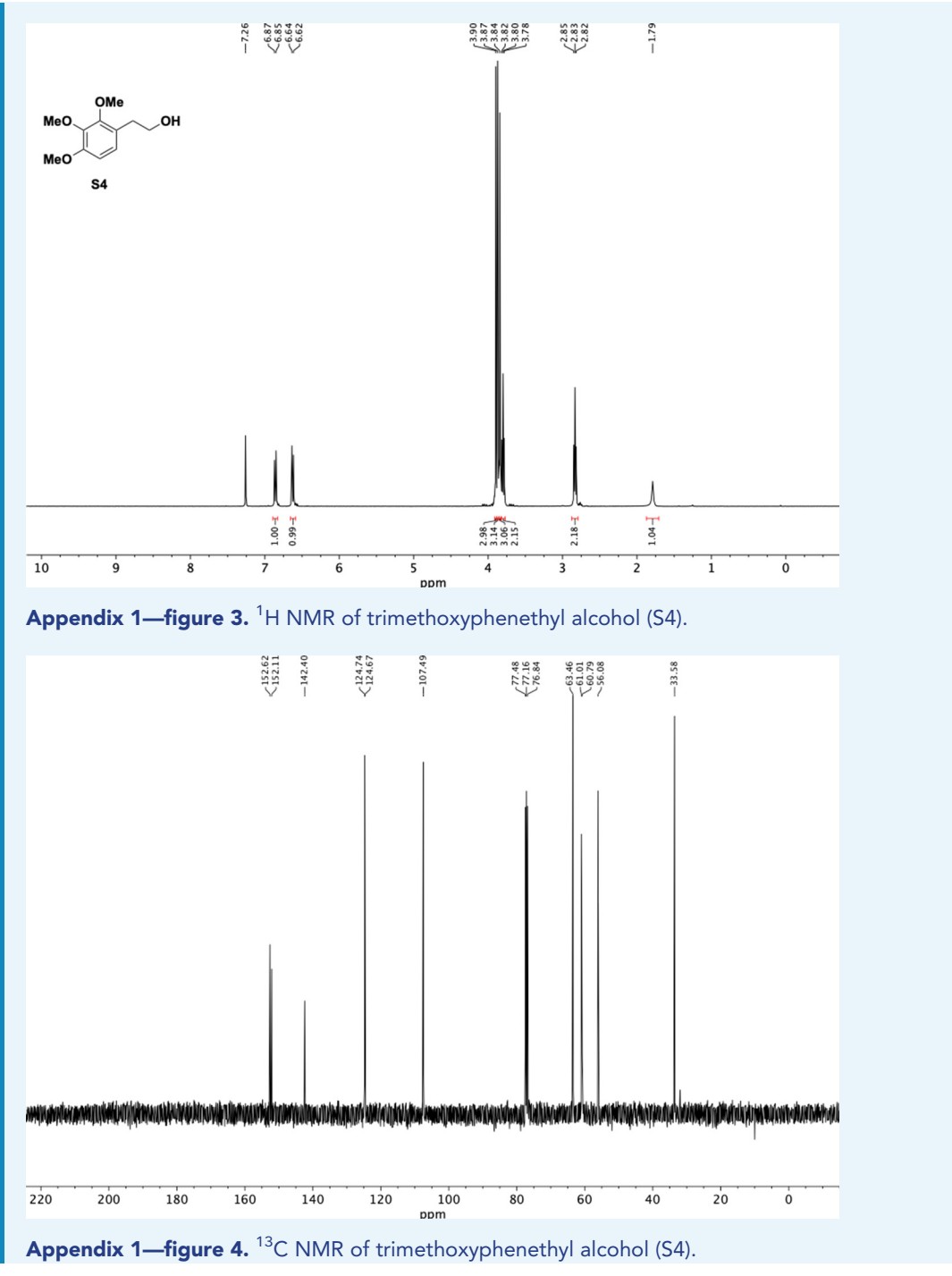

**Appendix 1—figure 3.** $^1$H NMR of trimethoxyphenethyl alcohol (S4).

**Appendix 1—figure 4.** $^{13}$C NMR of trimethoxyphenethyl alcohol (S4).

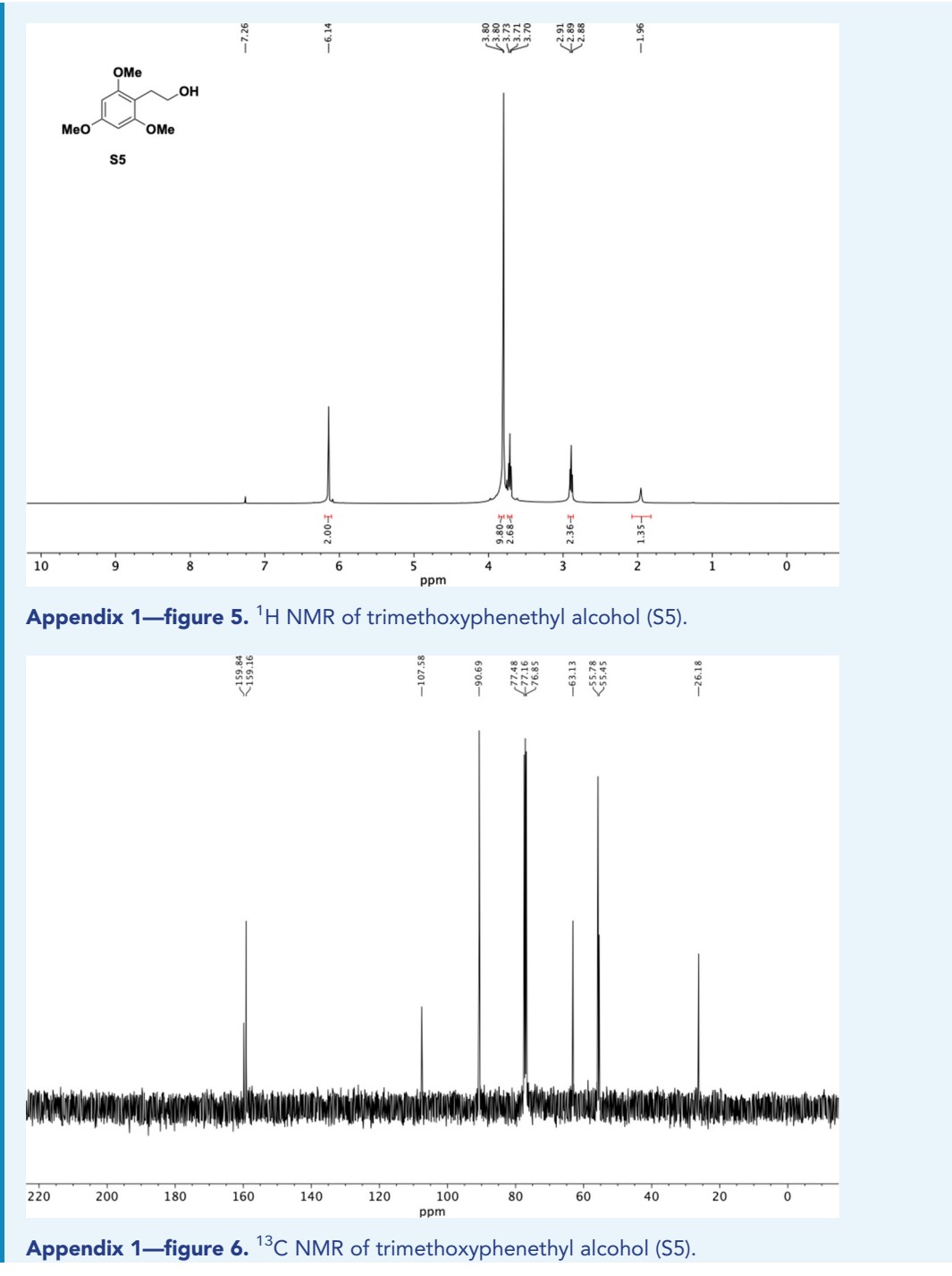

**Appendix 1—figure 5.** [1]H NMR of trimethoxyphenethyl alcohol (S5).

**Appendix 1—figure 6.** [13]C NMR of trimethoxyphenethyl alcohol (S5).

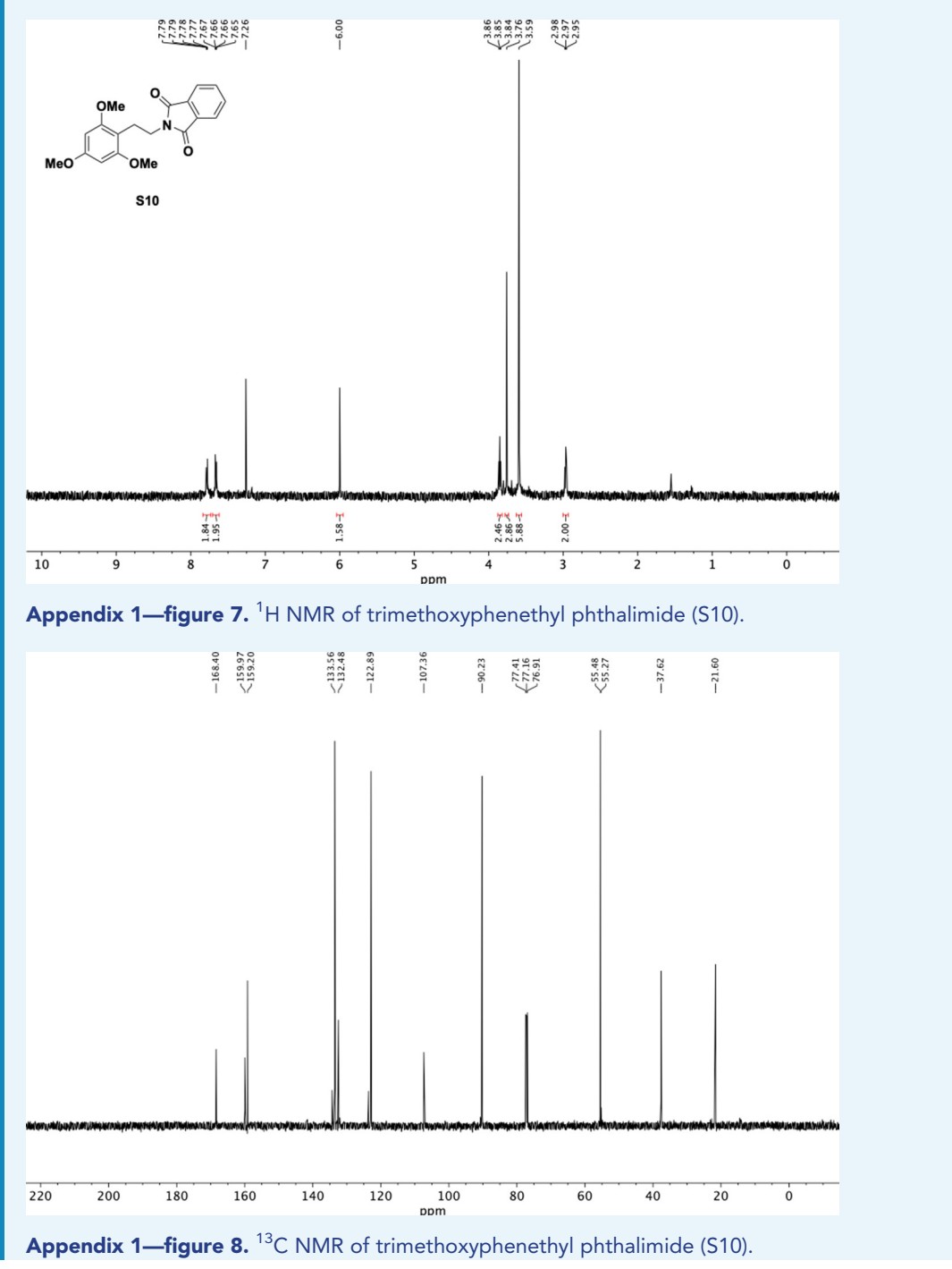

**Appendix 1—figure 7.** [1]H NMR of trimethoxyphenethyl phthalimide (S10).

**Appendix 1—figure 8.** [13]C NMR of trimethoxyphenethyl phthalimide (S10).

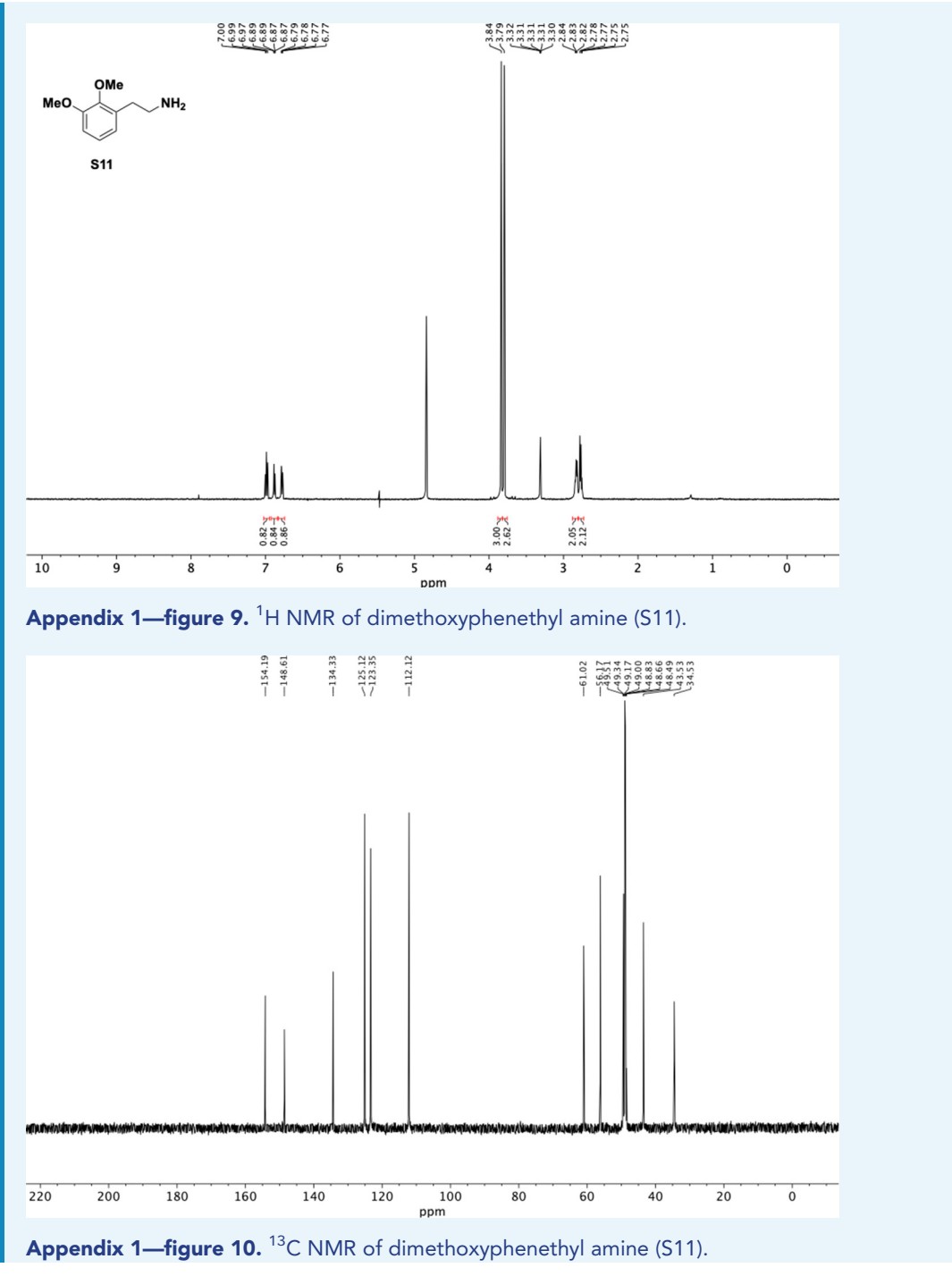

**Appendix 1—figure 9.** [1]H NMR of dimethoxyphenethyl amine (S11).

**Appendix 1—figure 10.** [13]C NMR of dimethoxyphenethyl amine (S11).

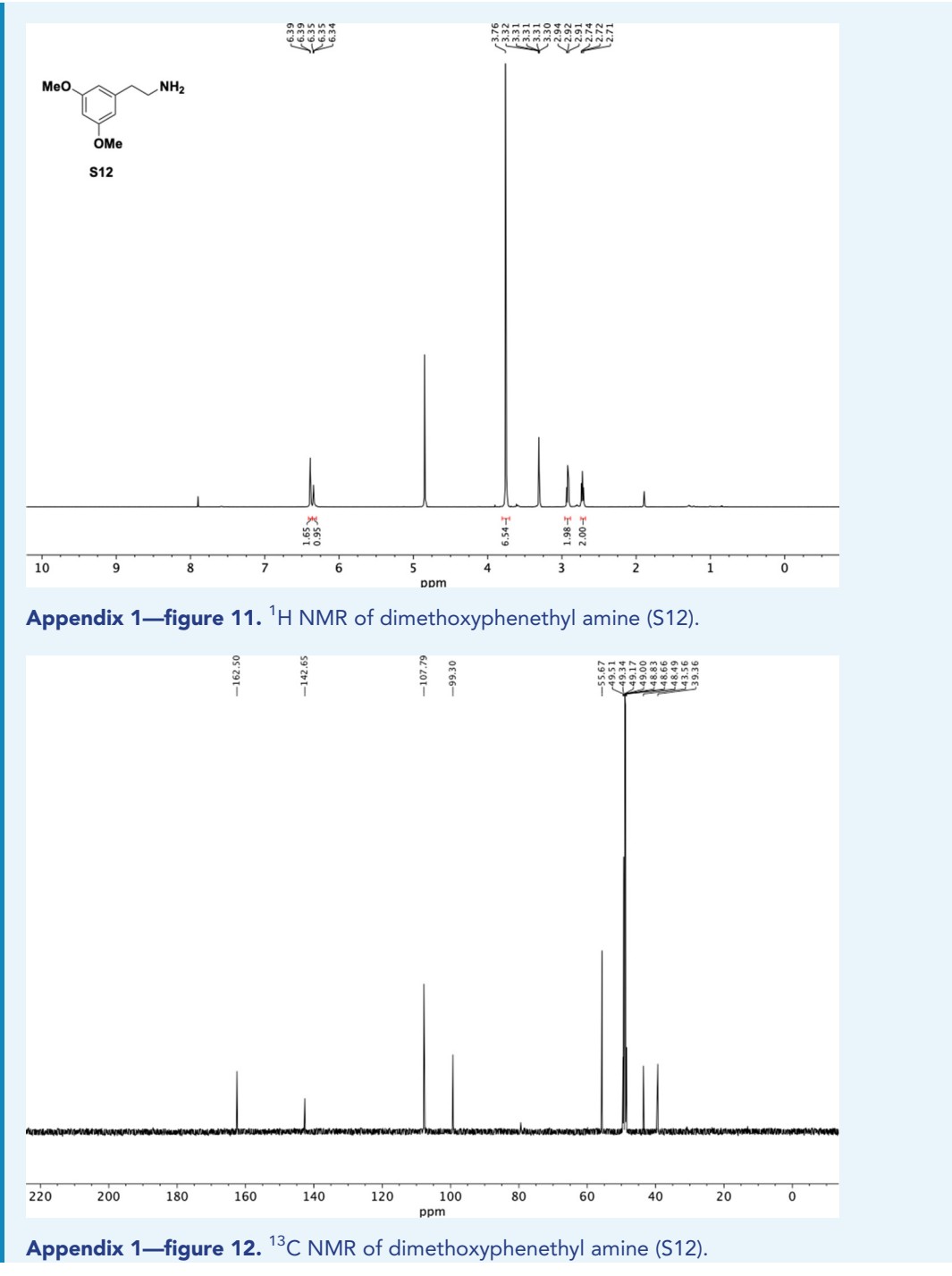

**Appendix 1—figure 11.** [1]H NMR of dimethoxyphenethyl amine (S12).

**Appendix 1—figure 12.** [13]C NMR of dimethoxyphenethyl amine (S12).

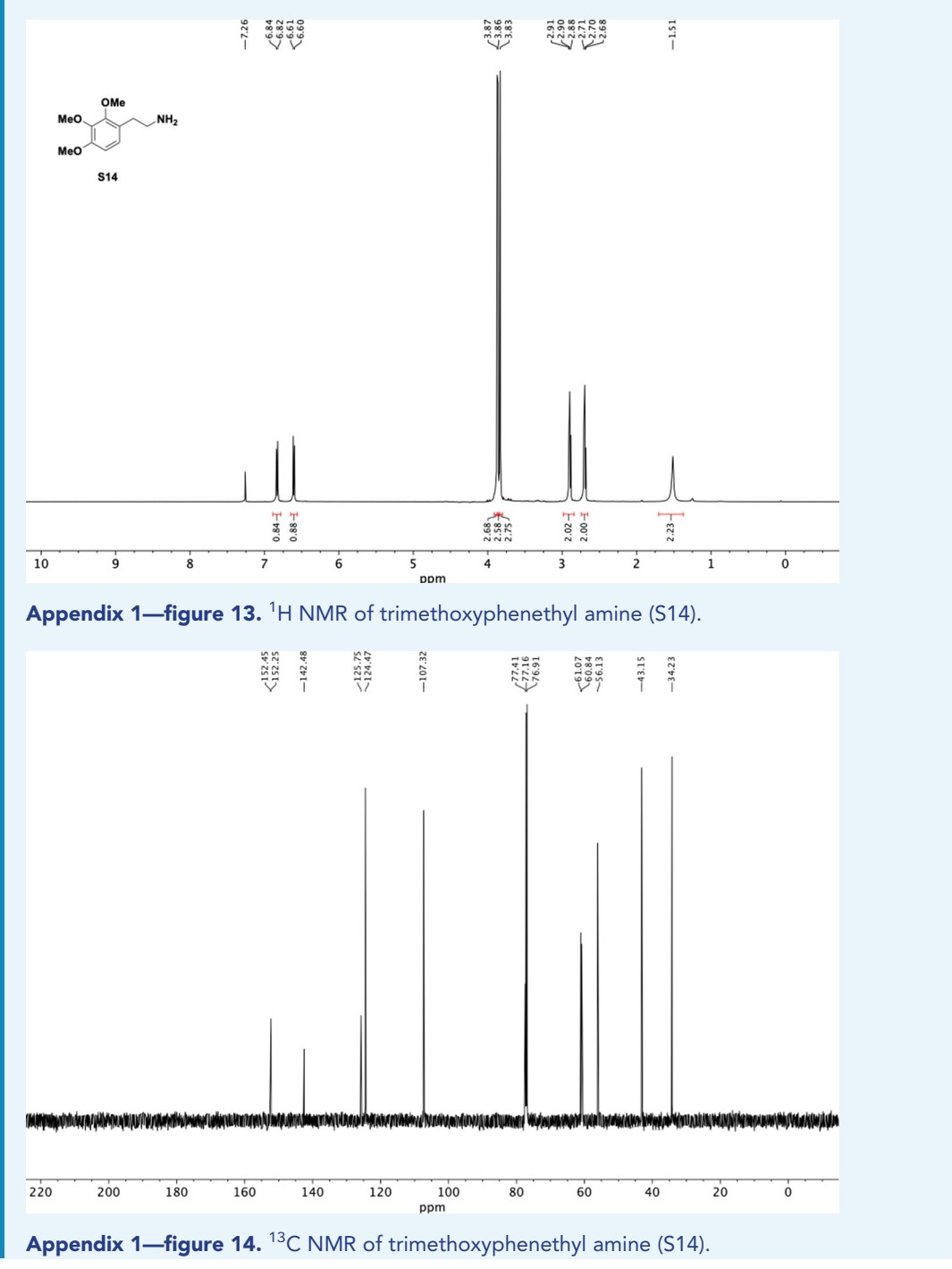

**Appendix 1—figure 13.** [1]H NMR of trimethoxyphenethyl amine (S14).

**Appendix 1—figure 14.** [13]C NMR of trimethoxyphenethyl amine (S14).

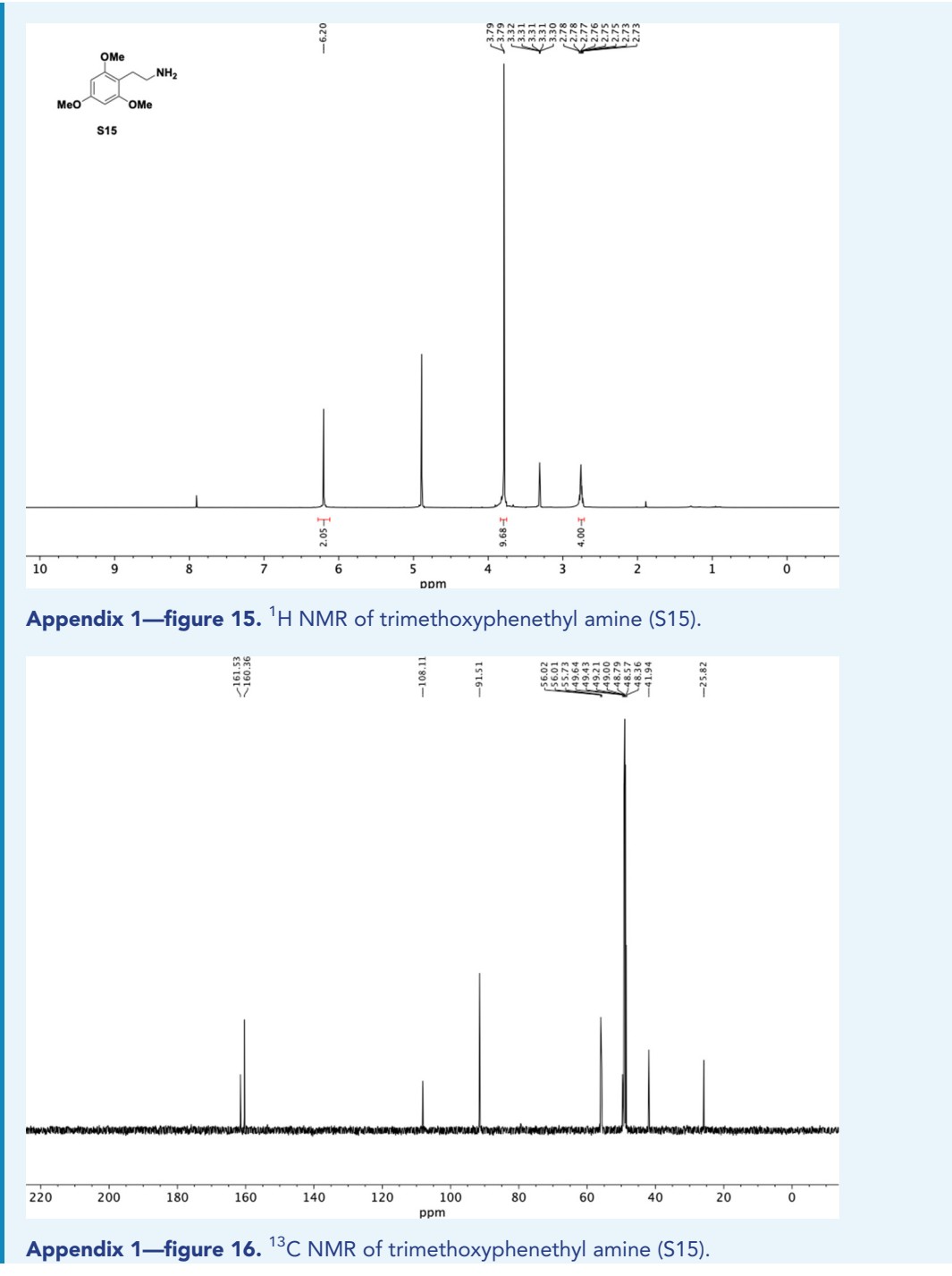

**Appendix 1—figure 15.** $^1$H NMR of trimethoxyphenethyl amine (S15).

**Appendix 1—figure 16.** $^{13}$C NMR of trimethoxyphenethyl amine (S15).

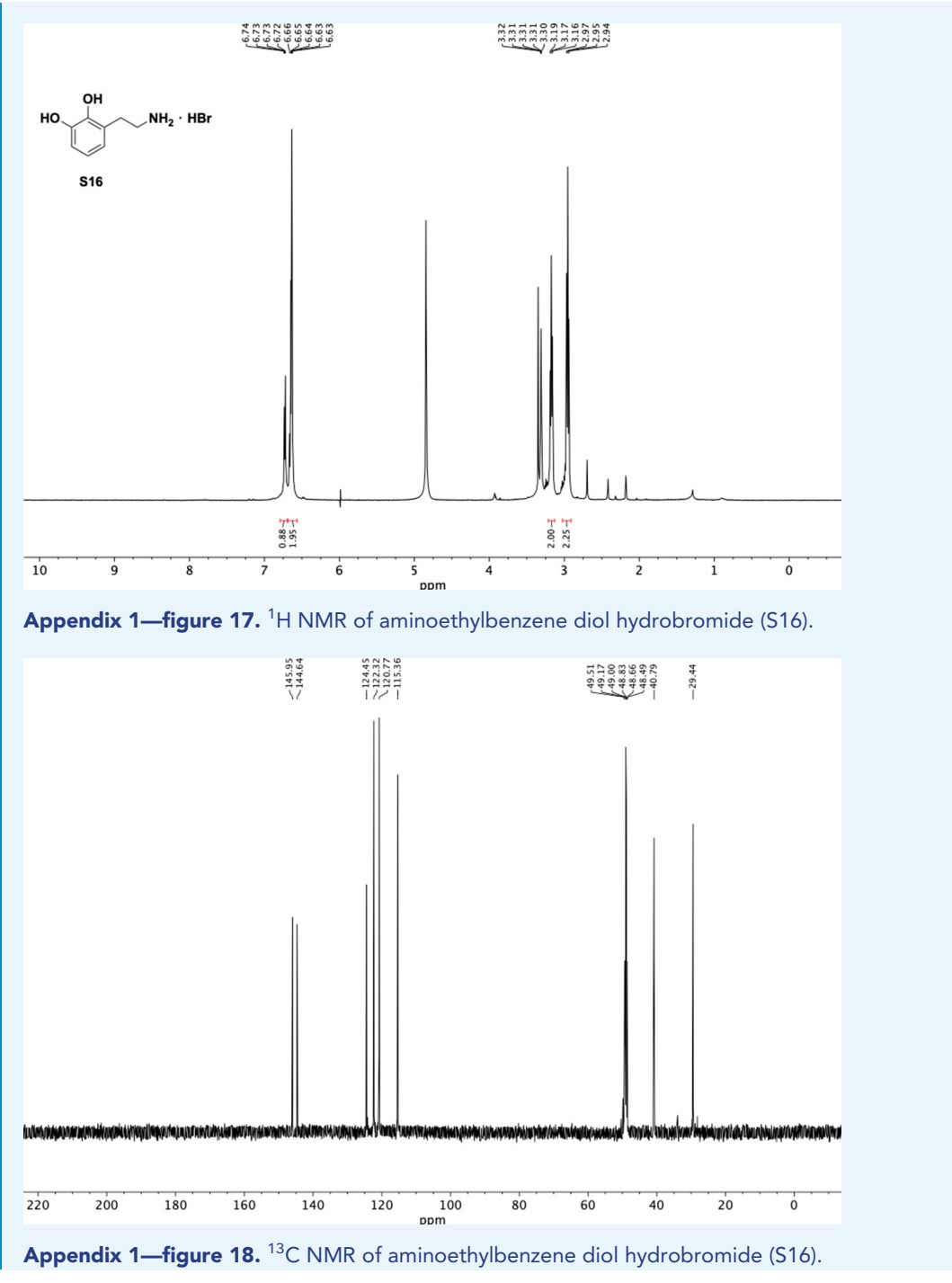

**Appendix 1—figure 17.** $^1$H NMR of aminoethylbenzene diol hydrobromide (S16).

**Appendix 1—figure 18.** $^{13}$C NMR of aminoethylbenzene diol hydrobromide (S16).

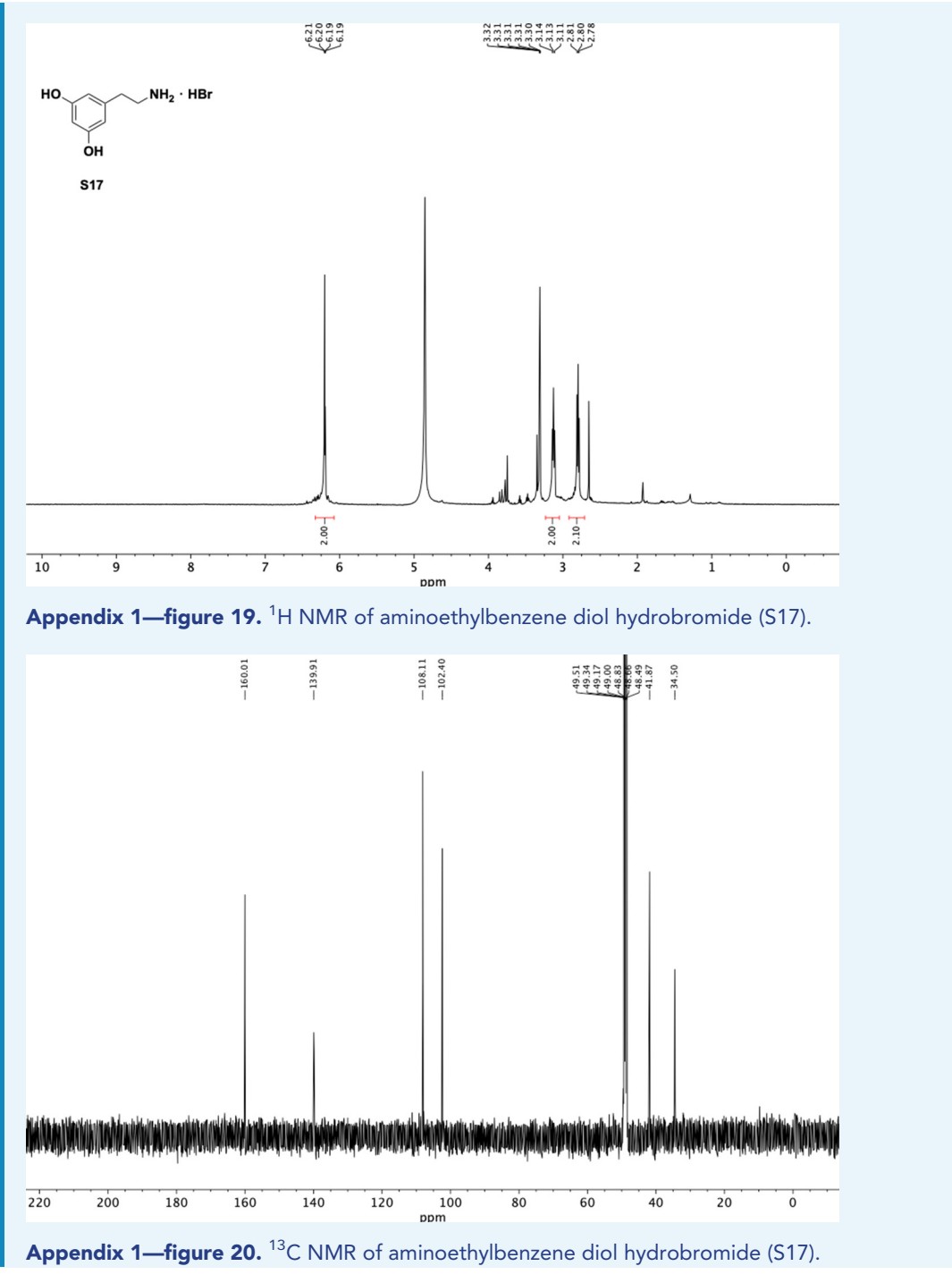

**Appendix 1—figure 19.** $^1$H NMR of aminoethylbenzene diol hydrobromide (S17).

**Appendix 1—figure 20.** $^{13}$C NMR of aminoethylbenzene diol hydrobromide (S17).

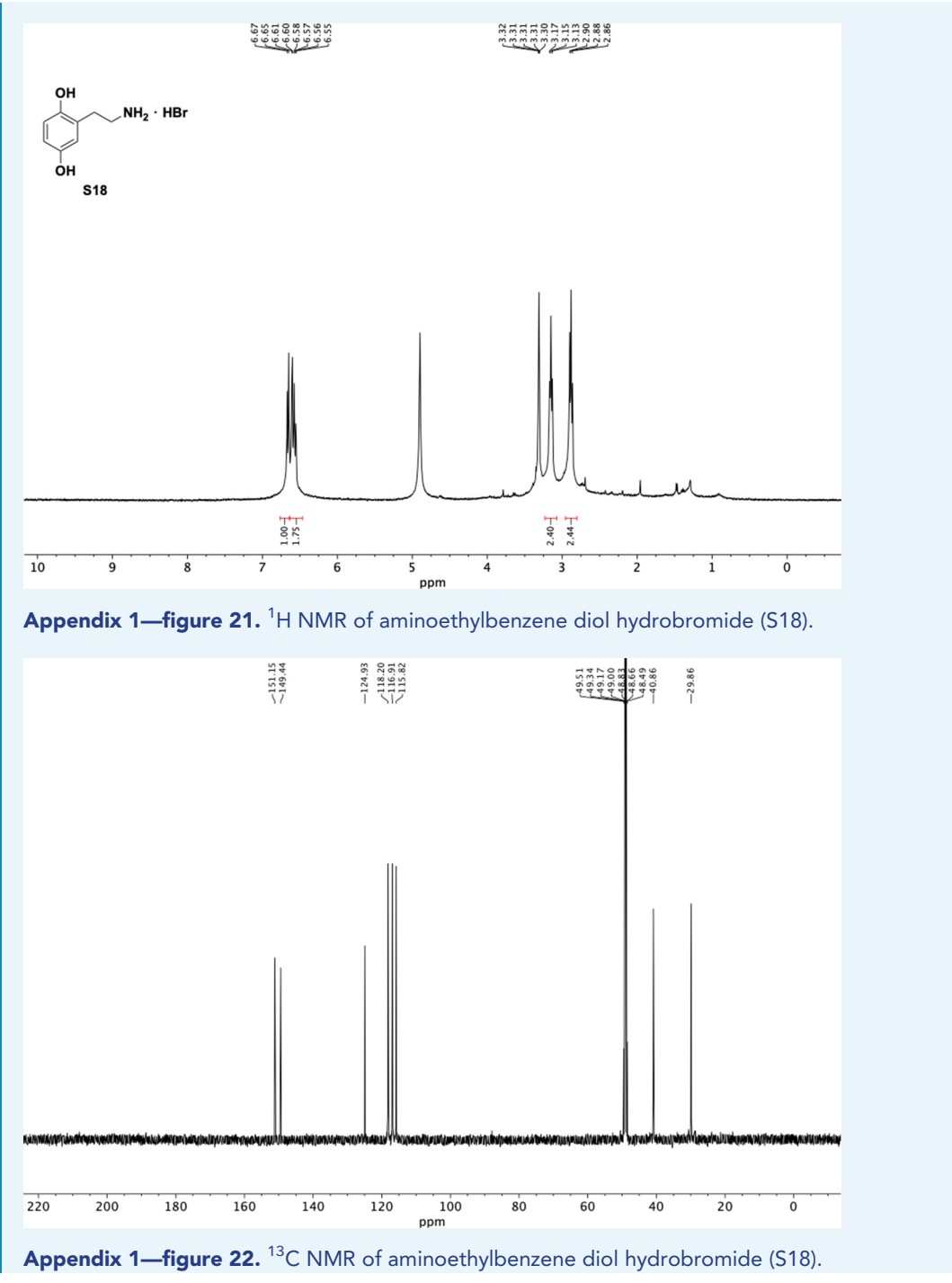

**Appendix 1—figure 21.** [1]H NMR of aminoethylbenzene diol hydrobromide (S18).

**Appendix 1—figure 22.** [13]C NMR of aminoethylbenzene diol hydrobromide (S18).

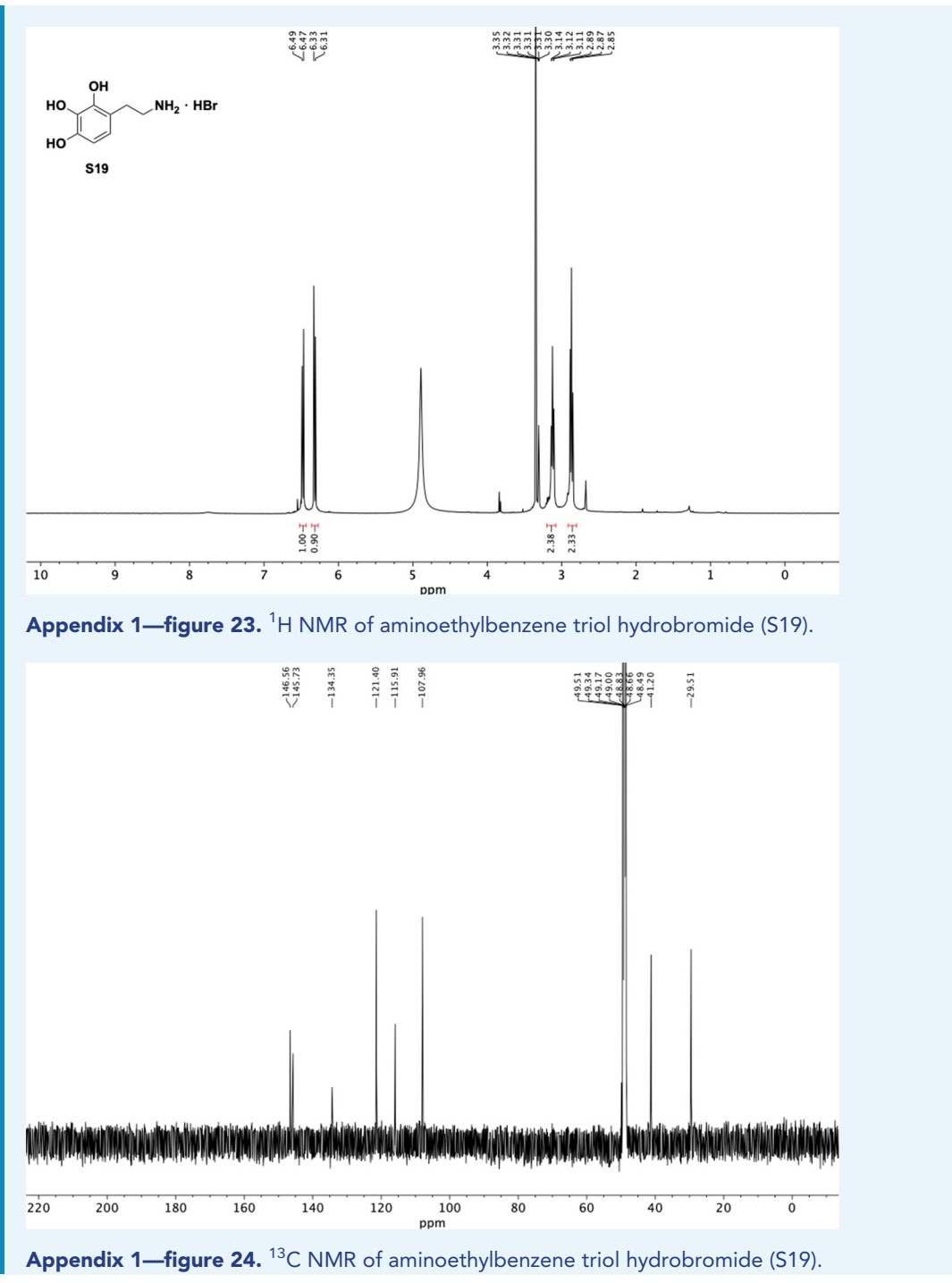

**Appendix 1—figure 23.** [1]H NMR of aminoethylbenzene triol hydrobromide (S19).

**Appendix 1—figure 24.** [13]C NMR of aminoethylbenzene triol hydrobromide (S19).

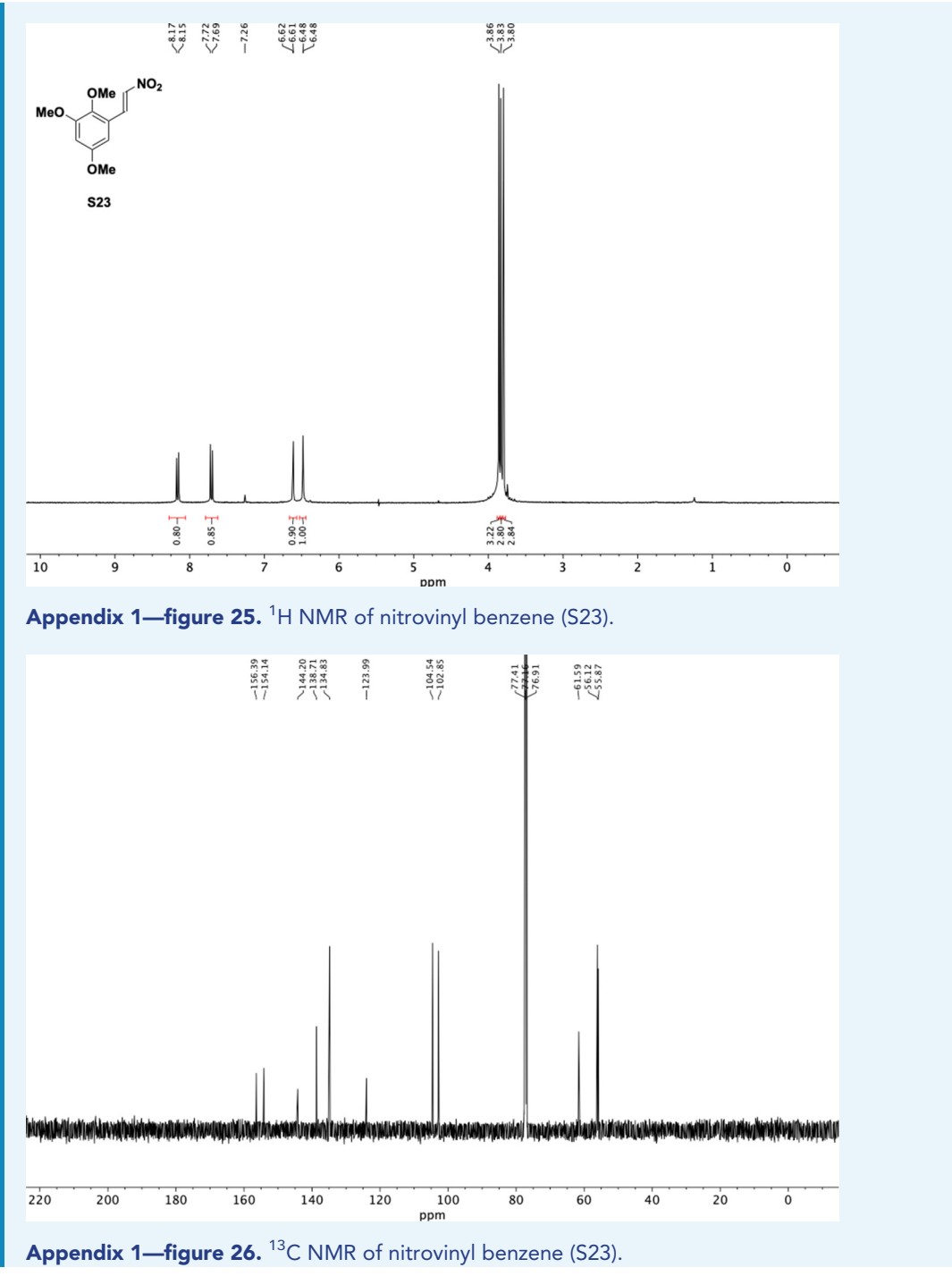

**Appendix 1—figure 25.** [1]H NMR of nitrovinyl benzene (S23).

**Appendix 1—figure 26.** [13]C NMR of nitrovinyl benzene (S23).

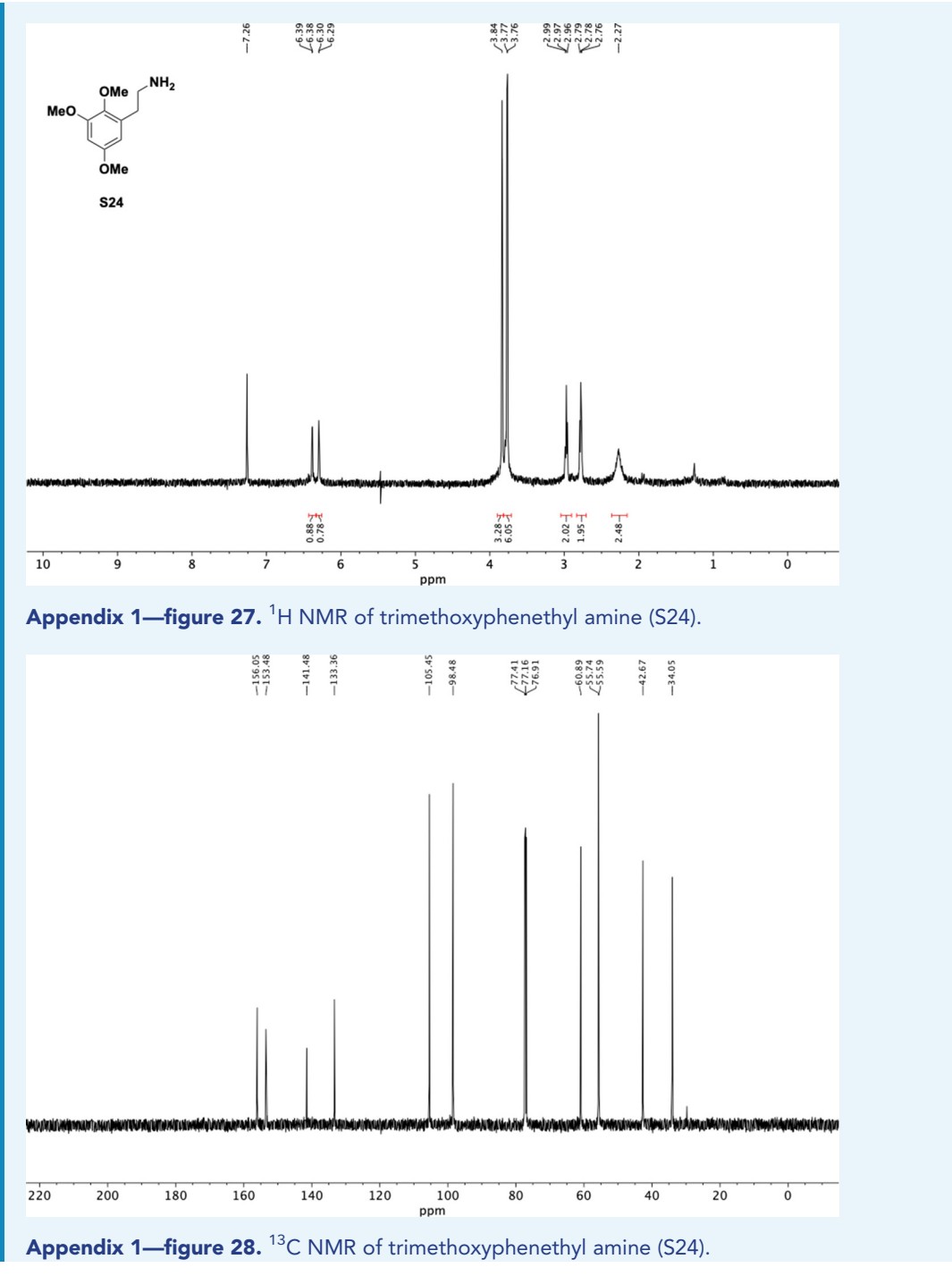

**Appendix 1—figure 27.** $^1$H NMR of trimethoxyphenethyl amine (S24).

**Appendix 1—figure 28.** $^{13}$C NMR of trimethoxyphenethyl amine (S24).

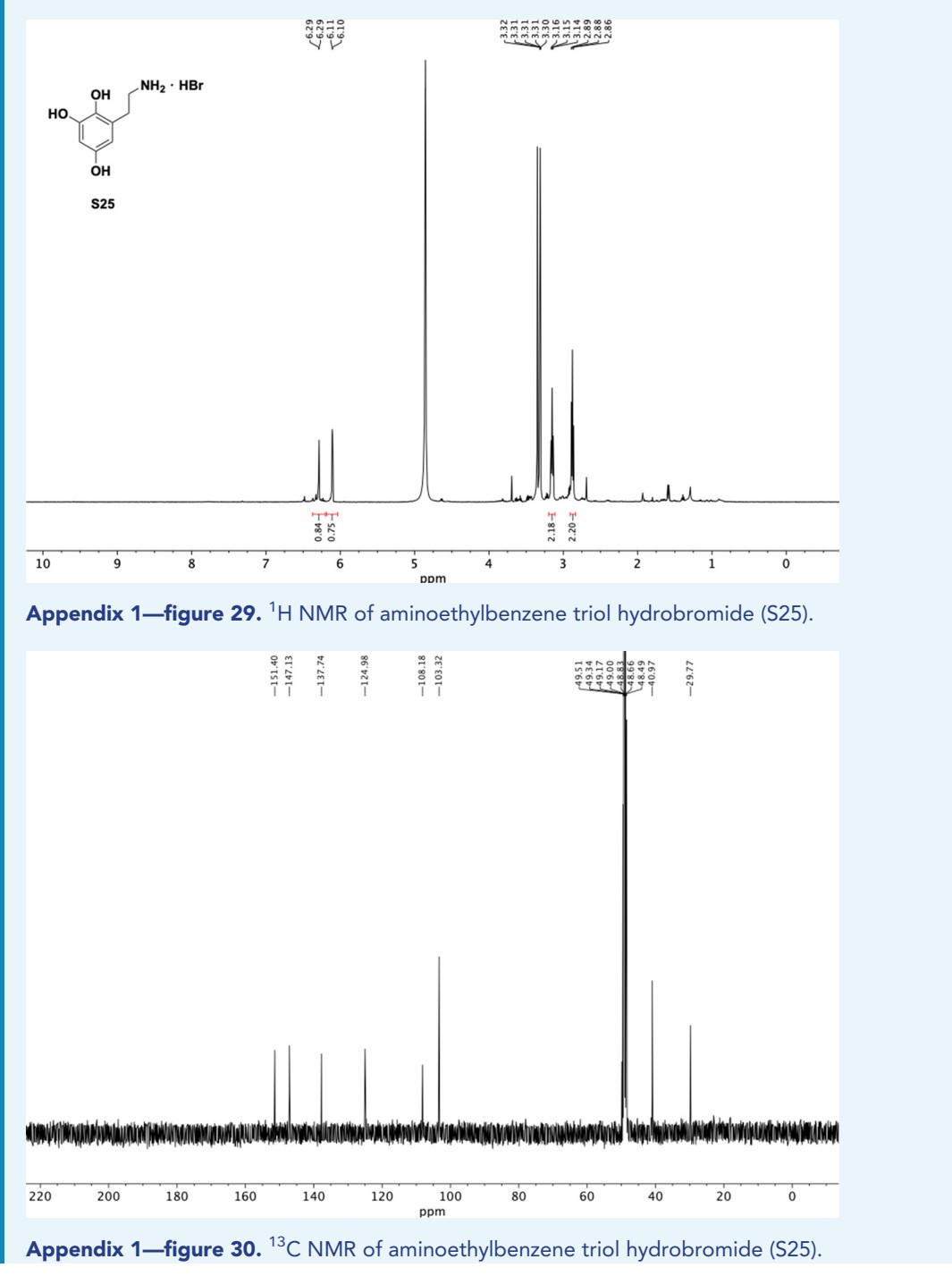

**Appendix 1—figure 29.** $^1$H NMR of aminoethylbenzene triol hydrobromide (S25).

**Appendix 1—figure 30.** $^{13}$C NMR of aminoethylbenzene triol hydrobromide (S25).

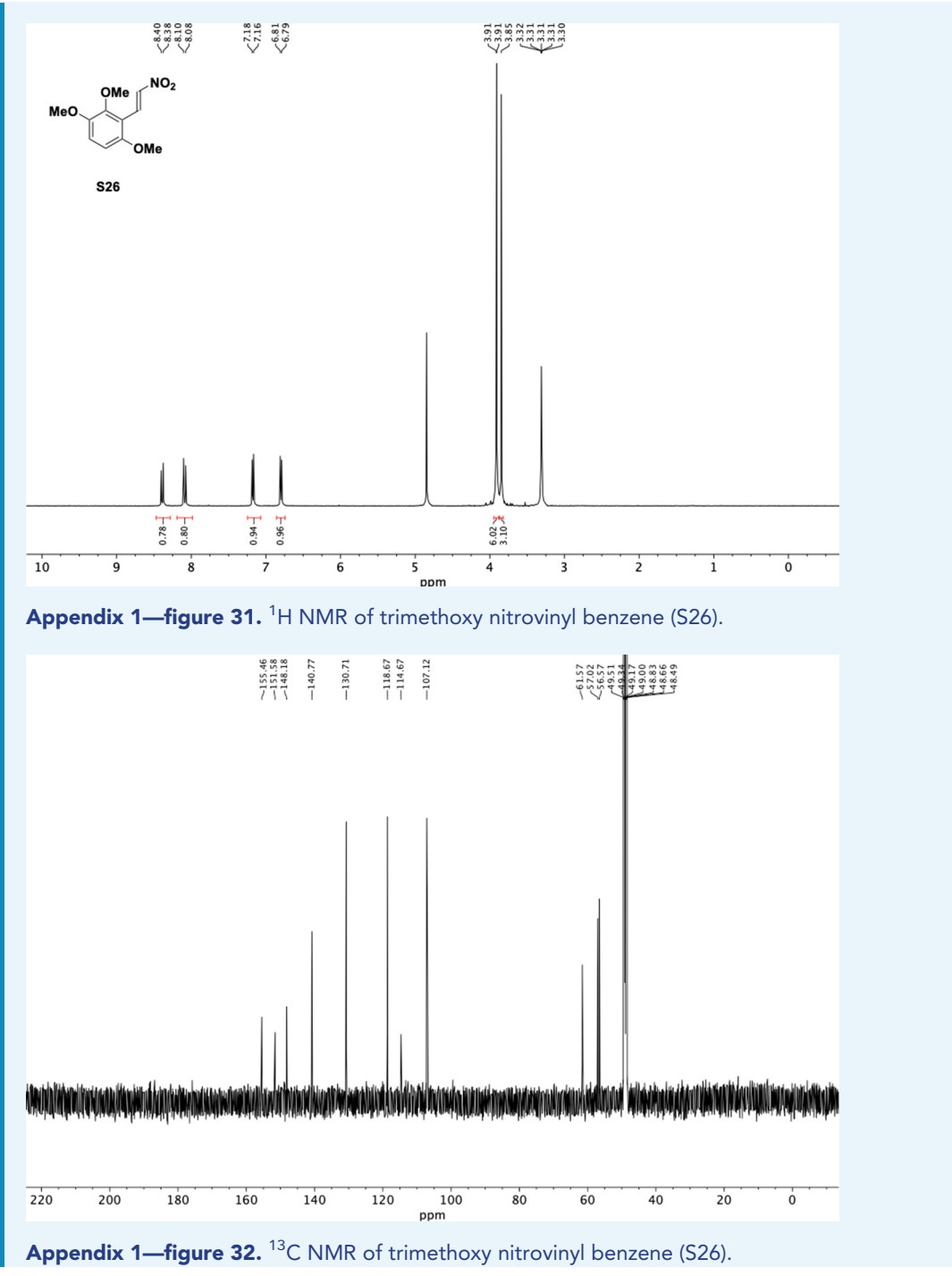

**Appendix 1—figure 31.** [1]H NMR of trimethoxy nitrovinyl benzene (S26).

**Appendix 1—figure 32.** [13]C NMR of trimethoxy nitrovinyl benzene (S26).

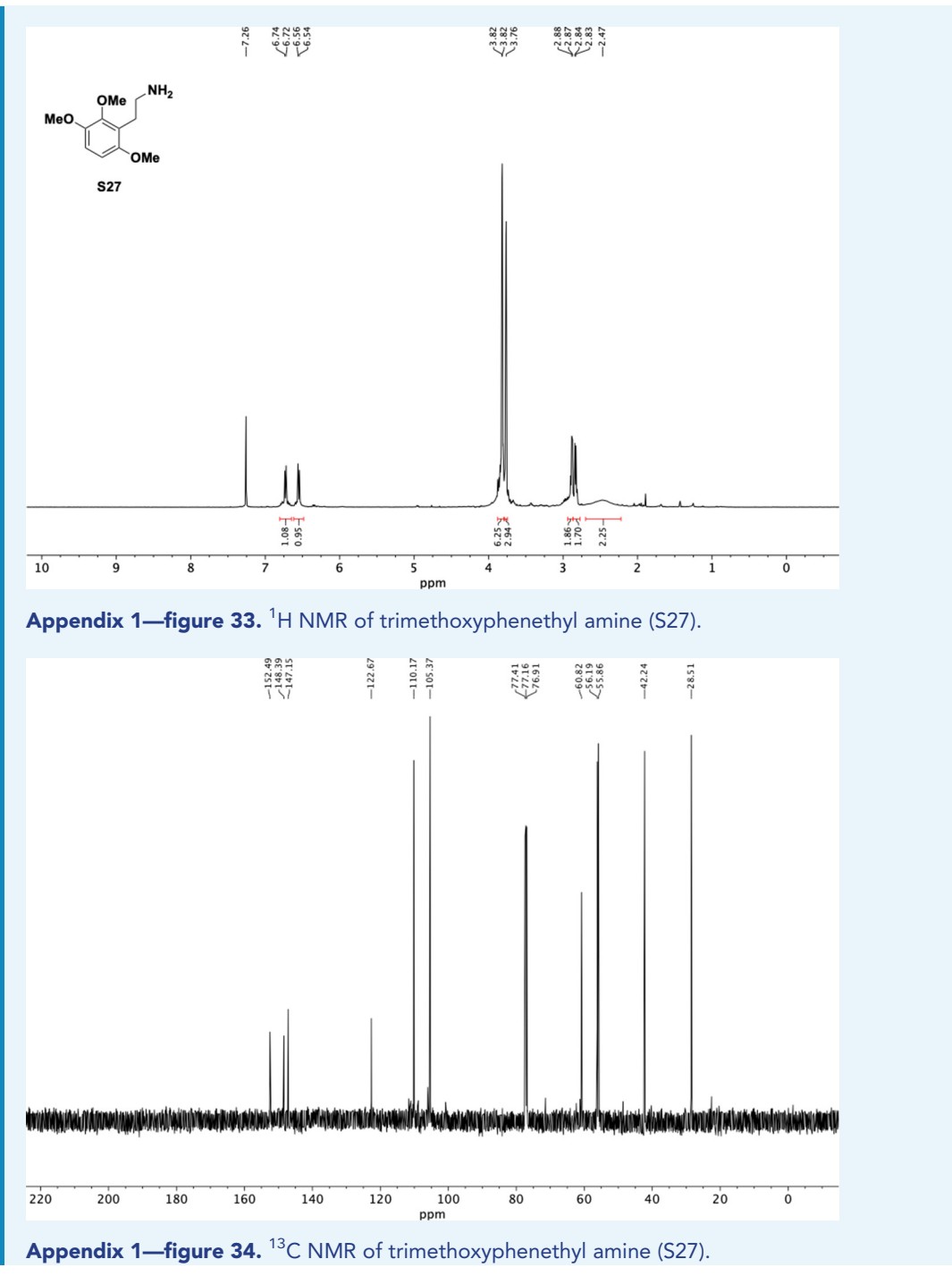

**Appendix 1—figure 33.** $^1$H NMR of trimethoxyphenethyl amine (S27).

**Appendix 1—figure 34.** $^{13}$C NMR of trimethoxyphenethyl amine (S27).

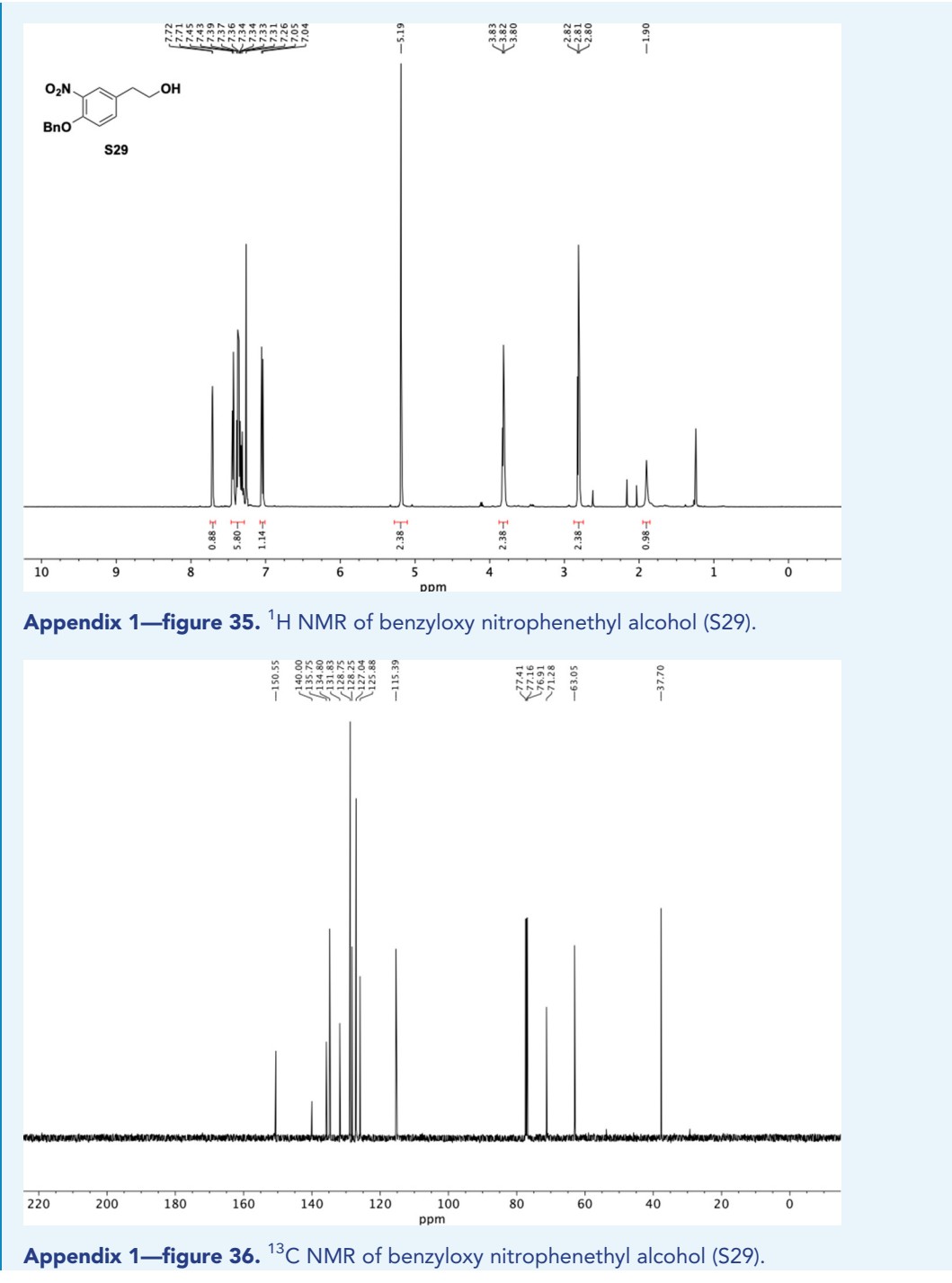

**Appendix 1—figure 35.** [1]H NMR of benzyloxy nitrophenethyl alcohol (S29).

**Appendix 1—figure 36.** [13]C NMR of benzyloxy nitrophenethyl alcohol (S29).

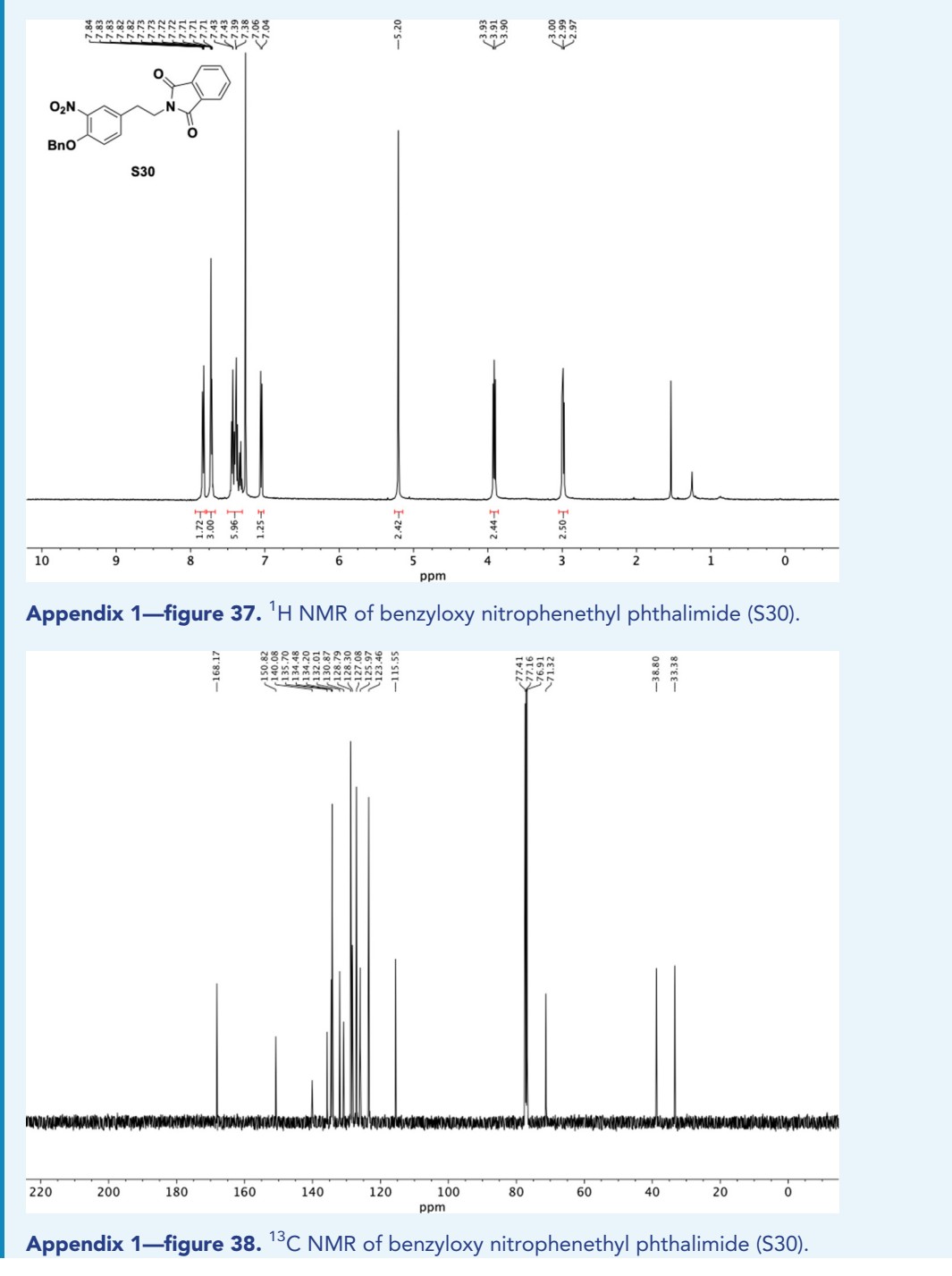

**Appendix 1—figure 37.** $^1$H NMR of benzyloxy nitrophenethyl phthalimide (S30).

**Appendix 1—figure 38.** $^{13}$C NMR of benzyloxy nitrophenethyl phthalimide (S30).

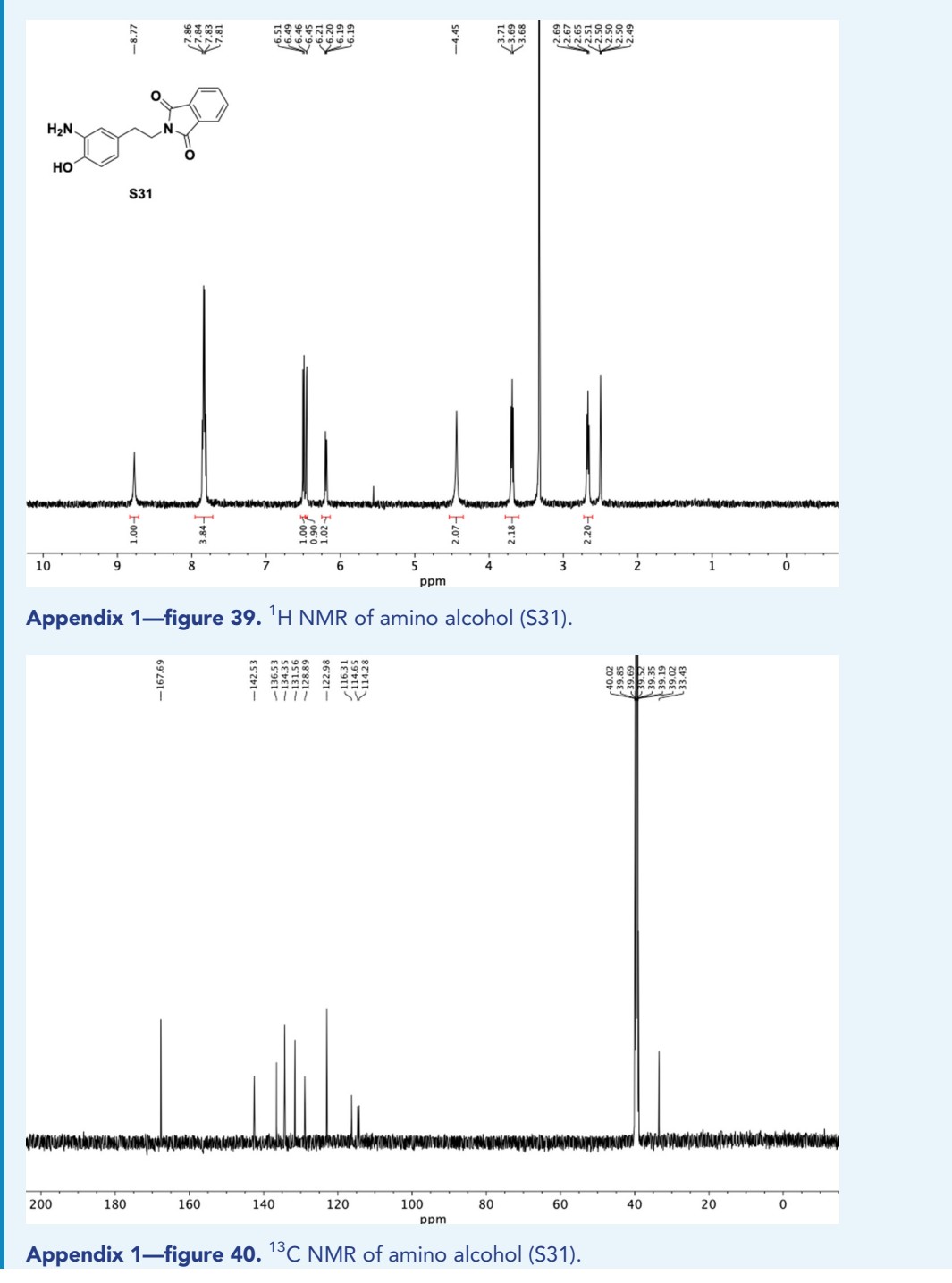

**Appendix 1—figure 39.** [1]H NMR of amino alcohol (S31).

**Appendix 1—figure 40.** [13]C NMR of amino alcohol (S31).

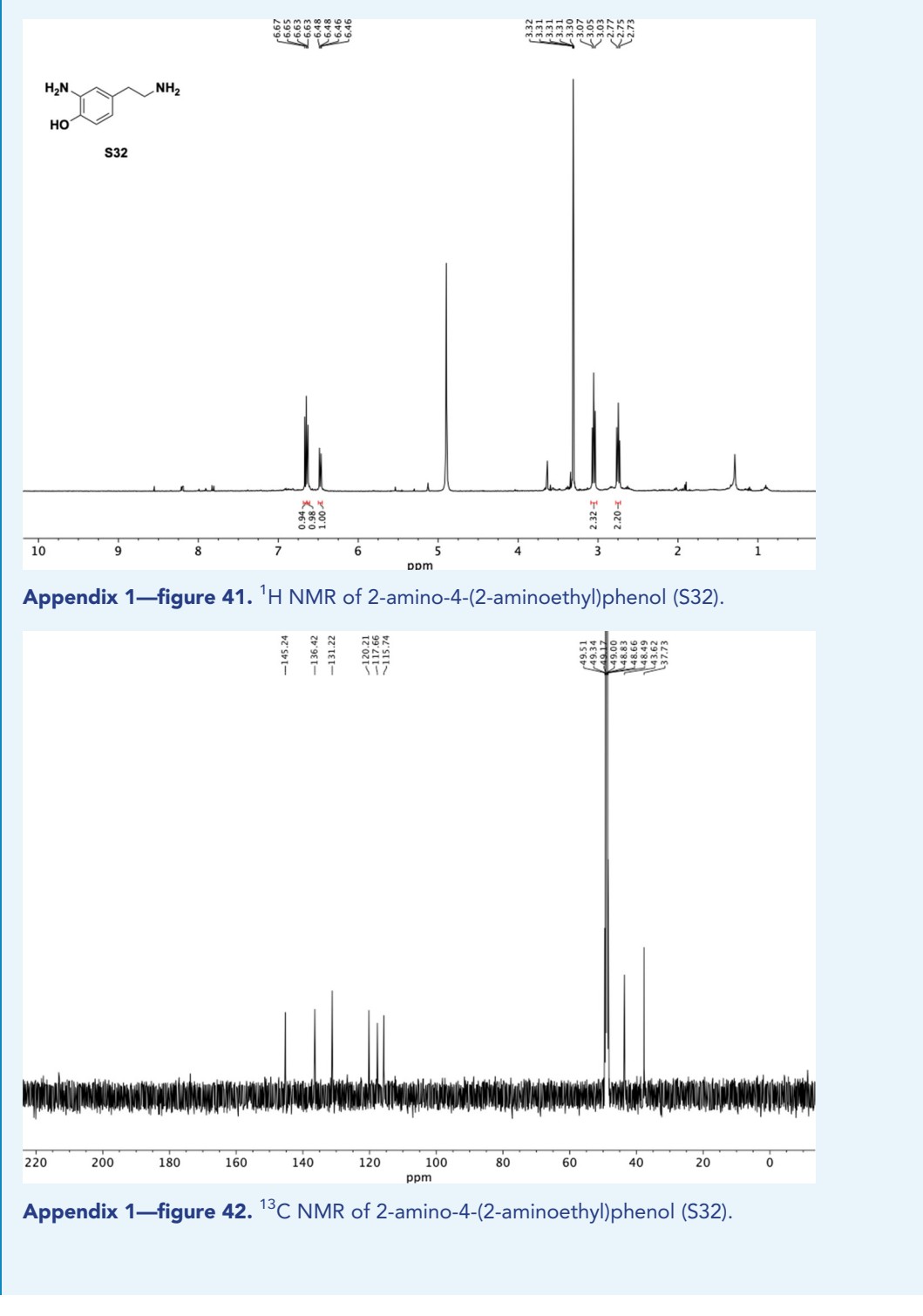

**Appendix 1—figure 41.** [1]H NMR of 2-amino-4-(2-aminoethyl)phenol (S32).

**Appendix 1—figure 42.** [13]C NMR of 2-amino-4-(2-aminoethyl)phenol (S32).

