## [Decision Letter]

**Acceptance summary:**

The manuscript builds on the authors recent study, in which they showed that a dopamine dehydroxylase (Dadh) from *Eggerthella lenta* degrades L-dopa, the precursor to the neurotransmitters dopamine, norepinephrine and epinephrine. In this manuscript, the authors first explore the substrate specificity of the *E. lenta* enzyme in more detail and find that it is rather narrow. Dadh from *E. lenta* degrades dopamine and norepinephrine, but not epinephrine or other, more distantly related catecholamines. Importantly, only dopamine and norepinephrine induce expression of Dadh, with both substrates promoting *E. lenta* growth by serving as alternative electron acceptors. The authors then use similar reasoning as they used for identification of *E. lenta* Dadh to prospect for additional Dadh-related enzymes from Actinobacteria that might dehydroxylate other catechols. The authors identify such activities for three other catechols including DOPAC, catechin and hydrocaffeic acid in Gordonia pamelaeae and *E. lenta*. Bioinformatic analyses and activity assays with a variety of bacterial communities indicate that these catechol hydroxylases are widely distributed. For the revision, the authors added biochemical data that specifically link the hcdh gene to hydrocaffeic acid dehydroxylation, and they provided additional data in support of the metabolic functions identified in low-abundance gut Actinobacteria being prevalent but variably distributed among human gut microbiotae. In aggregate, this work has important implications not only for the turnover of medical drugs in human and animal guts, but likely also for many other microbial ecosystems, since catechols comprise a widely distributed class of molecules.

**Decision letter after peer review:**

Thank you for submitting your article "A widely distributed metalloenzyme class enables gut microbial metabolism of host-and diet-derived catechols" for consideration by *eLife*. Your article has been reviewed by three peer reviewers, and the evaluation has been overseen by Detlef Weigel as Reviewing and Senior Editor. The following individual involved in review of your submission has agreed to reveal her identity: Ruth Ley (Reviewer #3).

The reviewers and I have discussed the reviews, and I have drafted this decision to help you prepare a revised submission.

Summary:

The current work builds on your recent study, in which you showed that a dopamine dehydroxylase (Dadh) from *Eggerthella lenta* degrades L-dopa, the precursor to the neurotransmitters dopamine, norepinephrine and epinephrine. In the current work, you first explore the substrate specificity of the *E. lenta* enzyme in more detail and find that it is rather narrow. Dadh from *E. lenta* degrades dopamine and norepinephrine, but not epinephrine or other, more distantly related catecholamines. Importantly, only dopamine and norepinephrine induce expression of Dadh, with both substrates promoting *E. lenta* growth by serving as alternative electron acceptors.

You then use similar reasoning as you used for identification of *E. lenta* Dadh to prospect for additional Dadh-related enzymes from Actinobacteria that might dehydroxylate other catechols. You identify such activities for three other catechols including DOPAC, catechin and hydrocaffeic acid in Gordonia pamelaeae and *E. lenta*. Bioinformatic analyses and activity assays with a variety of bacterial communities indicate that these catechol hydroxylases are widely distributed. In aggregate, this work has important implications not only for the turnover of medical drugs in human and animal guts, but likely also for many other microbial ecosystems, since catechols comprise a widely distributed class of molecules.

Essential revisions:

The reviewers were all very positive, but also agreed that the work would be enhanced by more definitive data showing directly that at least one of the newly identified enzymes is responsible for the observed chemical transformations. This could come either from in vitro assays with purified protein or genetic assays. One of the major conclusions of the manuscript is that catechol dehydroxylation is performed by distinct subtypes of molybdenum-dependent enzymes, but this statement is only fully supported if you can unequivocally show that the identified enzymes perform the chemical transformation they suggested.

A more general suggestion is that the disparate nature of the data make for a somewhat disconnected presentation. We would strongly encourage you to think of ways to connect the different data sources more strongly. For example, given the isolate data, from generally low abundance Actinobacteria, how prevalent do you think these functions actually are in human guts? Your human gut data (Figure 2E) suggest that individuals can have log order differences in Dadh copy number and that some ex vivo communities are responsive to dopamine, while others are not. Why might this be? Are there other enzymes, such as those in the SSN, that might perform this function? In Figure 2—figure supplement 7, *E. lenta* counts do not increase in response to dopamine in all ex vivo samples – what might be behind this observation? The observed functional and metabolic diversity across human individuals is obviously fascinating and more could be done to discuss it in the context of the rest of the paper.

Notwithstanding these comments, there was general agreement that this is a fine example of the integrated use of different approaches to elucidate a previously mysterious metabolic activity of the gut microbiome (catechol dehydroxylation), both by discovering the family of enzymes responsible for it and demonstrating its apparent benefit for the encoding bacteria.

---

## [Author Response]

Essential revisions:The reviewers were all very positive, but also agreed that the work would be enhanced by more definitive data showing directly that at least one of the newly identified enzymes is responsible for the observed chemical transformations. This could come either from in vitro assays with purified protein or genetic assays. One of the major conclusions of the manuscript is that catechol dehydroxylation is performed by distinct subtypes of molybdenum-dependent enzymes, but this statement is only fully supported if you can unequivocally show that the identified enzymes perform the chemical transformation they suggested.

We thank the reviewers for this excellent suggestion and agree that our conclusions would be enhanced by more definitive data linking one of the candidate dehydroxylases to its proposed biochemical activity.

First, we want to clarify that the gut Actinobacteria used in our study are genetically intractable, preventing the use of genetic knock-out studies in metabolizing strains or use of plasmids to express enzymes in non-metabolizing hosts. Therefore, we feel that chemical genetics (tungstate-dependent inhibition of catechol dehydroxylation) and comparative genomics are the best proxies for traditional knock-out/knock-in experiments.

Heterologous expression is similarly challenging due to the complex assembly of molybdenum-dependent enzymes, which require active chaperones for insertion of the co-factors and, for membrane-anchored proteins such as those identified in *Eggerthella*, requires translocation of the folded protein across the cell membrane by means of the Twin Arginine Translocation (TAT) system (Iobbi-Nivol and Leimkuhler, 2013). In our previous study on the discovery of the dopamine dehydroxylase (*dadh*) (Rekdal et al., 2019), we reported attempting heterologous expression of *dadh*, trying >20 constructs and multiple heterologous hosts. All of these efforts failed to provide active, soluble protein. During the submission and revision of this manuscript, we attempted heterologous expression of *hcdh, dodh,* and *cldh*, again exploring multiple constructs and phylogenetic hosts (*E. coli* and *Shewanella* sp. ANA-3), without successfully obtaining soluble, active protein. These negative results likely reflect the complexity of assembling these multi-subunit metalloenzymes, which likely requires system-specific chaperones and maturation factors that are present in the native organisms but may not be present in the heterologous host.

To link one of the additional candidate dehydroxylases identified in this study to its proposed biochemical activity, we turned to native purification, the strategy we used to confirm the identity of the dopamine dehydroxylase. We focused on the hydrocaffeic acid dehydroxylase (Hcdh) from *E. lent*a A2, using an anaerobic activity-based purification strategy to identify the protein responsible for hydrocaffeic acid dehydroxylation. Following ammonium sulfate precipitation, hydrophobic interaction chromatography, anion exchange chromatography, and size exclusion chromatography, we isolated a combined set of fractions that contained five major bands as assessed by SDS-PAGE (Figure 4—figure supplement 4) and quantitatively dehydroxylated hydrocaffeic acid into *m*-hydroxyphenylacetic acid (*m*-HPPA) under anaerobic conditions (Figure 4D). As in our cell suspension assays, oxygen inhibited the enzymatic activity (Figure 4D and Figure 4—figure supplement 1). We cut out the band with the apparent correct size (130.5 kDa) (Figure 4—figure supplement 4) and confirmed using proteomics that this band harbored the molybdenum-dependent enzyme that we had assigned as the hydrocaffeic acid dehydroxylase (Hcdh, Elenta-A2_02815) (Figure 4—figure supplement 5 and Supplementary file 2F) based on our RNA-sequencing, tungstate inhibition, and comparative genomics data. The protein band also contained three other proteins, suggesting that these proteins elute close to each other on the SDS-page gel. The Hcdh proteomics data suggested that we recovered peptides with >60% coverage of the Hcdh peptide sequence, and notably the coverage starts at the very end of the predicted TAT signal sequence cleavage site, which strongly suggests the immature Hcdh protein (134 kDa) undergoes TAT signal sequence cleavage as part of maturation into its final form (130.5 kDa) (Figure 4—figure supplement 5).

Having established the presence of Hcdh, we diluted the fractions 1:5 in buffer and performed an additional set of independent enzyme reactions to evaluate the preliminary substrate scope of Hcdh. Consistent with our lysate data (Figure 4A), the enzyme only dehydroxylated only hydrocaffeic acid and did not accept dopamine, (+)-catechin, or DOPAC (Figure 4E).

As the original anaerobic native purification protocol for the dopamine dehydroxylase took more than five months to develop, we believe that further optimization is necessary to increase the yield and purity of the Hcdh protein. However, we are confident that this preliminary experiment provides initial biochemical evidence that links the *hcdh* gene to hydrocaffeic acid dehydroxylation, further supporting the proposal that distinct enzymes dehydroxylate distinct substrates.

We have updated the main text (subsection “Gut Actinobacteria dehydroxylate individual catechols using distinct enzymes”, sixth paragraph, and Figure 4D, E) and Figure 4—figure supplements 4 and 5, Supplementary file 2F) to reflect these new findings.

Finally, during the revision, we performed additional follow-up experiments screening all members of the gut Actinobacterial library for metabolism of DOPAC, (+)-catechin, and hydrocaffeic acid using LC-MS analysis. Our repetition experiment corroborated our previous findings, except in the case of (+)-catechin metabolism by *Eggerthella lenta* AB12#2. We had previously found that this strain lacked metabolism, but we now found it displayed low metabolism (<10%) towards this substrate. This strain does not encode the putative catechin dehydroxylase, suggesting that in this strain another enzyme might be responsible for the observed metabolism. As we have now confirmed the assignment of the Hcdh enzyme and the molybdenum-dependence of this reaction in *E. lenta*, we do not feel that this discrepancy alters the major conclusions of our study. We have therefore updated the main text (subsection “Gut Actinobacteria dehydroxylate individual catechols using distinct enzymes, third paragraph, Figure 4C, and Figure 4—source data 1 to reflect these new findings.

A more general suggestion is that the disparate nature of the data make for a somewhat disconnected presentation. We would strongly encourage you to think of ways to connect the different data sources more strongly.

We thank the reviewers for this suggestion and agree that a more integrated discussion and highlight of the different experiments could make for a more compelling presentation. We have taken your suggestions into account in an effort to connect data sources across experiments as described in the responses to the specific individual comments provided below.

For example, given the isolate data, from generally low abundance Actinobacteria, how prevalent do you think these functions actually are in human guts?

It is true that the Actinobacteria in which we identity the candidate catechol dehydroxylases are generally found at low abundances in human samples. However, the distribution of catechol dehydroxylation in the human gut microbiota remained an unanswered question that was not addressed in our original submission.

To address this question, we have conducted two additional experiments.

First, we explored the distribution and abundance of *dadh, hcdh, cadh, dodh,* and the recently identified catechol lignan dehydroxylase *cldh* (Bess et al., 2019)among >1800 publicly available human gut metagenomes (Nayfach, Fischbach and Pollard, 2015). Although these catechol dehydroxylases are found at generally low abundances in these metagenomes, they differ in their distribution. *Dadh* and *hcdh* are the most prevalent (in >70% and >90% of individuals, respectively), followed by *cadh* (30%), and *dodh* (20%) and *cldh* (25%) (Figure 5A). Notably, the variable prevalence of catechol dehydroxylase-encoding genes in these metagenomes was consistent with their distribution among individual human gut Actinobacterial isolates (Figure 4C). This analysis ties together our experiments with individual human gut Actionbacteria and our work with complex communities, suggesting that gene prevalence among individual isolates is relevant to the broader prevalence in gut metagenomes.

Second, we incubated complex gut microbiota communities from unrelated humans (n=12) ex vivo with hydrocaffeic acid, (+)-catechin, and stable-isotope deuterium-labeled dopamine and DOPAC and analyzed dehydroxylation by LC-MS/MS. In this experiment, we observed dehydroxylation of dopamine, hydrocaffeic acid, and DOPAC across the majority of subjects (n=10, n=11, n=12, respectively), indicating that the metabolic activities of low-abundance gut Actinobacteria are observed in complex gut communities (Figure 5B-D). However, catechol metabolism varied between compounds and also between subjects, with some individuals metabolizing all compounds, and some metabolizing none (Figure 5B-D). We found that (+)-catechin was depleted without production of the corresponding dehydroxylated metabolites, consistent with this compound undergoing a wide range of metabolic reactions in complex communities (Figure 5—source data 1) (Takagaki and Nanjo, 2013); van't Slot and Humpf, 2009; Aura et al., 2008). To investigate whether the metabolism correlated with the presence of specific dehydroxylase enzymes, we further investigated DOPAC dehydroxylation. We separated the 12 samples into a group of 9 metabolizers and 3 non-metabolizers (in which no biological replicate displayed dehydroxylation activity). qPCR enumeration in these cultures revealed that the abundance of the candidate DOPAC dehydroxylase gene *dodh* discriminated metabolizing and nonmetabolizing subjects (p<0.001, unpaired t-test) (Figure 4E), and correlated significantly with dehydroxylation activity within the 9 metabolizers (Pearson’s correlation, r=0.73, R^2^=0.53, p<0.05) (Figure 4F). This gene-activity correlation is consistent with our previous observations that *dadh* SNP status correlates with dopamine metabolism in human subjects ex vivo(Rekdal et al., 2019).

Altogether, these data establish the metabolic functions identified in low-abundance gut Actinobacteria are prevalent but variably distributed among human gut microbiotas, testifying to a genetic and metabolic diversity among individuals. These data also strengthen the case for our assignment of *dodh* as a candidate DOPAC dehydroxylase by demonstrating a link between this gene and DOPAC dehydroxylation by complex human gut microbial communities.

We have updated the main text (subsection “Catechol dehydroxylases are variably distributed in metagenomes and correlate with metabolism by complex gut microbiota samples ex vivo”; Figure 5) to include these new findings.

Your human gut data (Figure 2E) suggest that individuals can have log order differences in Dadh copy number and that some ex vivo communities are responsive to dopamine, while others are not. Why might this be? Are there other enzymes, such as those in the SSN, that might perform this function? In Figure 2—figure supplement 7, E. lenta counts do not increase in response to dopamine in all ex vivo samples – what might be behind this observation?

We appreciate these questions and agree that this is an interesting question. We want to clarify that the overarching goal of this experiment was not to investigate the variable abundance of *dadh* and *E. lenta* across human samples, but rather to investigate the response of these human gut communities to dopamine in an electron acceptor-limited growth medium. Together with our experiments with defined communities, our qPCR and *dadh* SNP data are consistent with the proposal that dopamine increases the fitness of dopamine-dehydroxylating *E. lenta* strains in microbial communities by serving as an alternative electron acceptor.

Regarding the comment regarding the variability in *dadh* copy number abundance, we believe that baseline variability in copy number abundance across human individuals likely reflects differing abundance of *E. lenta*, which is supported by our metagenomic results. Further analysis and experimentation beyond the scope of this study would be required to identify the factors shaping this distribution.

Multiple factors could account for the small number of samples that do not display an increase in *E. lenta* or *dadh* abundance in response to dopamine. These include: 1) the complex gut microbial community supported *E. lenta* growth sufficiently such that dopamine does not add any additional advantage, as is seen in BHI medium where *E. lenta* obtains no growth benefit from dopamine (Figure 1—figure supplement 1) 2) enzymes or organisms not targeted by our qPCR primers were responsible for metabolism in these communities 3) Dopamine could be metabolized through other pathways, which our experimental setup did not account for as we used the colorimetric assay for catechol detection to screen for metabolism.

We have updated the main text to discuss this observation (subsection “Dopamine promotes gut bacterial growth by serving as an alternative electron acceptor”).

The observed functional and metabolic diversity across human individuals is obviously fascinating and more could be done to discuss it in the context of the rest of the paper.

The functional and metabolic diversity in catechol dehydroxylation is a notable observation from our study, and we thank the reviewers for pointing out its significance. In addition to conducting the metagenomic analysis and ex vivo experiments described above, we have now added a paragraph to our Discussion section that describes the potential implications of the genetic and metabolic diversity seen across individual gut bacterial strains, human gut communities, and mammalian microbiota samples (subsection “Catechol dehydroxylase reactivity is present across the gut microbiotas of mammals representing distinct diets and phylogenetic origins”).

Notwithstanding these comments, there was general agreement that this is a fine example of the integrated use of different approaches to elucidate a previously mysterious metabolic activity of the gut microbiome (catechol dehydroxylation), both by discovering the family of enzymes responsible for it and demonstrating its apparent benefit for the encoding bacteria.

We thank the reviewers for recognizing the significance of our work and agree that our study is strong example for the integration of diverse approaches to elucidate the molecular basis of an elusive but prominent metabolic function of the gut microbiota.